# Tropospheric Ozone Precursors: Global and Regional Distributions, Trends, and Variability

Yasin Elshorbany[1*], Jerald R. Ziemke[2], Sarah Strode[2,3], Hervé Petetin[4], Kazuyuki Miyazaki[5], Isabelle De Smedt[6], Kenneth Pickering[7], Rodrigo J. Seguel[8], Helen Worden[9], Tamara Emmerichs[10], Domenico Taraborrelli[10], Maria Cazorla[11], Suvarna Fadnavis[12], Rebecca R. Buchholz[9], Benjamin Gaubert[9], Néstor Y. Rojas[13], Thiago Nogueira[14], Thérèse Salameh[15], Min Huang[16]

*Correspondence to: elshorbany@usf.edu

[1] School of Geosciences, College of Arts and Sciences, University of South Florida, St. Petersburg, FL, USA

[2] NASA Goddard Space Flight Center, Greenbelt, Maryland, USA

[3] Goddard Earth Sciences Technology and Research (GESTAR II), Maryland, USA

[4] Earth Sciences Department, Barcelona Supercomputing Center, Barcelona, Spain

[5] Jet Propulsion Laboratory, California Institute of Technology, Pasadena CA

[6] BIRA-IASB, Ringlaan 3 Av. Circulaire, 1180 Brussels, Belgium

[7] Dept. of Atmospheric and Oceanic Science, University of Maryland, College Park, MD USA

[8] Center for Climate and Resilience Research, Department of Geophysics, Faculty of Physical and Mathematical Sciences University of Chile, Chile.

[9] Atmospheric Chemistry Observations & Modeling Laboratory (ACOM), NSF National Center for Atmospheric Research (NSF NCAR), Boulder CO, USA.

[10] Institute of Climate and Energy Systems, ICE-3: Troposphere, Forschungszentrum Jülich GmbH, Jülich, Germany

[11] Universidad San Francisco de Quito USFQ, Instituto de Investigaciones Atmosféricas, Diego de Robles y Av Interoceánica, Quito, Ecuador.

[12] Center for Climate Change Research, Indian Institute of Tropical Meteorology, MoES, Pune, India.

[13] Department of Chemical and Environmental Engineering, Universidad Nacional de Colombia, Bogota, Colombia.

[14] University of São Paulo, São Paulo, Brazil.

[15] IMT Nord Europe, Institut Mines-Télécom, Univ. Lille, Centre for Energy and Environment, 59000, Lille, France.

[16] Earth System Science Interdisciplinary Center, University of Maryland, College Park, MD, USA.

**Abstract**
Tropospheric ozone results from in-situ chemical formation and stratosphere-troposphere
exchange (STE), with the latter being more important in the middle and upper troposphere than in
the lower troposphere. Ozone photochemical formation is nonlinear, and results from the oxidation
of methane and non-methane hydrocarbons (NMHCs) in the presence of nitrogen oxide
($NO_x$=NO+$NO_2$). Previous studies showed that $O_3$ short- and long-term trends are nonlinearly
controlled by near-surface anthropogenic emissions of carbon monoxide (CO), volatile organic
compounds (VOCs), and nitrogen oxides, which maybe also impacted by the long-range transport
(LRT) of $O_3$ and its precursors. In addition, several studies have demonstrated the important role
of STE in enhancing ozone levels, especially in the midlatitudes. In this article, we investigate
tropospheric ozone spatial variability and trends from 2005 to 2019 and relate those to ozone
precursors on global and regional scales. We also investigate the spatiotemporal characteristics of
the ozone formation regime in relation to ozone chemical sources and sinks. Our analysis is based
on remote sensing products of the Tropospheric Column of Ozone (TrC-$O_3$) and its precursors,
nitrogen dioxide (TrC-$NO_2$), formaldehyde (TrC-HCHO), and total column of CO (TC-CO) as
well as ozonesonde data and model simulations. Our results indicate a complex relationship
between tropospheric ozone column levels, surface ozone levels, and ozone precursors. While the
increasing trends of near-surface ozone concentrations can largely be explained by variations in
VOC and $NO_x$ concentration under different regimes, TrC-$O_3$ may also be affected by other
variables such as tropopause height and STE as well as LRT. Decreasing or increasing trends in
TrC-$NO_2$ have varying effects on the TrC-$O_3$, which is related to the different local chemistry in
each region. We also shed light on the contribution of $NO_x$ lightning and soil NO and nitrous acid
(HONO) emissions to trends of tropospheric ozone on regional and global scales.

## 1. Introduction

Tropospheric ozone ($O_3$) is an important air pollutant due to its diverse effects on air quality, ecosystem (Mills et al., 2018), health (Lefohn et al., 2018; Fleming et al., 2018), and climate (Boucher et al., 2013; Myhre et al., 2013; Zanis et al., 2022). $O_3$ is a photochemical product that results from the oxidation of methane ($CH_4$) and non-methane hydrocarbons (NMHCs) in the presence of nitrogen oxides ($NO_x$). Tropospheric ozone burdens can also be affected by stratosphere-troposphere exchange (STE) (Stohl et al., 2003; Zeng et al., 2010; Trickl et al., 2011; Li et al., 2024) and long-range transport (LRT) of ozone (e.g., Hov et al., 1978; Ravetta et al., 2007; Itahashi et al., 2020). $O_3$ is considered a short-lived climate forcer (SLCF) and is the third-most important greenhouse gas with an effective radiative forcing of $(0.47^{+0.23}_{-0.23})$ W $m^{-2}$; Forster et al., 2021). Since the mid-1990s, free tropospheric ozone trends based on in situ measurement and satellite retrievals have increased with high confidence (HC) by 1-4 nmol $mol^{-1}$ $decade^{-1}$ across the northern mid-latitudes and 1-5 nmol $mol^{-1}$ $decade^{-1}$ within the tropics (Gulev et al., 2021). In the Southern Hemisphere, with more limited observation coverage compared with the Northern Hemisphere, the tropospheric column ozone shows an increase since the mid-1990s by less than 1 nmol $mol^{-1}$ $decade^{-1}$ with medium confidence at southern mid-latitudes (Gulev et al., 2021, Cooper at al., 2020). Tropospheric $O_3$ short- and long-term trends are nonlinearly controlled by anthropogenic emissions of carbon monoxide (CO), volatile organic compounds (VOCs), and nitrogen oxides ($NO_x=NO+NO_2$) as well as STE, especially in the midlatitudes (Li et al., 2024). Meteorological parameters such as wind speed and wind direction may also enhance the LRT of $O_3$, affecting regional ozone burdens, especially in the free troposphere (e.g., Glotfelty et al, 2014; Itahashi et al., 2020). Methane, with an assessed total atmospheric lifetime of $9.1 \pm 0.9$ years (Szopa et al., 2021), is also a crucial driver of tropospheric ozone (Fiore et al., 2002; Isaksen et al., 2014). Its accelerated growth rate of $7.6 \pm 2.7$ nmol $mol^{-1}$ $yr^{-1}$ between 2010 and 2019 (Canadell et al., 2021) is largely driven by anthropogenic activities (Szopa et al., 2021). NOAA GML observations of methane (NOAA, 2024) show that methane concentrations in the atmosphere have increased sharply since 2005 (an 8% increase from 2005 to 2023). Future scenarios show that emission control measures can influence future changes to air pollutants. Although the global increases in $CH_4$ abundance may offset benefits to surface $O_3$ from local emission reductions (Fiore et al., 2002; Shindell et al., 2012; Wild et al., 2012; Szopa et al., 2021), recent reports (e.g., Itahashi et al., 2020; Zanis et al., 2022), showed the dominant role of precursor emission changes in projecting surface ozone concentrations under future climate change scenarios. In this study, we investigate the relation between ozone trends and the trends of its precursors, with a focus on $NO_2$, CO, and HCHO.

Coupled Model Intercomparison Project Phase 6 (CMIP6) overestimates observed surface $O_3$ concentrations in most regions, with larger variability over Northern Hemisphere (NH) continental regions (e.g., Tarasick et al., 2019; Turnock et al., 2020). CMIP6 models simulate large increasing trends of surface concentrations of $O_3$ and $PM_{2.5}$ in East and South Asia with an annual mean increase of up to 40 ppb and 12 $\mu gm^{-3}$, respectively, over the historical periods (1850-2014; Turnock et al., 2020). However, these studies found also that CMIP6 models consistently underestimate $PM_{2.5}$ concentrations in the NH, especially during the winter months, and with larger variability near natural source regions, indicating missing sources (e.g., HONO) of $O_3$ (e.g., Elshorbany et al., 2014).

Satellite observations have the advantage of large spatial and consistent temporal coverage. Tropospheric columns of ozone (TrC-$O_3$), in Dobson unit (1 DU=$2.69\times10^{20}$ molecules $m^{-2}$),

are usually used to represent tropospheric ozone levels. The tropospheric column of a species is the species' concentration integrated from the surface to the top of the troposphere, the tropopause. The tropopause height is dynamically changing, and it varies over time, increasing or decreasing as a function of several factors, including tropospheric and stratospheric temperature (warming or cooling). Steinbrecht et al (1998) found that observed tropospheric warming of $0.7 \pm 0.3$ K per decade leads to an increase in the tropopause high and a decrease (at a rate of 16 DU/decade) in the observed column ozone levels. Similarly, after removing the variations related to major natural forcings, including volcanic eruptions, ENSO (El Niño–Southern Oscillation), and QBO (Quasi–Biennial Oscillation), Meng et al. (2021) concluded that a continuous rise of the tropopause in the Northern Hemisphere (NH) from 1980 to 2020 is evident, which they related mainly to tropospheric warming caused by anthropogenic emissions. Steinbrecht et al (1998) and Meng et al. (2021) calculate the same rate of tropopause increase for the periods 1980-2000 and 1980-2020, respectively. We investigate the trends in TrC-$O_3$ and ozone precursors at different column depths and determine their relationships.

Global models play a vital role in interpreting the observed trends in ozone precursors, verifying the consistency of emission inventories with observed precursor concentrations, and relating trends in ozone precursor emissions to ozone trends. Because satellite measurements are often sensitive to species concentrations above the surface, models provide additional information on the vertical distribution of ozone precursors needed to relate emissions or surface trends to a column or free tropospheric observations. For example, chemical transport models are used to relate Ozone Monitoring Instrument (OMI) $NO_2$ columns to surface $NO_2$ concentrations and their trends over the United States (e.g. Lamsal et al 2008, 2015; Kharol et al, 2015) since they provide vertical information on the $NO_2$ distribution. Models are also used to infer $NO_x$ emission trends from observations (e.g. Richter et al., 2005; Stavrakou et al., 2008; Miyazaki et al, 2016) or to examine whether simulations driven by state-of-the-art emissions inventories can reproduce observed changes in $NO_x$ (Itahashi et al., 2014; Godowitch et al, 2010). Models also provide insight into the role of background $NO_2$ versus local sources in relating satellite-observed $NO_2$ columns to $NO_x$ emissions changes (Silvern et al, 2019). Similarly, global models are vital for understanding trends in CO, since the lifetime of CO allows both local emissions and long-range transport and the global background to influence regional trends of CO and $O_3$. Duncan and Logan (2008) attributed the decreasing CO in the NH from 1998-1997 to decreasing European emissions and highlighted the role of Indonesian fires in driving interannual variability. Numerical models can also be used to assimilate satellite CO observations to invert for CO emission fluxes, often highlighting differences between bottom-up and top-down inventories (e.g., Kopacz et al., 2010; Fortems-Cheiney et al., 2011; Elguindi et al., 2020; Gaubert et al., 2020). For instance, several modeling studies found that the increasing emissions from China in recent years in some emission inventories were inconsistent with the negative trends observed by MOPITT (Yin et al, 2015; Strode et al., 2016; Zheng et al, 2019), while the decreases over the United States and Europe are supported by the observed decrease in CO. Jiang et al (2017) and Zheng et al (2019) also found that a decrease in biomass burning contributes to the negative CO trend in the NH. Mean calculated $O_3$ burden using CMIP6 simulation (Griffiths et al, 2021) revealed an increase of 44% from 1850 to the mean of the period of 2005-2014 and by another 17% until 2100 using the SSP370 experiments. Other sources of $NO_x$ such as lightning and soil emissions play an important role in controlling the $O_3$ budget, especially in low-$NO_x$ regions. We investigate these sources and the role they play in determining $O_3$ trends and variability on regional and global scales, as well as their determining factors.

Previous literature demonstrates the importance of controlling the emissions of ozone
precursors to effectively reduce surface $O_3$ levels. Therefore, a thorough and rigorous
understanding of the trends and variability for $O_3$ precursors is of paramount importance for a
global abatement strategy of $O_3$ levels. In this study, we use ozonesonde, remote sensing, and
global models to evaluate tropospheric $O_3$ and $O_3$ precursor trends of CO, HCHO, and $NO_2$,
on regional and global scales.
## 2.    Methodology
### 2.1. Trend Analysis
We analyze the historical trends of tropospheric ozone and its precursors CO, $NO_2$, and HCHO,
from 2005 to 2019. For trend analysis, we use two methods, the Quantile regression (QR)
method (Chang et al., 2023), and the Weighted Least Squares (WLS). For $NO_2$, CO, and HCHO
trends are calculated based on the QR method (Chang et al., 2023), as follows: (1) we first
compute the deseasonalized monthly time series of $NO_2$ and HCHO tropospheric columns
(hereafter referred to as TrC-$NO_2$, TrC-HCHO), and CO atmospheric column (TC_CO), (2) we
use the quantile regression method for computing the trend, focusing here on the median, and (3)
uncertainties at a 95% confidence level are estimated using the block bootstrapping approach,
through 1000 iterations with blocks size of $N^{0.25}$ with N the number of monthly values. They are
calculated over a 1°x1° grid and only in cells where at least 75% of the monthly values are
available. TC_CO column (see sec. 2.2.1) time series trends are also calculated as Weighted
Least Squares (WLS) of the monthly anomaly, weighted by the monthly regional standard
deviation (for comparison with the QR method). The tropospheric ozone column (TrC-$O_3$),
trends are calculated based on the WLS method. Tropospheric columns of satellite observations
are calculated based on the WMO thermal definition of the tropopause. To account for varying
tropospheric column definitions used in previous literature, we also evaluate the trends at varying
column depths.
### 2.2. Data resources
In this section, we present the different data repositories and their characteristics.
### 2.2.1.    Satellite data
A list of the applied satellite data products and their resolution is shown in Table 1. For
Tropospheric ozone data, we use the Ozone Monitoring Instrument/Microwave Limb Sounder
(OMI/MLS) product (Ziemke et al., 2006). The OMI/MLS product is the residual of the OMI total
ozone column and the MLS stratospheric ozone column, available as gridded monthly means. The
OMI/MLS tropospheric column ozone product applies all necessary data quality flags to both OMI
total ozone and MLS profile ozone; the OMI/MLS product further includes cloud filtering by
omitting all scenes with OMI reflectivity greater than 0.30. The tropospheric $NO_2$ column
retrievals used were the QA4ECV project (http://www.qa4ecv.eu/ecvs) version 1.1 level 2 (L2)
product for OMI (Boersma et al., 2017a), GOME-2 (Boersma et al., 2017b), and SCIAMACHY
(Boersma et al., 2017c). The ground pixel sizes of the OMI, GOME-2, and SCIAMACHY
retrievals are 13 km×24 km, 80 km×40 km, and 60 km×30 km, with local Equator overpass times
of 13:45, 09:30, and 10:00 LT, respectively. We also use HCHO tropospheric columns retrieved
from OMI (De Smedt et al. 2018) from the QA4ECV project. Atmospheric total column CO
daytime observations were obtained from the MOPITT instrument aboard the Terra Satellite
(Barret et al., 2003; Buchholz et al., 2017). Monthly daytime L3 data were obtained at 1º gridded
horizontal resolution from the NASA Langley Research Center Atmospheric Science Data Center
(ASDC, 2024), using version 9 (V9) retrievals, and the joint near-infrared/thermal-infrared product
(Deeter et al., 2022). Low-quality data were excluded by applying the provided quality flag.

Table 1 Satellite data products and their reference periods.

| Parameter | Resolution (Satellite pixel size) | Instrument/Platform | Reference Period | Reference |
|---|---|---|---|---|
| NO$_2$ | 1°x1° (13 km x 24 km) | OMI/Aura | 2005–2020 | Boersma et al., 2017a |
| NO$_2$ | 1°x1° (40 km x 80 km) | GOME-2/METOP-A | 2007–2018 | Boersma et al., 2017b |
| NO$_2$ | 1°x1° (30 km x 60 km) | SCIAMACHY/ENVISAT | 2005–2011 | Boersma et al., 2017c |
| CO | 1°x1° (22 km x 22 km) | MOPITT/TERRA | 2002–2020 | Deeter et al., 2022 |
| HCHO | 1°x1° (13 km x 24 km) | OMI/Aura | 2004–2020 | De Smedt et al., 2018 |
| Ozone | 1°x1° | OMI/MLS | 2004–2020 | Ziemke et al., 2006 |


### 2.2.2. Ozonesonde Data

Direct sampling of ozone throughout the atmospheric column by ozonesondes on board of high-
altitude balloons is a primary source of information of the ozone abundance and changes in the
free troposphere. Ozonesonde data have been used extensively for satellite ozone product
validations, trend analyses, and as a priori climatology profiles for satellite retrieval algorithms
(McPeters and Labow, 2012; Labow et al., 2015; Hubert et al., 2021; Christiansen et al., 2022;
Newton et al., 2016). Ozonesondes networks around the globe have been providing the ozone
community with accurate in situ measurements of high vertical resolution (100-m) for the last 5
decades in the Northern Hemisphere (Krizan and Lastovicka, 2005), nearing 3 decades at
stations in the tropics (Thompson et al., 2017), and in the last decade, new efforts are
contributing with data from undersampled regions such as the tropical Andes (Cazorla and
Herrera, 2022). Other important contributions include dedicated campaigns for regional studies
(e.g. Newton et al., 2016; Fadnavis et al., 2023). Figure 1 shows a map with ozonesonde stations
around the globe whose data are publicly available from data providers (station names,
coordinates, and links for data access in the Supplementary Material, Table S1). In this work, we
present a review of ozonesonde trends calculated and published in previous studies (Wang et al.,
2022 and Christiansen et al., 2022).

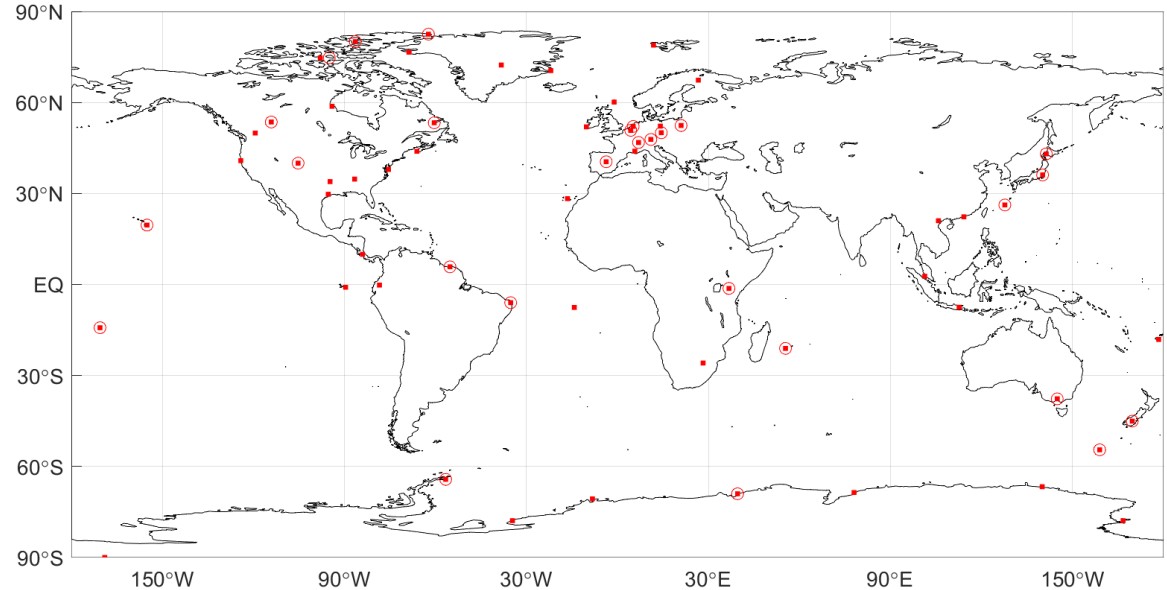

Figure 1: Ozone-sounding stations around the globe (red squares) whose data are publicly
available (Table S1). Stations that meet the criteria to calculate trends (Wang et al., 2022) are
circled in red.

### 225 2.2.3. Model simulations of ozone precursors and their vertical distribution

Model simulations provide information on the vertical distribution of trace gases that can help
interpret the observed columns. Here, we use a Goddard Earth Observing System (GEOS) Earth
System Model (Molod et al, 2015) simulation run with the GMI chemistry mechanism (Duncan
et al, 2007; Strahan et al, 2007; Nielsen et al, 2017) to simulate the contributions of the lower,
middle, and upper troposphere to the tropospheric columns of ozone and its precursors. The
model configuration is described in Fisher et al (2024) and summarized here. The MERRA-2
reanalysis (Gelaro et al., 2017) constrains the GEOS-GMI meteorology. The GEOS-GMI
meteorology is replayed to the MERRA-2 meteorology as described in Orbe et al (2017).
Anthropogenic emissions of $NO_2$, CO, and VOCs are based on the MACCity inventory (Granier
et al, 2011) through 2010 and the RCP8.5 emissions afterward, with $NO_2$ emissions scaled based
on OMI. The emissions are downscaled to higher resolution using the EDGAR 4.2 emission
inventory (Janssens-Maenhout et al., 2013). Biomass burning emissions for the analysis period
come from the Fire Energetics and Emissions Research (FEER) product (Ichoku and Ellison,
2014). Liu et al (2022) evaluated another GEOS simulation with GMI chemistry with satellite
observations of $TrC-O_3$, $TrC-NO_2$, $TrC-HCHO$, and $TC-CO$.

### 241 3. Data Analysis and Discussion

### 242 3.1. $TrC-O_3$ Sensitivity to Tropopause

Calculated $TrC-O_3$ depends on several factors such as tropospheric ozone levels, atmospheric
warming (e.g., due to GHG emissions) or cooling (stratospheric or tropospheric (e.g., after major
volcanic eruptions), and tropopause height (TH). Atmospheric warming or cooling can lead to a
decrease or an increase, respectively, of TrC-O$_3$ due to the respective change in the TH. Several
methods are used to determine the TH. The WMO thermal definition for the first TH, the lowest
altitude level at which the lapse rate decreases to 2º K km$^{-1}$ or less, provided that the average
lapse rate between this level and all higher levels within 2 km does not exceed 2º K km$^{-1}$. A
second tropopause may be also found if the lapse rate above the first tropopause exceeds 3°K
km$^{-1}$ (WMO, 1992; Hoffmann and Spang (2022). Other studies define the TH based on fixed
pressure levels (from ground to 150, 200, 300, and 400 hPa). Mean OMI/MLS TrC-O$_3$ values in
July (2005-2019) calculated based on the WMO thermal definition, are shown in Figure 2. TrC-
O$_3$ values are comparable to previously reported CMIP6 and satellite measurements (Griffiths et
al., 2021). Partial ozone columns (OC) calculated from the ground to different pressure levels,
150, 200, and 300 hPa show increasing OC values with increasing column depth, with calculated
OC at 150 and 200 hPa being the closest to the TrC-O$_3$ WMO values, still overestimating OC in
the northern hemisphere (50-90º N), especially for the 150 hPa OC, see Figure 2.

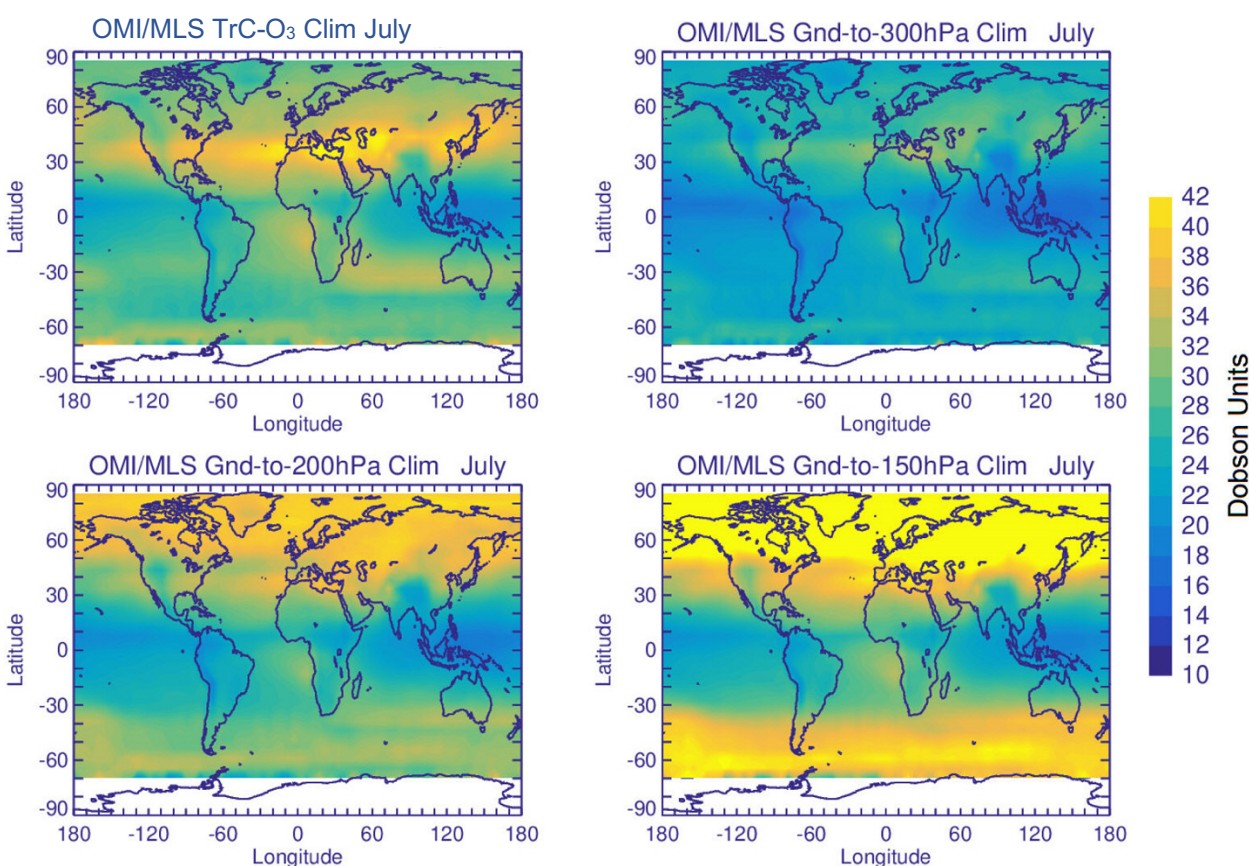

Figure 2: Global Mean (2005-2019) Column Ozone based on the WMO definition, and for
different column depths.

Steinbrecht et al (1998) found that observed tropospheric warming of 0.7± 0.3 K per decade
leads to an increase in the TH and a decrease in total ozone. They also calculated a decrease of
16 DU per kilometer increase in TH. These results indicate the importance of TH on calculated
long-term ozone trends. This could also affect comparisons between trends calculated based on
different TrC-O$_3$ definitions and near-surface ozone levels. The time series of deseasonalized TH
from 2004 to 2021 are shown in Figure 3 together with their zonal mean trends. Trends in TH are
positive reaching 60 meters/decade except in a narrow band in the tropics from 10ºS to 20ºN and
at 30ºS, where TH decreases at a rate up to 30 meters/decade. TH in the tropical regions is also
characterized by high variability (see Figure 3). These results are also consistent with recent
reports showing a positive trend of TH from 20-80ºN at a rate of 50-60 m/decade (Meng et al.,
2021). They related this increase primarily to tropospheric warming. These results show that
using a fixed pressure level for the tropopause may not be accurate given the change in TH over
time. In the following sections, tropospheric columns will be calculated based on the WMO
tropopause definition.

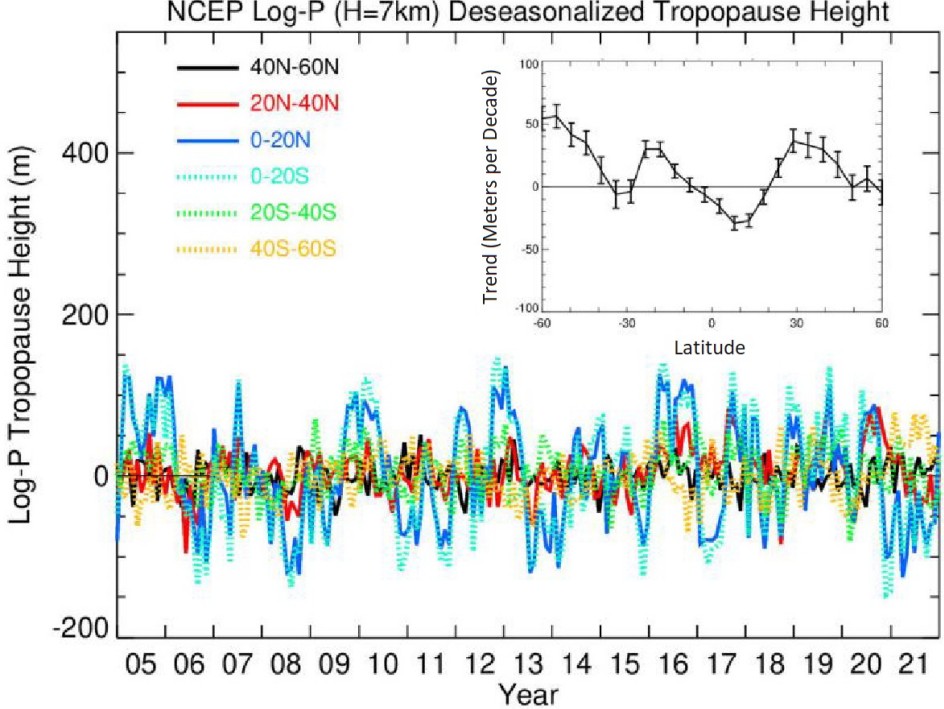


Figure 3: National Centers for Environmental Prediction (NCEP) WMO (2K/km) tropopause
log-P height time series with trends (meters/decade) embedded.

### 3.2. Spatial Distribution of $O_3$ and its Precursors

Tropospheric $O_3$ results from in-situ photochemical formation and STE. In-situ $O_3$ results from
the photolysis of $NO_2$. Therefore, the sources and fate of $NO_2$ in the atmosphere determine $O_3$
burden and distribution. $NO_2$ is formed from the reaction of hydrogen peroxyl ($HO_2$) and alkyl
peroxyl ($RO_2$) radicals with NO (R 3.2-1). While photolysis of $NO_2$ is the main source of ozone,
high $NO_2$ levels can suppress $O_3$ levels as $NO_2$ reacts with OH radical forming $HNO_3$ (R 3.2-2 to
R 3.2-4), thus reducing the oxidation rate of hydrocarbons and respectively $HO_2$ and $RO_2$ levels,
leading to a net loss of $O_3$ (e.g., Finlayson-Pitts and Pitts, 2000; Elshorbany et al., 2010,
Archibald et al., 2020). Ozone production efficiency is calculated as the ratio of the number of
$NO_2$ molecules photolyzed to form $O_3$ to that lost due to the reaction with OH forming $HNO_3$.
Under NO-sensitive conditions, the decrease in $NO_x$ leads to a reduction in OH, HCHO, and $O_3$.
However, under high NO conditions, a reduction in $NO_x$ could lead to an increase in
photochemical products, OH, HCHO, and $O_3$ because a reduction in $NO_2$ leads to a decrease in
OH loss rate, thus higher $HO_2$ and $RO_2$ production (Elshorbany et al., 2012; Archibald et al.,
297 2020).

R 3.2-1      $HO_2/RO_2$   +   NO                        →   $NO_2$
R 3.2-2      $NO_2$        +   hν (hν < 424 nm)          →   $O(^3P)$  +   NO
R 3.2-3      $O(^3P)$      +   $O_2$   +   M             →   $O_3$    +   M
R 3.2-4      OH            +   $NO_2$ (M)               →   $HNO_3$ (M)

The observed mean tropospheric columns of $O_3$, $NO_2$, and HCHO and atmospheric column of
CO from 2005 to 2019 are shown in Figure 4. The unit for column number density is
Pmolec/cm$^2$ ($\times 10^{15}$ molecules per square centimeter), except for TrC-$O_3$, which is Dobson. $NO_2$
concentration has decreased since 2005 in North America, Europe, and Australia, mainly due to
strict measures to reduce air pollution (Lamsal et al., 2015). Since $O_3$ is a photochemical product
that is formed based on non-linear chemistry, a reduction in $NO_2$ may lead to an increase or
decrease in tropospheric $O_3$ levels based on the dominant photochemical regime in the respective
region. In addition, tropospheric ozone levels may be affected by STE especially in the middle
and upper troposphere (Li et al., 2024), as well as LRT, especially in the free troposphere (e.g.,
Glotfelty et al, 2014; Itahashi et al., 2020). The highest values of the $NO_2$ tropospheric column
are in the northern hemisphere between 10 ºN and 50ºN, especially over the eastern US, northern
Europe, and east and south Asia, with elevated levels in the Southern Hemisphere (SH) between
10 and 30ºS, especially in sub-Saharan Africa, and Brazil. TrC-$O_3$ is also highest over the band
of 20-50º N, especially over the eastern coast of the US, southern Europe, and east Asia. Some
differences exist between TrC-$O_3$ and TrC-$NO_2$ spatial patterns which is due to factors including
different lifetime, photochemical sensitivity (see sec. 3.4), and STE. On average, the northern
hemisphere has higher TC-CO than the southern hemisphere due to a larger number of sources
(Buchholz et al., 2021). Additionally, high amounts of CO are found in regions with large
anthropogenic sources (e.g., eastern China) or in regions with large and regular fire seasons (e.g.,
central Africa) (Buchholz et al., 2021). HCHO and CO show a similar spatial pattern over
western Africa due to emissions from biomass burning (Marais et al., 2012, Buchholz et al.,
2021). In the following sections, global and regional trends of TrC-$O_3$ are investigated along
with tropospheric ozone precursors.

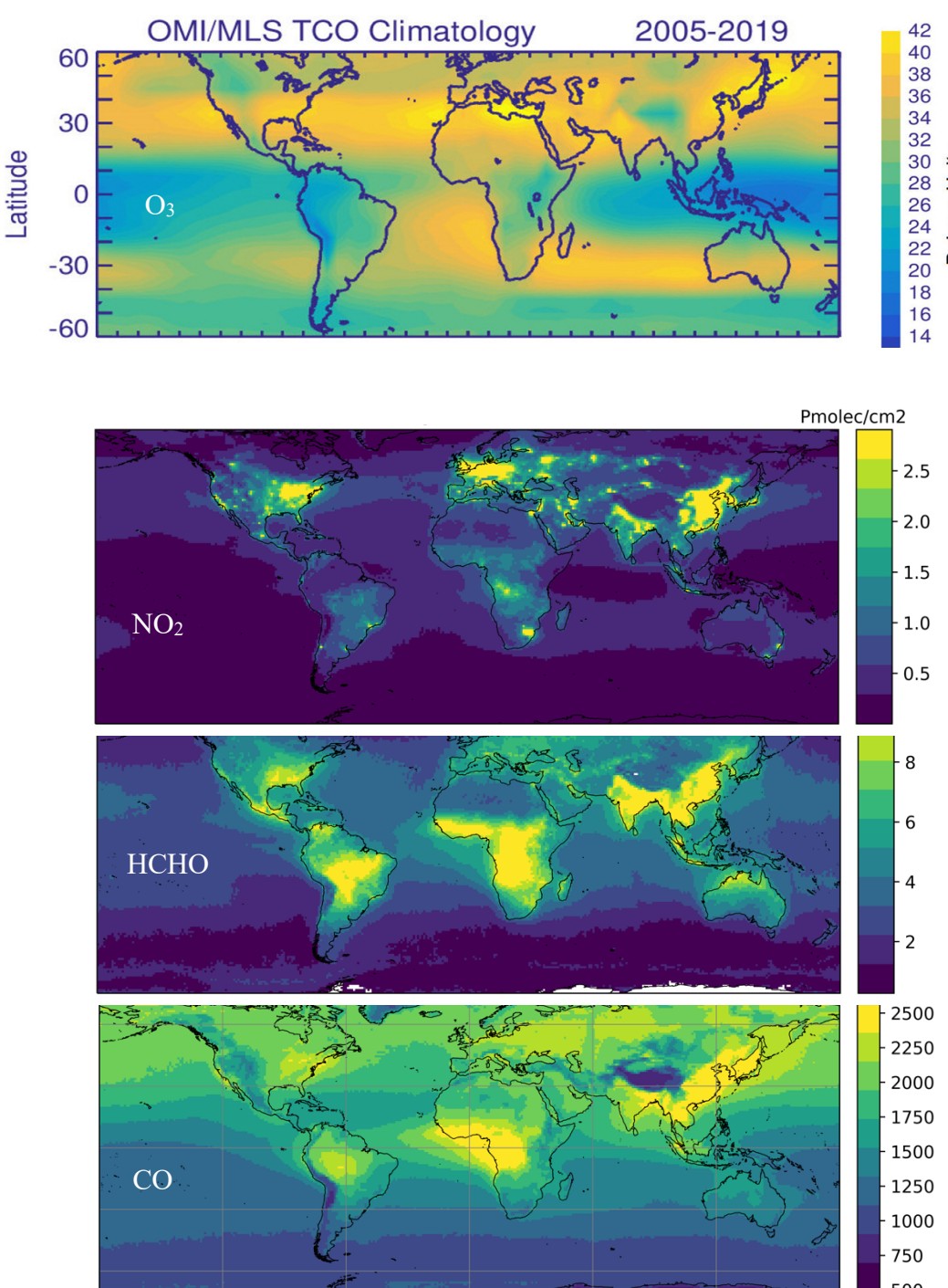

Figure 4: Mean (2005-2019) of TrC-$O_3$, TrC-$NO_2$, TrC-HCHO, and TC-CO.

### 3.3. Simulated $O_3$ Precursors

Ozone and its precursors differ in their vertical distribution through the troposphere. In this section, we use the GEOS -simulations to show how the lower, middle, and upper troposphere contribute to the simulated columns of $O_3$ and its precursors to complement the column information from satellites. Figure 5 shows the simulated mean (2005-2019) contributions to tropospheric columns of $O_3$, $NO_2$, formaldehyde, and CO, partitioned into the lower (up to 700hPa), middle (700-400hPa), and upper (400hPa to tropopause) portions of the troposphere for the tropical band (30ºS:30ºN) and the global mean. The middle and upper troposphere make

large contributions to the simulated TrC-O₃ and its variability (Figure 5). The lower troposphere
makes the largest contribution to the TrC-HCHO since it is mainly a photochemical product
(e.g., Elshorbany et al., 2009), and all three levels make substantial contributions to the CO
column. Globally, the relative contributions for TrC-O₃, TrC-HCHO and CO are similar to those
of the tropics. However, for TrC-NO₂ the lower troposphere makes a smaller contribution in the
tropics than globally.

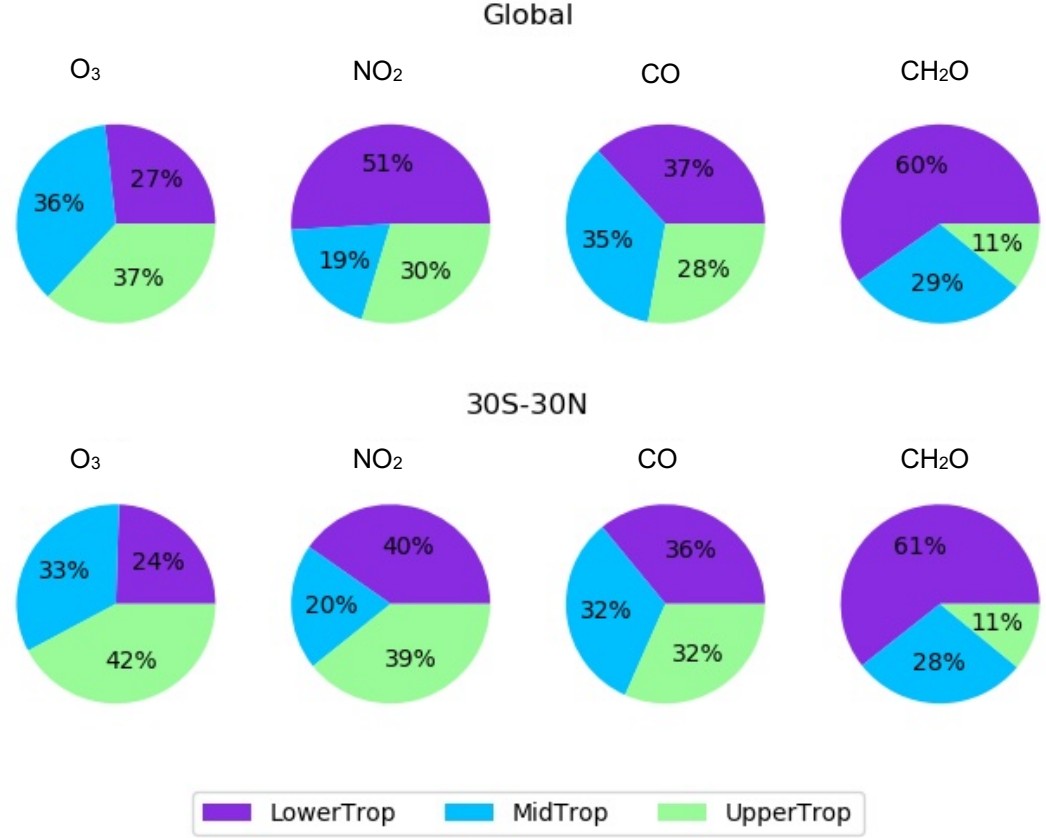

Figure 5: Simulated average (2005-2019) contributions to the tropospheric columns of O₃, NO₂,
formaldehyde, and CO from the lower (surface-700hPa), middle (700-400hPa), and upper
troposphere (400hPa-tropopause) using NASA GEOS-GMI. The top row is for the global mean,
while the bottom row is averaged from 30°S-30°N.
**3.4. Tropospheric Trends**
**3.4.1.       Global Tropospheric Ozone**
Global TrC-O₃ trends calculated for different column depths are shown in Figure 6. Compared to
TrC-O₃, OC trends up to 150 hPa seem to be the closest despite OC values being much higher
than that of the TrC-O₃ (Figure 2). All trends with high confidence, HC (at 95% confidence) are
positive indicating increasing trends of ozone columns, regardless of the tropopause height. Low
confidence, LC (at 2 σ levels) decreasing TrC-O₃ trends were also found in some locations, e.g.,
South Australia, South Africa, and the northeastern coast of the US. Increasing trends in the
northern midlatitudes may also be partially related to STE (Willimas et al, 2019; Li et al., 2024).
While the annual trends inform about overall trends, seasonal trends provide insights into local
chemistry and meteorology. For example, during the boreal summer months, June, July, and
August (JJA), TrC-O$_3$ HC trends are similar to the annual trends except for HC decreasing trends
over South America and South Africa and HC increasing trends over the west and central Africa
and Central America (Figure S7). During the boreal winter months, HC trends are also similar to
the annual trends (Figure 6) except for HC increasing trends over Europe, North America, South
America, and South Africa (Figure S7).

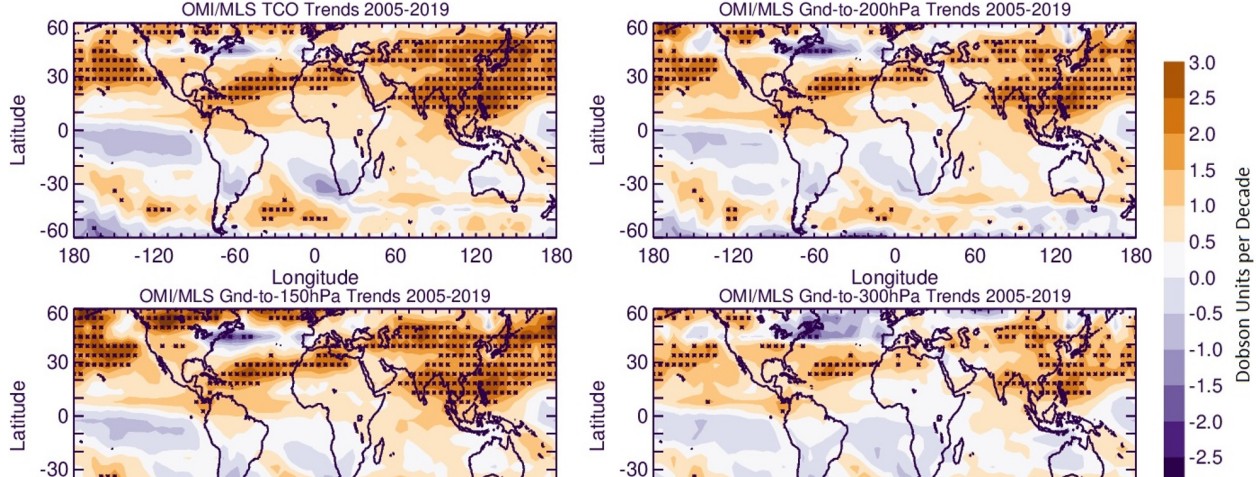

Figure 6: Trends in tropospheric column ozone, based on the WMO thermal definition, and the
trends on ozone columns (from ground to 150, 200, and 300 hPa). Trends are calculated based on
deseasonalized monthly data from 2005 to 2019. Asterisks denote 95% confidence trends.

The time series of OMI/MLS TrC-O$_3$ averaged over several latitudinal bands and at different
column depths are shown in Figure 7. Zonal mean TrC-O$_3$ compares well with partial ozone
columns in the tropics (from 30ºS to 30ºN) with the OC of up to 300 hPa differing by about 10
DU from the TrC-O$_3$ (Figure 7b). The lowest TrC-O$_3$ trends are located in the northern
hemisphere (30 – 60ºN) at 0.78±1.16 DU/decade, followed by the southern hemisphere (30-60ºS
(0.95±0.75 DU/decade) and the tropical band (30-30ºN (1.06±0.40 DU/decade). In addition, the
continental trends over Australia, South Africa, and South America in the 30 ºS -60ºS band are
essentially negative and the positive trends in this band are contributed mainly by oceanic
regions (see Figure 6). The positive trends in the 30ºN -60ºN band are slightly offset by the
negative trends over the northeastern US and western Europe (see Figure 6).

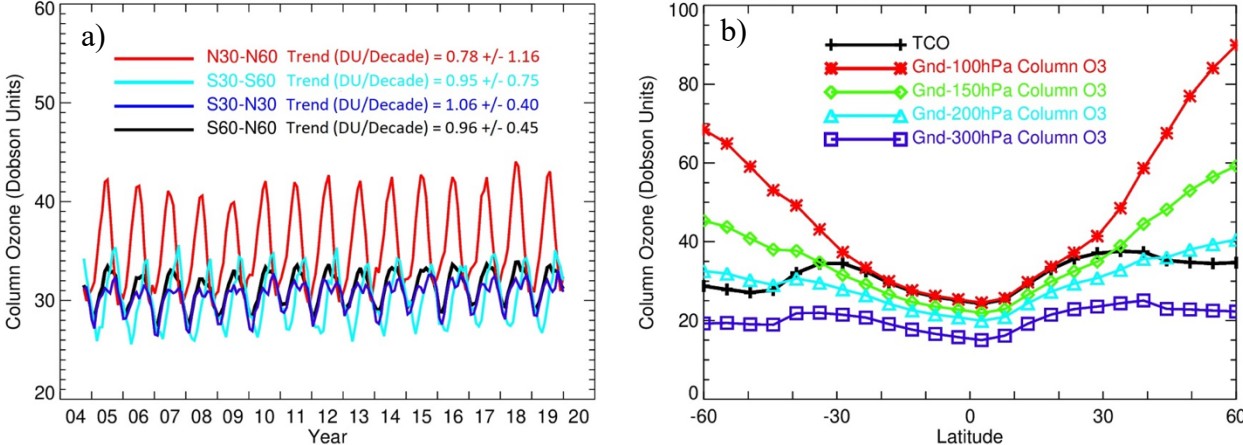


Figure 7: Time series and zonal mean trends of OMI/MLS TrC-O₃ in different latitudinal bands
(left) and zonal mean of different column depths (right) from 2005-2019.

385       Observed trends for the time period before COVID-19 (2005-2019) show that OC trends
were highest in the northern latitudes (0-30º N) reaching about 1.5 DU/decade, followed by the
northern midlatitudes 30-60ºN (Figure 8). The high trends in the 30-60ºN band are dominated by
transpacific impacts as well as some impacts from East Asia. The positive trends in the southern
hemisphere (0-30º S) are mainly over Amazonia and Southeast Asia, being offset by small
negative trends over Western Australia and South Africa. The trends during the time period
(2005-2021) show a decline in $O_3$ column trends in the northern hemisphere but a slightly
increasing trend in the southern hemisphere (Figure 8b). The decreasing trends in the northern
hemisphere during the COVID-19 is consistent with previous literature showing a decrease in
several pollutants including $NO_2$ and $O_3$ due to the extended lockdown periods imposed during
the pandemic (e.g., Bauwens et al., 2020; Elshorbany et al., 2021; Steinbrecht et al., 2021; Putero
et al., 2023). The decrease of $NO_2$ in some parts of Europe and the northeastern USA led to a
decrease in tropospheric $O_3$.

398       Zonal mean trends (Figure 8) show that OC up to 150 hPa is almost identical to that of
TrC-$O_3$ except for the high latitudes 45º-60º S and 45º-60º N. The decreasing trends above 30ºN
and 30ºS are due to the offsetting impact of negative trends over the northeastern US and western
Europe in the north, and Australia and South Africa in the south, respectively. This impact is less
apparent in the 150 hPa OC due to the lower positive trends in that band compared to TrC-$O_3$.
The 200 hPa OC comes next with a very good agreement from 60º S to 10º N. followed by the
100 hPa which is only in good agreement from 30º S to 30ºN, while the 300 hPa OC was the
farthest from the TrC-$O_3$. The decrease of $O_3$ in the northeastern US and western Europe is
consistent with decreasing $NO_2$ trends and NO-sensitive conditions dominating these regions.
The decreasing trends of $NO_2$ (see below) are due to the successful measures applied since 2004
to mitigate air pollution in these regions. The increase of $O_3$ in the western US maybe due to
LRT from eastern Asia (e.g., Itahashi et al., 2020).

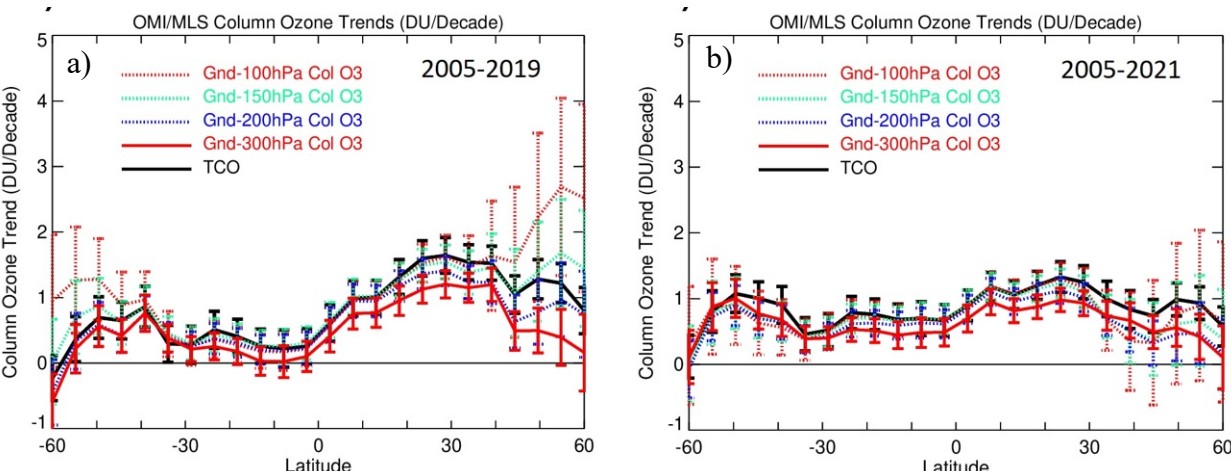


Figure 8: Tropospheric column ozone (TrC-O₃) and trends for different column depths before the
COVID-19 pandemic (2005-2019) and including the pandemic (2005-2021).

### 3.4.2.    Free tropospheric trends


Trends of ozone in the free troposphere presented here are based on previous work published in
the literature. Despite the high stability of ozonesonde measurements across the global networks
over several decades (Stauffer et al., 2022), the spatial sparsity of sounding stations and non-
uniform sampling frequency among sites is a limitation in using these data to produce trends.
These shortcomings have constrained the ability to include data from many stations in previously
published analyses. For example, Chang et. al (2020) estimated that at least 18 profiles per
month are needed at a single station to calculate accurate long-term trends, while uncertainty
increases at lower sampling rates (Chang et al 2024). However, such high sampling frequency is
only achieved at three European stations (Hohenpeissenberg, Germany; Payerne, Switzerland,
and Uccle, Belgium), while the rest of the global stations work at lower sampling rates.
Nonetheless, high-quality ozonesonde observations continue to be the gold standard against
which satellite measurements are validated. Likewise, ozonesonde data continue to provide
spaceborne observations with climatological feedback. Thus, recent studies have softened the
sampling frequency criteria in order to take advantage of the valuable data set collected by the
global ozonesonde networks. For example, the latest trend studies establish the minimum
frequency requirement to calculate trends to at least three profiles per month (Wang et al., 2022;
Christiansen et al., 2022) with at least eight months of sampling in a year, and at least 15 annual
means for an analysis of about two decades (Wang et al., 2022). With these criteria, recent
ozonesonde trend analyses indicate that ozone concentration increased globally by 1.8+/-1.3
ppbv/decade in the free troposphere within 800 to 400 hPa (Christiansen et al., 2022). However,
there is high regional variability, as illustrated in Figure 9 where ozone trends published by
Wang et. al. (2022) (1995-2017 data between 950-250 hPa) are organized by regions and
stations. For example, ozone in East Asia (Japan) has been increasing at a rate of 3.5 to 5
ppbv/decade, particularly since 2010 (Christiansen et al., 2022), which may lead to transpacific
LRT of O₃ to the western US (e.g., Itahashi et al., 2020). Over the Southwestern Indian Ocean
(La Réunion), trends are of similar magnitude (>4.5 ppbv/decade). In tropical South America,
over the Atlantic basin region (Paramaribo and Natal), sounding measurements also show ozone
increases by almost 3 ppbv/decade (Natal), but other regions in South America continue to lack
sufficient measurements to produce trends. At tropical stations in Africa (Nairobi) and the
Pacific Ocean (Hilo and American Samoa) trends are also positive, although of lower
magnitudes (0.83-1.7 ppbv/decade). In contrast, polar stations both at the Arctic and Antarctica
as well as the Southern Ocean show overall decreasing ozone concentrations to low-confidence
trends. Exceptions are the Eureka station in Canada and Lauder station in New Zealand, which
both show slight ozone increases (less than 0.5 ppbv/decade). The direction of regional trends by
Wang et. al. (2022) is consistent with regional trends presented in similar independent research
(Christiansen et al., 2022). As atmospheric composition continues to become modified under the
current regime of climate change, building consistent and longer time series of ozonesonde
measurements at other regions will continue to be an important source of firsthand information to
assess tropospheric ozone changes and trends.

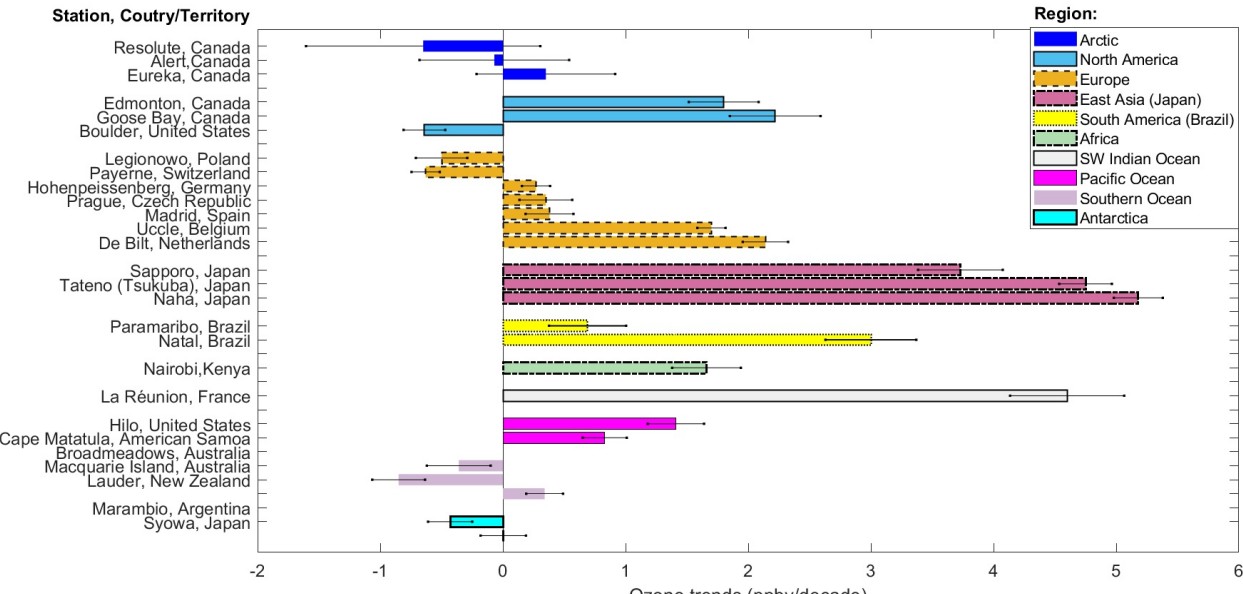


Figure 9: Ozone trends in the free troposphere from ozonesonde measurements calculated by
Wang et. Al. (2022) and organized by region and station. Data covers the 1995-2017 period
within 950 to 250 hPa. Error bars show 1-σ uncertainty. The coordinates of ozonesonde stations
are listed in Table S1.

### 3.4.3. Regional Ozone Trends

As shown in Figure 10, the highest OMI/MLS regional trend is observed over East Asia
(2.16±1.27 DU/decade) while the lowest trend is calculated over Eastern USA (0.63±1.72)
followed by Western Europe (0.89±1.60) and Australia (1.05±1.44) DU/decade. We next
calculate the monthly trends from the GEOS-GMI simulation to investigate how the simulated
trends vary through the tropospheric column.


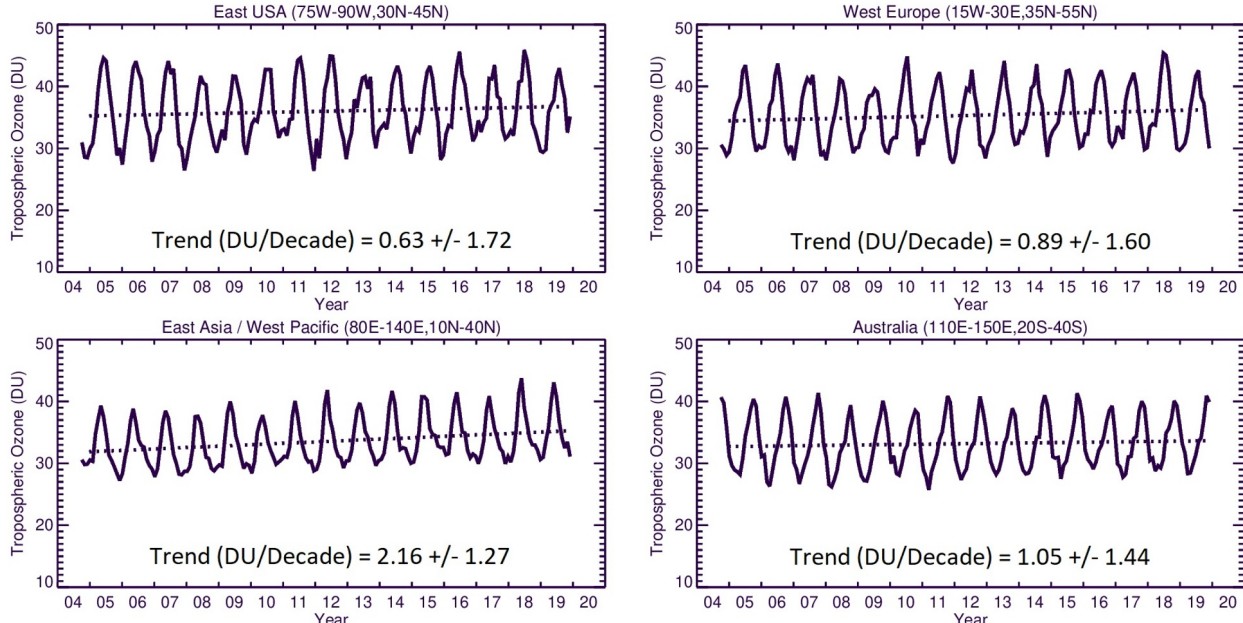


Figure 10: OMI/MLS observed regional mean trends of TrC-O$_3$.

The simulated trends in partial columns (lower, middle, and upper troposphere), as well as the
TrC-O$_3$, TrC-NO$_2$, TrC-HCHO, and TC-CO from 2005 to 2019, are shown in Figure 11. The
simulated tropospheric columns of TrC-O$_3$ and TrC-HCHO show a positive trend in most
regions (Figure 11), consistent with the results of Liu et al (2022) using a different GEOSCCM
simulation. Liu et al (2022) highlighted the importance of formaldehyde trends for analyzing the
simulated trends in tropospheric ozone. Considering different latitude bands, the highest trends
are simulated between 30º S and 60º N, consistent with calculated trends based on satellite
observations (see sec. 3.4). In contrast, the simulated NO$_2$ and CO trends are mostly negative,
although positive trends are simulated over East Asia. The largest NO$_2$ negative trends are in the
northern hemisphere between 30ºN and 60ºN. The decrease in NO$_2$ trends is consistent with the
successful measures to curb emissions of pollution criteria in the US and Europe. The increased
trends in TrC-O$_3$ but decreased trends in TrC-NO$_2$, and TC-CO might indicate STE contribution
(Trickl et al., 2020; Li et al., 2024) in addition to the local chemistry.


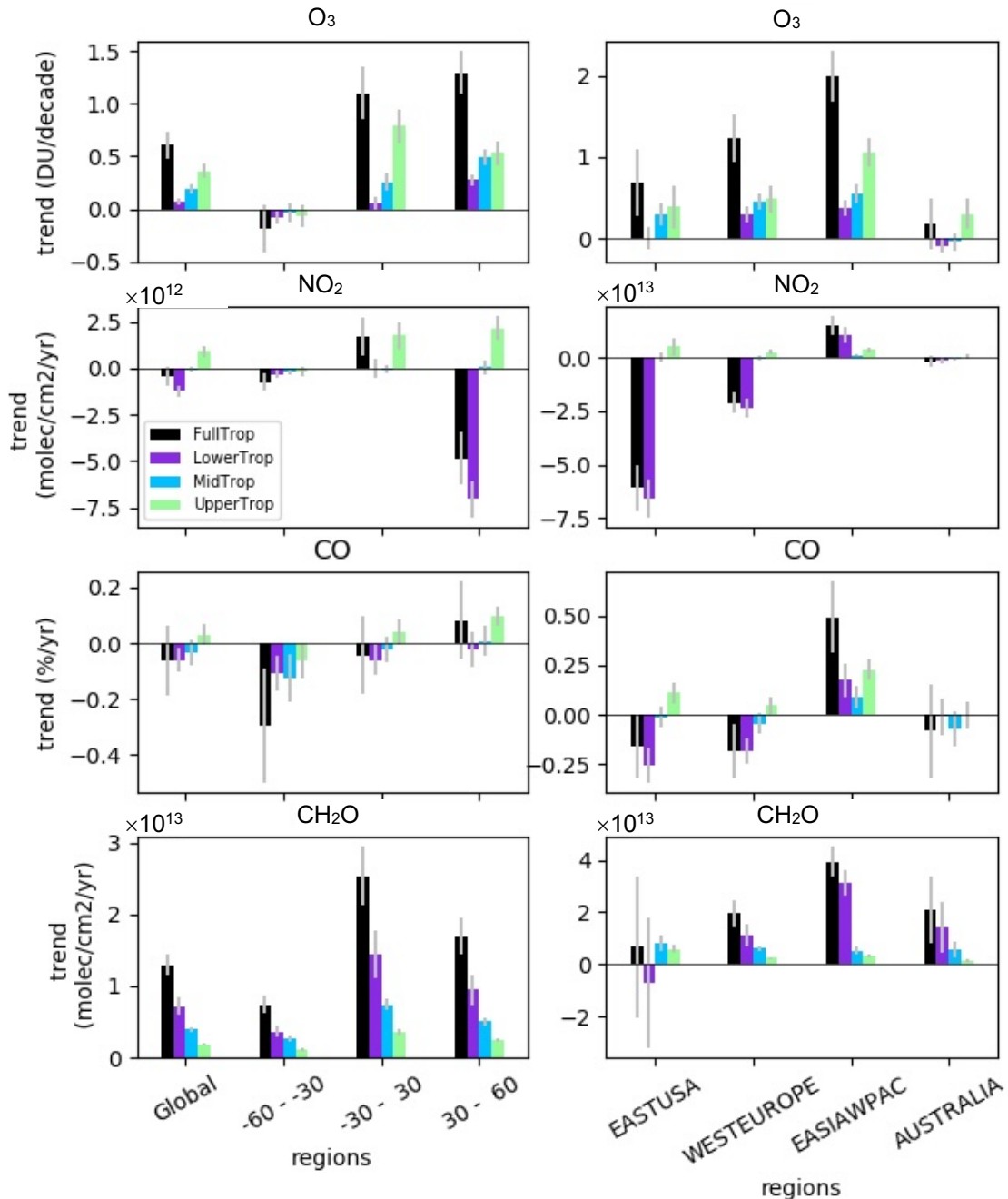


Figure 11: Global and regional trends in $O_3$, $NO_2$, CO, and HCHO calculated from the GEOS-
GMI simulation for the tropospheric column (black), lower troposphere (purple), middle
troposphere (blue), and upper troposphere (green) from 2005 to 2019. The lower, middle, and
upper troposphere are defined as in Figure 5.

492

The GEOS-GMI simulation provides an estimate of the relative contribution from different
portions of the tropospheric column to the column trends and shows that this contribution varies
by region and constituent. The middle and upper troposphere make the largest contributions to
the simulated TrC-$O_3$ trend globally, with large contributions from the upper troposphere driving
the simulated TrC-$O_3$ trend at 30°S-30°N (Figure 11). The middle and upper troposphere

contribute most of the simulated positive TrC-O$_3$ trend over the eastern USA, while all three
levels contribute over western Europe and East Asia. The upper troposphere makes the primary
contribution to the simulated trend over Australia. Simulated TrC-O$_3$ trends are also quite
comparable to those observed by OMI/MLS within the measurement model uncertainty (see
Figure 10 and Figure 7). Over Australia, the OMI/MLS trend of 1.05±1.44 DU/decade is higher
than the model trend of about 0.18±0.308 DU/decade (see Figure 11). However, since OMI/MLS
trend has a calculated uncertainty (2σ) of 1.44 DU/decade, both the model and OMI/MLS for
Australia are not statistically different.
While the upper troposphere is a major driver of the simulated TrC-O$_3$ trends, the lower
troposphere is the largest contributor to the simulated trends in the tropospheric NO$_2$, CO, and
HCHO globally and over many regions (Figure 11). Exceptions include the simulated NO$_2$ in the
tropics (30°S-30°N), which is dominated by the upper troposphere, the simulated HCHO column
over the eastern USA, which is driven by the middle and upper troposphere; an important role
for upper tropospheric CO over East Asia; and the CO trend over Australia driven by the middle
tropospheric contribution. Figure 11 also shows that in some regions, such as the eastern USA
for all 3 precursors, the upper and lower tropospheric trends counteract each other, reducing the
magnitude of the column trend. In the following sections, we investigate trends and variability in
O$_3$ precursors, NO$_2$, CO, and HCHO.

### 3.4.4.  NO$_2$ Trends

The TrC-NO$_2$ trends over 2005-2019 are shown in Figure 12 with a regional summary in Figure
13. On a global scale, there is a strong spatial variability of the TrC-NO$_2$ trends. About a third of
the oceans show HC increase of TrC-NO$_2$ trends (at 95% confidence level), especially at mid-
latitude, with trends up to +0.01 Pmolec/cm$^2$/yr while only a few cells in the equatorial Pacific
show an HC decrease.

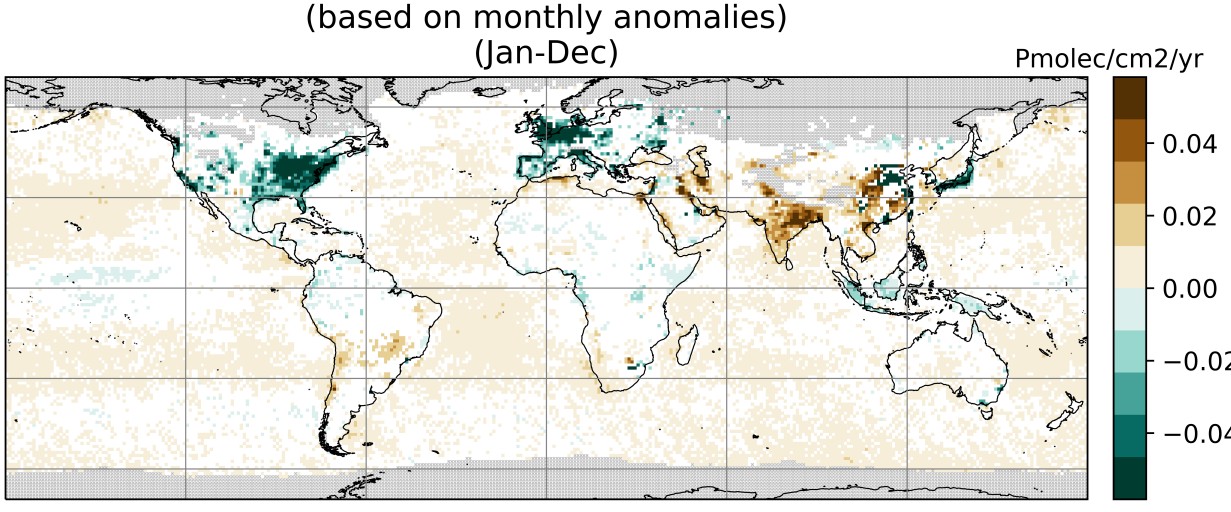

Figure 12: Global trends of OMI NO$_2$ tropospheric column (TrC-NO$_2$) over 2005-2019 (see text
for details on the calculation of the trends). Grey areas correspond to areas without enough data,
white areas correspond to regions where the trends remain at low confidence (at a 95%
confidence level).

528       Regional trends are shown in Figure 13. For high-confidence trends in a given region, the
numbers correspond to the percentiles 5/50/95 of trends among the different cells of the region
where trends are considered high-confidence. Each region is tagged with a circle whose size is
proportional to the p50 of the high-confidence trends (red for positive and green for negative),
which allows us to quickly see regions where the trend is strong. For instance, for Eastern Asia
(this region includes 1442 1°x1° grid cells) about 15% of the grid cells (about 216 grid cells) in
this region show a high-confidence decrease in TrC-$NO_2$. Over these specific 216 cells with a
high-confidence decrease of TrC-$NO_2$, the 5th and 95th percentile of the trend is -0.34 and -0.01,
respectively, Pmolec/$cm^2$/yr. About 28% of the grid cells in this region show a high-confidence
increase of TrC-$NO_2$ (which means about 403 grid cells). Over these specific 403 cells with a
high-confidence increase of TrC-$NO_2$, the 5th (resp 95th) percentile of the trend is +0.01 (resp
0.05) Pmolec/$cm^2$/yr. Therefore, the Eastern Asia region shows sub-regions with high-
confidence decreasing TrC-$NO_2$, others with high-confidence increasing TrC-$NO_2$, and the rest
with low-confidence (positive and negative) trends. This figure allows us to quickly understand
the distribution of the trends within a given region while the overall regional trend is given by
the 50$^{th}$ percentile and the circles tagging each region. It's a regional summary of what is shown
in the trend global map. In Eastern Asia, the area where trends are with high-confidence positive
is more extended than for the high-confidence decrease (28% versus 15%), but the trend values
tend to be smaller (at least when comparing the 50$^{th}$ percentiles, -0.05 versus +0.01
Pmolec/$cm^2$/yr). The map of regions is included in the supplement. Canada is included in
northern America but as shown in the trend map, most of Canada does not have OMI data
Over continental areas, high-confidence positive and negative trends are found in about
15-20% of the grid cells each (Figure 12). Regions with predominantly decreasing TrC-$NO_2$
include western and southern Europe (where about 50-60% of cells with a high-confidence
decrease), northern America (40% of cells with a high-confidence decrease, mostly located in the
eastern United States), Japan, and Indonesia. In absolute terms, these negative trends reach
values of about -0.03 Pmolec/$cm^2$/yr. Specific eastern regions of China also show similar high-
confidence TrC-$NO_2$ decreases but overall, a larger part of the country faces increasing trends up
to +0.03 Pmolec/$cm^2$/yr. Similar positive trends are observed over most of India, as well as in
specific parts of south-eastern Asia (mainly Vietnam) and the Middle East (mainly Iran and
Iraq). Conversely, TrC-$NO_2$ trends in Africa and South America remain mainly low-confidence ,
except in a few specific regions with high-confidence increases (e.g. South Africa, Morocco,
Chile, and parts of Brazil).
The trends in $NO_2$ have varying effects on the tropospheric ozone column, which is
related to the different local chemistry in each region. The concomitant decrease in TrC-$O_3$ and
TrC-$NO_2$ trends over some parts of the eastern US, and western Europe is consistent with the
strict $NO_x$ control measures that were applied over the last two decades. STE can also contribute
to increased TrC-$O_3$ trends, especially in the mid-latitudes. A decreasing trend of TrC-$NO_2$ but
an increasing trend of TrC-$O_3$ is present in some other regions such as in the central US, which
might be due to local chemistry and STE.

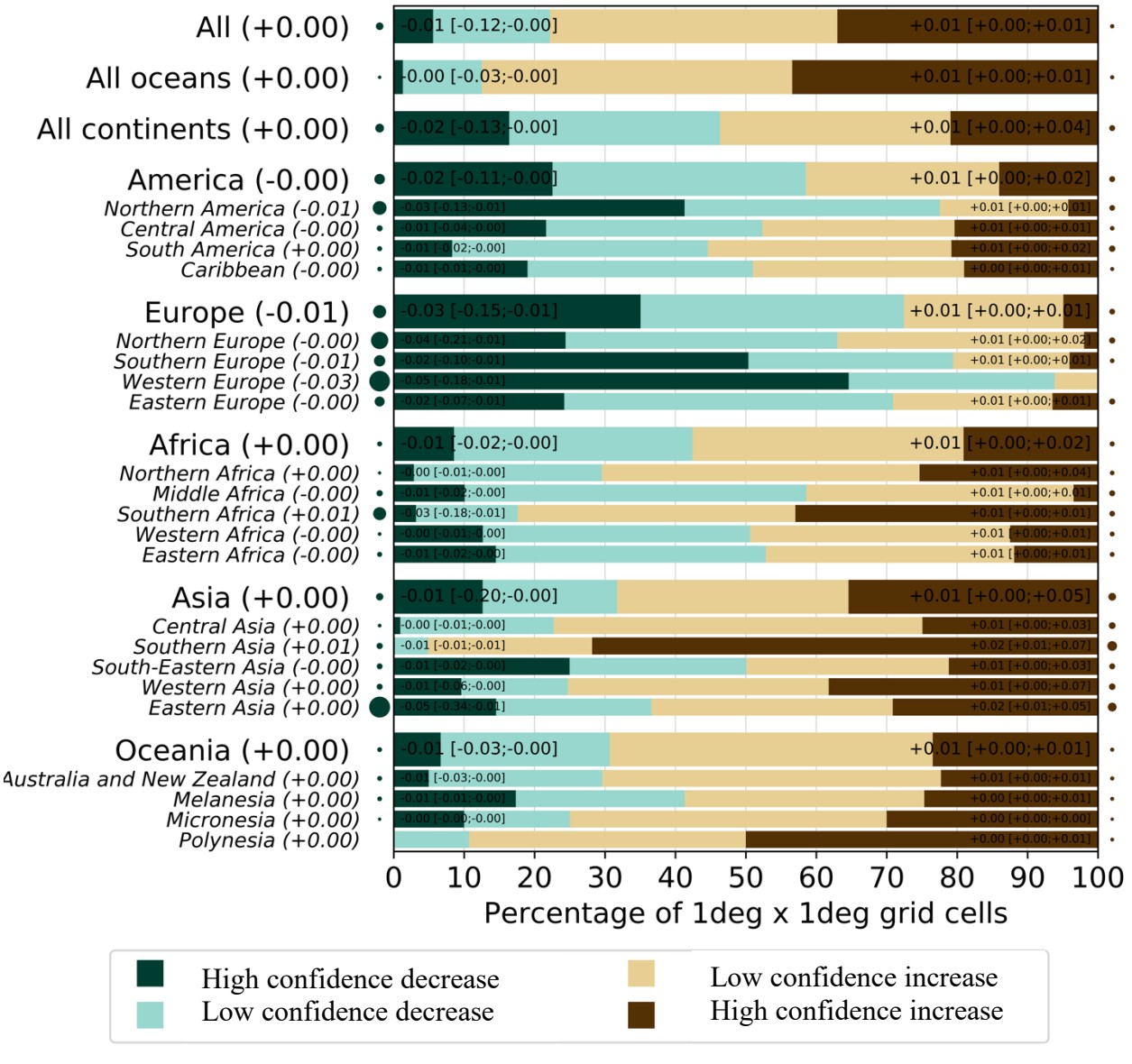

Figure 13: Summary of the high-and low-confidence regional trends of OMI $NO_2$ tropospheric column (TrC-$NO_2$) trends over 2005-2019, at a 95% confidence level (see text for details on the calculation of the trends). For each region, the trend on the bars is in the format: p50 [p5; p95], which represents the $50^{th}$[$5^{th}$, and $95^{th}$] percentiles of the trends.

Figure 14 shows the time series of regional mean tropospheric $NO_2$ concentrations from three satellite instruments, OMI for 2005-2020, GOME-2 for 2007-2018, and SCIAMACHY for 2005-2012. All the instruments exhibit common large seasonal and year-to-year variations over both industrial regions and biomass-burning areas. Slight systematic differences among the instruments can mainly be attributed to the different overpass times. The satellite observations show positive trends over China by 2010, followed by a continued decrease. Over the USA and Europe, all the retrievals show a downward trend over the analysis period. Over the US, the observed TrC-$NO_2$ levels decreased rapidly during 2005–2009 and subsequently show weaker reductions, as discussed by Jiang et al. (2018). A similar slowdown trend is found in Europe. Over India, the

OMI observations show positive trends over the 14 years (+1.6 % yr$^{-1}$). The seasonal and year-to-year variations over Southeast Asia and northern and central Africa are associated with changes in biomass-burning activity (Ghude et al., 2009).

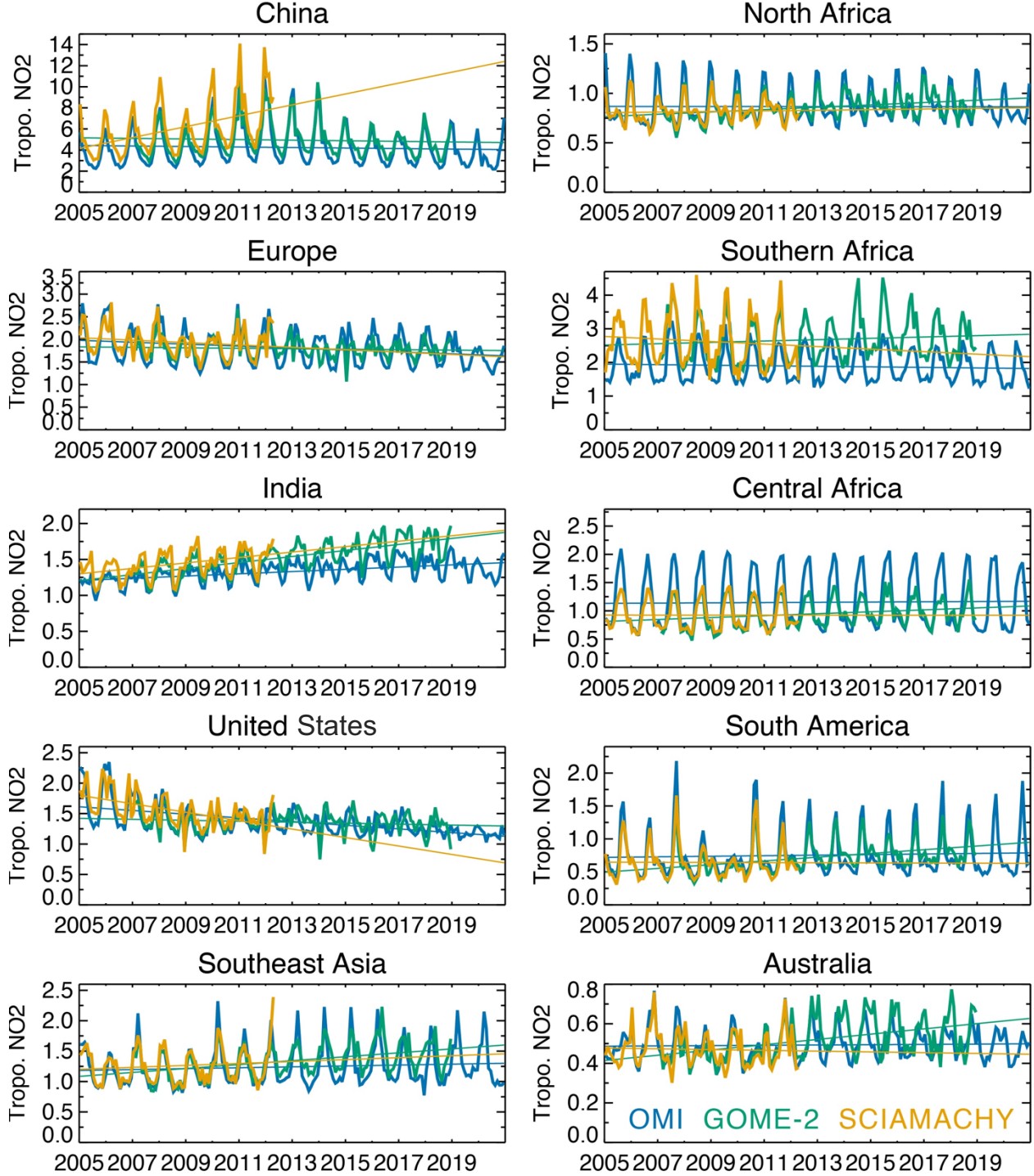

Figure 14: Time series of regional monthly mean tropospheric NO$_2$ columns (in $10^{15}$ molecules cm$^{-2}$) averaged over China (110–123° E, 30–40° N), Europe (10° W–30° E, 35–60° N), the US (70–125° W, 28–50° N), India (68–89° E, 8–33° N), South America (50–70° W, 20° S–Equator), northern Africa (20° W–40° E, Equator–20° N), central Africa (10–40° E, Equator–20° S),

southern Africa (25–34° E, 22–31° S), southeastern Asia (96–105° E, 10–20° N), and Australia
(113–155° E, 11–44° S) obtained from OMI (black), GOME-2 (blue), and SCIAMACHY (red).

### 3.4.5.    CO Trends

CO trends are calculated based on MOPITT v9 products, see sec. 2.2.1. Observed CO trends
below show a slowing in the trend compared to a previous analysis (Buchholz et al. (2021). In
the northern hemisphere, CO trends are largely negative over the US and Europe, which is
consistent with improvements in combustion efficiency and policies implemented to reduce air
pollution since 2004. Except for small sporadic positive trends, no HC trends can be calculated
over Central Asia (India and China), while there is a strong negative trend in East China due to
the recent strong focus on air quality improvement, and no HC trend in the SH.

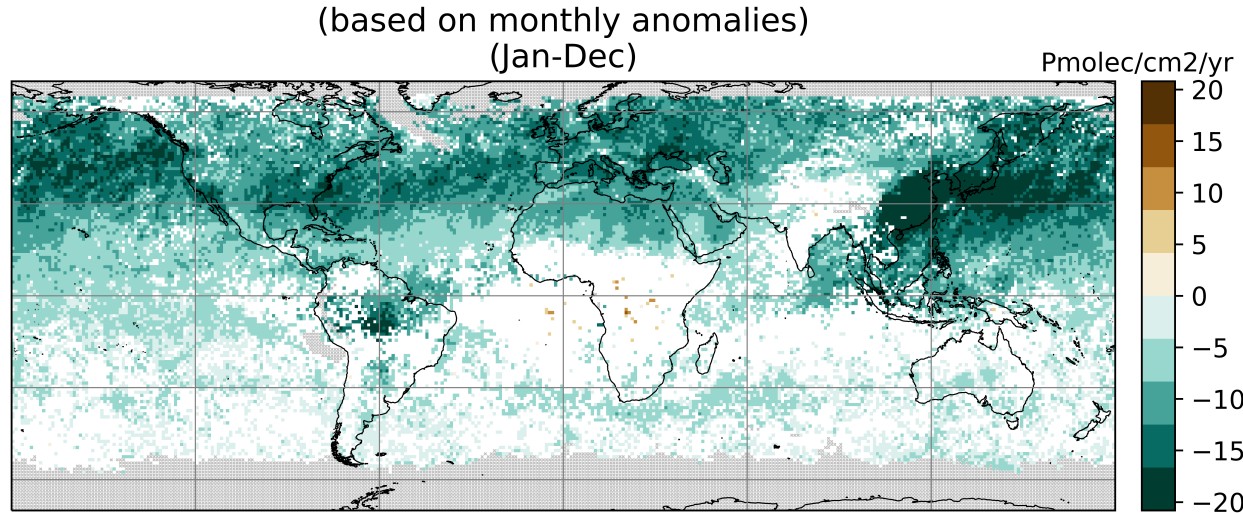

Figure 15: Trends in TC-CO from MOPITT V9J data, 2005-2019 (see text for details on the
calculation of the trends). Grey areas correspond to areas without enough data, white areas
correspond to regions where the trends remain statistically low-confidence at a 95% confidence
level.

A regional summary of the trends in the global map is shown in Figure 16. CO trends are
predominantly negative everywhere except for some sporadic positive trends over middle Africa.
Decreasing TC-CO trends are highest in Europe, followed by Asia and America with about 86%,
75%, and 69% of their cells being negative, respectively. The 50 percentiles of the trends in
these cells are -12.01, -10.21, and -10.16 Pmolec/cm$^2$/yr, respectively. Africa shows the lowest
decreasing trends as the negative trends in North Africa are being offset by small increasing
trends in middle Africa. Overall, about 41% of the cells in Africa show decreasing trends, and
50% of the trends in these cells account for -8.71 Pmolec/cm$^2$/yr. Thus, even though the NH
accounts for most of CO emissions, decreasing trends of TC-CO are evident in these regions.


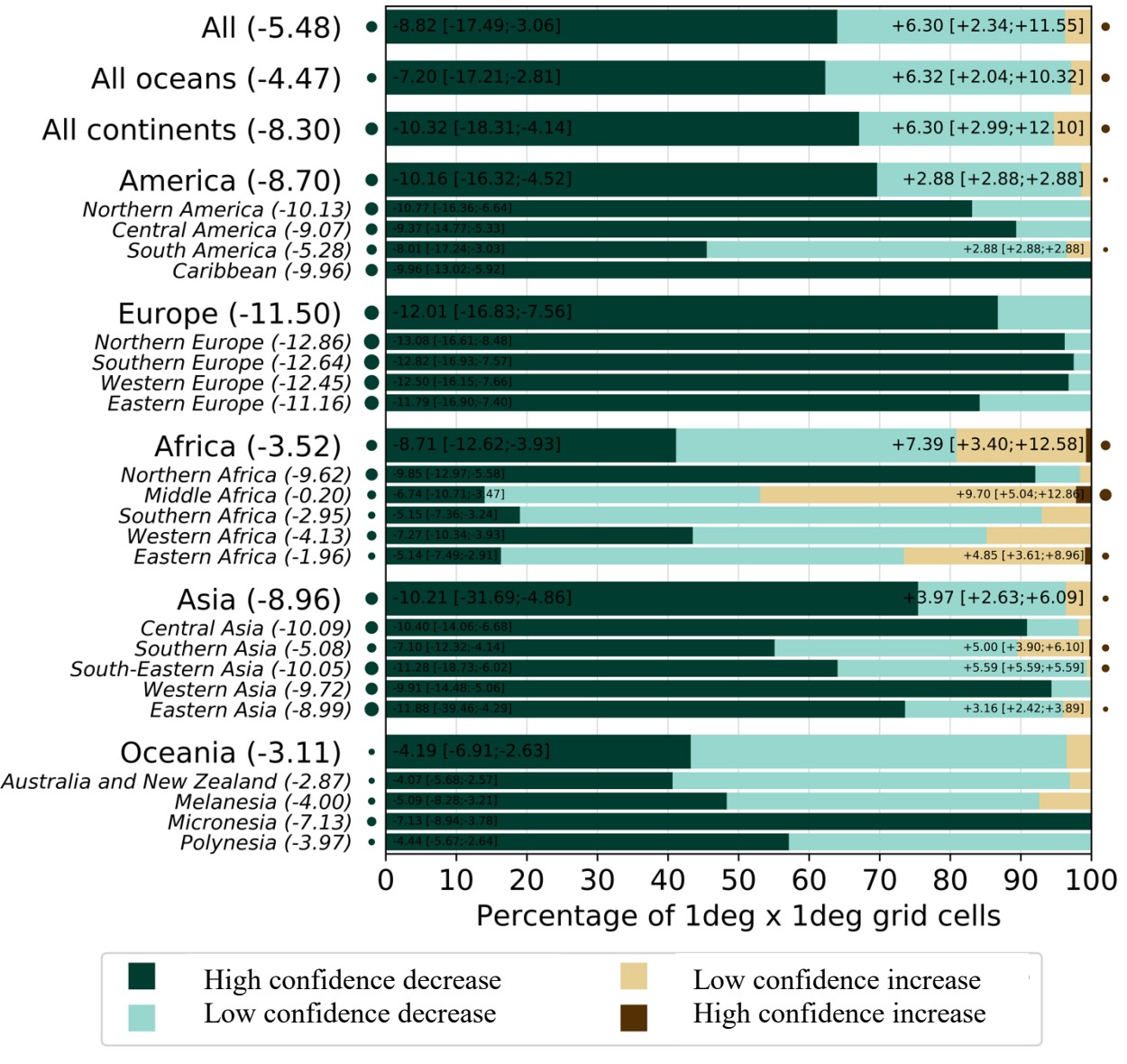

Figure 16: Summary of the statistically high- and low-confidence regional trends of MOPITT TC-CO trends over 2005-2019, at a 95% confidence level (see text for details on the calculation of the trends). For each region, the trends reported on the left (resp. right) represent the 50th[5th, and 95th] percentiles of the trends calculated over the different grid cells showing a high-confidence TC-CO increase or decrease.

Shown below are also the trends in the MOPITT column average volume mixing ratio (VMR) anomalies from 2005 to 2019 (Figure 17) using QR as well as Weighted least squares (WLS)) as Buchholz et al. (2021). The region boundaries are the same as used in Fig. 10 and 11. Results show a HC decreasing trend in the NH (-0.35 ±0.1% annually), a smaller decreasing trend in the Mid-latitudes (-0.26 ±0.1% annually), and LC trend in the SH (-0.14 ±0.1% annually). The three anthropogenic regions investigated in the NH all show strong decreases in CO. The larger negative trend over Australia (-0.2 ±0.1% annually) than the average SH, suggests sources from

the other two land regions (Southern Africa and South America) may be counteracting negative
trends in CO for the SH.

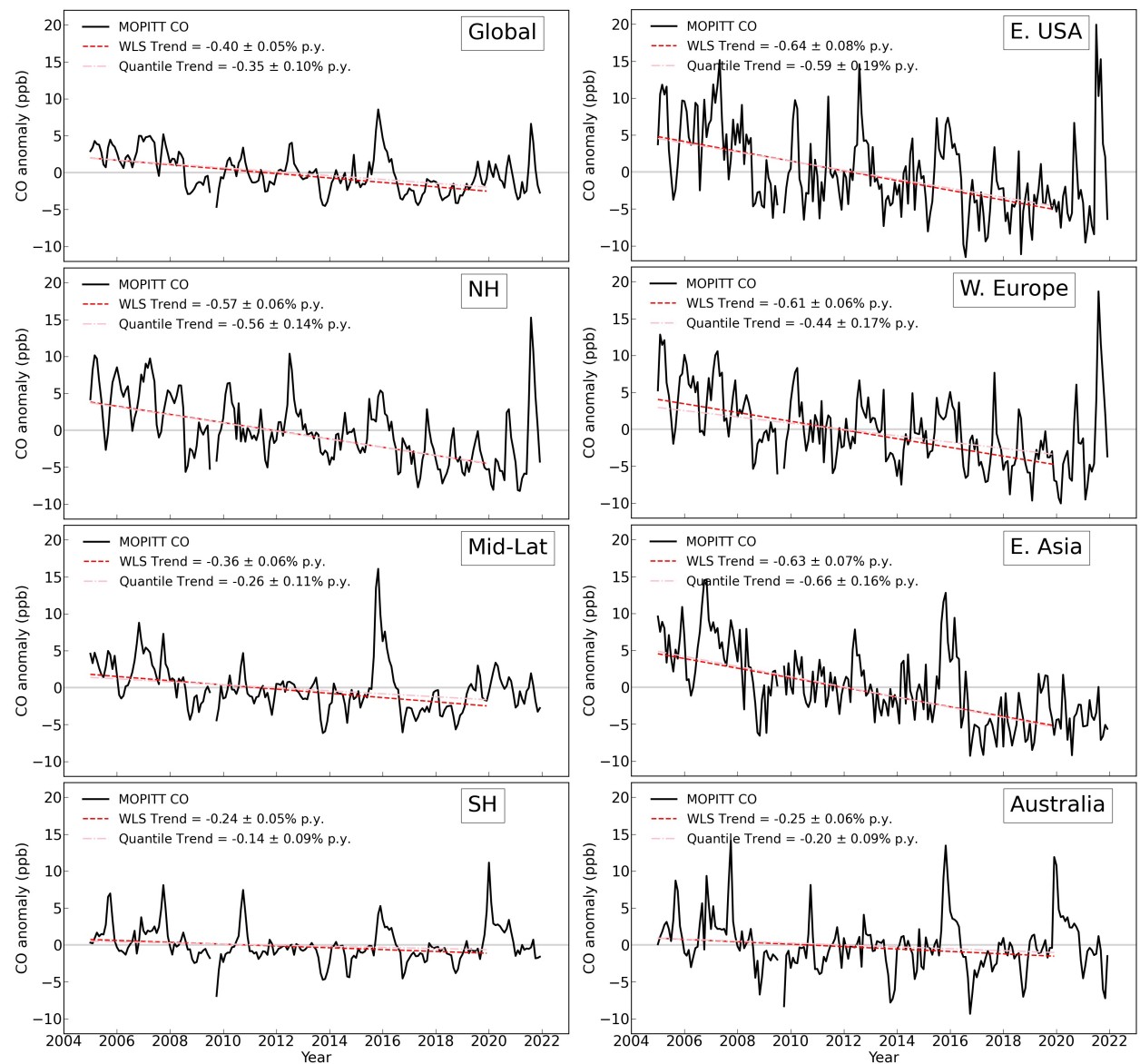


Figure 17: MOPITT monthly average CO anomalies in column average volume mixing ratio
(VMR, ppb), 2005-2021 (black). Updated dataset based on Buchholz et al. (2021). Data is Level
3, monthly average daytime observations, using version 9 joint NIR/TIR retrievals (V9J).
Regions are defined in Figure 10 and Figure 11. Trends are calculated on anomalies 2005-2019.
The weighted Least Squares trend (red) is weighted by the monthly regional standard deviation.
The quantile regression trend is also shown (pink). Grey dashed lines indicate a zero trend.

We also compare CO trends with Community Earth System Model (CESM) simulations
(Supplement Fig S1). While the magnitude of modeled CO tends to be underestimated relative to
observations, the anomalies between the model and measurements are comparable, indicating the
model reproduces interannual variability well. The negative trends in the NH are also reproduced
by CESM, although to a smaller degree than observations, suggesting that the trends in sources
or loss processes (such as OH oxidation) are underestimated in the model. These processes will
impact the feedback into modeled ozone and the resulting interpretation of driving factors for
ozone abundance and variability. Interestingly, CESM correctly represents a negative trend in
CO for the NH and East Asia while GEOS-GMI has a positive CO trend in those regions (Fig.
11), likely due to the well-known misrepresentation of East Asia air quality improvements in
emission inventories (Yin et al, 2015; Strode et al., 2016; Zheng et al, 2019). In the SH, CESM
does not predict HC trends.

### 3.4.6. HCHO Trends
HCHO, mainly a photochemical product results from hydrocarbon oxidation. HCHO is itself a
source of OH and ozone through its photolysis producing $HO_2$, which can be recycled back to
OH if sufficient NO levels are present.
R 3.4-1 $\qquad HCHO + h\nu\ (\lambda < 325\ nm) \rightarrow H + HCO$
R 3.4-2 $\qquad H + O_2 + M \rightarrow HO_2 + M$
R 3.4-3 $\qquad HCO + O_2 \rightarrow HO_2 + CO$
R 3.4-4 $\qquad HO_2 + NO \rightarrow OH + NO_2$
Unlike higher aldehydes, the OH reaction with HCHO leads also to the formation of a formyl
radical (HCO), which ultimately forms $HO_2$ (R 3.4-3).
R 3.4-5 $\qquad HCHO + OH \rightarrow H_2O + HCO$
Due to its solubility, the variability of HCHO also depends on the presence of clouds, and wet
deposition ultimately represents another important sink for HCHO (Lelieved and Crutzen, 1991).
Overall, HCHO plays a key role in the $O_3$ budget, both in polluted and remote regions.
Trends of the OMI HCHO tropospheric columns (hereafter referred to as TrC-HCHO) are
computed as described for OMI TrC-$NO_2$. TrC-HCHO trends over 2005-2019 are shown in
Figure 18 with a regional summary in Figure 19. The first global feature to highlight on the
global trends map is the presence of stripes along the OMI orbits. The number of rows affected
by the OMI row anomaly has increased over the years (Boersma et al., 2018). The affected rows
are filtered out in the HCHO data, but the change in the sampling and the related increase in the
noise impact the trend analysis. Along orbit stripes in the trend analysis should be ignored but
zonal trends are still valid (Figure 18).

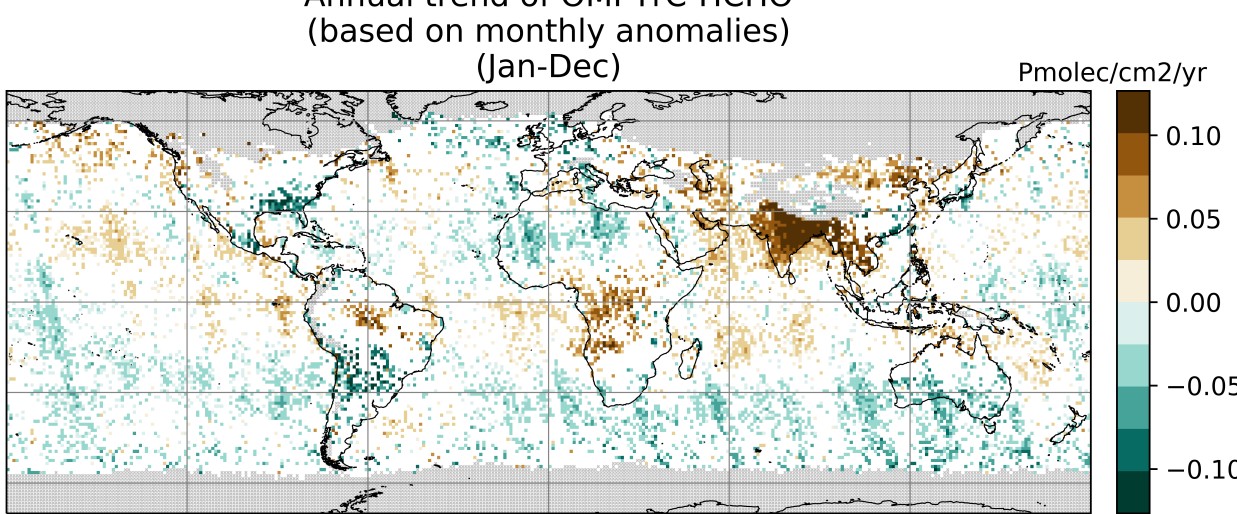

Annual trend of OMI TrC-HCHO
(based on monthly anomalies)
(Jan-Dec)
Pmolec/cm2/yr


Figure 18: Global trends of OMI HCHO tropospheric column (TrC-HCHO) over 2005-2019 (see text for details on the calculation of the trends). Grey areas correspond to areas without enough data, white areas correspond to regions where the trends remain statistically low-confidence at a 95% confidence level.

Despite the fact that TrC-HCHO trends remain LC over a large part of the globe, specific regions do highlight clear trends. The region with clearest changes is unambiguously southern Asia where about 65% of the cells show increasing trends with a median of +0.09 Pmolec/cm$^2$/yr. The other regions with a large portion (25-30% of the cells) of increasing trends include the rest of Asia and central Africa, with median TrC-HCHO trends ranging between +0.05 and +0.08 Pmolec/cm$^2$/yr, as well as some parts of central Brazil (Amazonians). Conversely, some HC decreases of TrC-HCHO are observed in the south-eastern US, the southern half of Southern America, North and western Africa, and southern Australia, although part of them overlap with the aforementioned stripes and might thus not be real.

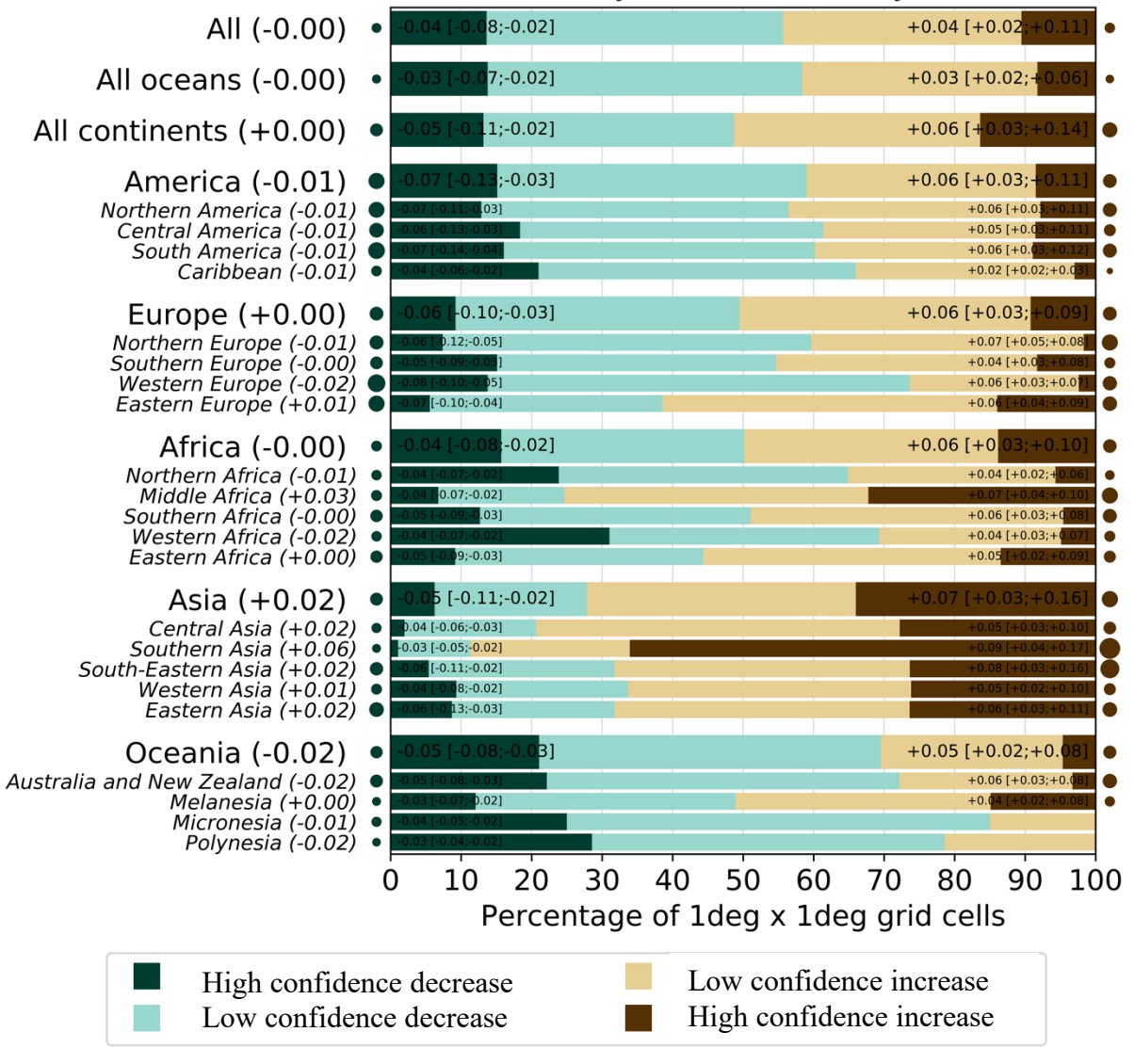

Figure 19: Summary of the statistically high- and low-confidence regional trends of OMI HCHO tropospheric column (TrC-HCHO) trends over 2005-2019, at a 95% confidence level (see text for details on the calculation of the trends). For each region, the trends reported on the left (resp. right) represent the 50th[5th, and 95th] percentiles of the trends calculated over the different grid cells showing a HC TrC-HCHO increase or decrease.

HCHO trends varies with that of $O_3$ (sec. 3.4.1) which might be due to several factors, such as their different sensitivity to $NO_x$ and hydrocarbons (Luecken et al., 2018) but also possible STE contribution to tropospheric ozone levels, especially in midlatitudes (Willimas et al., 2019; Li et al., 2024). For example, while TrC-$O_3$ is increasing in the southeastern US, TrC-$NO_2$, TC-CO, and TrC-HCHO are decreasing, which, in addition to the local chemistry, might indicate a STE signal. TrC-$NO_2$ trends are decreasing over the northern coast of Australia while those of TrC-$O_3$ and TrC-HCHO are increasing. While the increase of HCHO/$NO_2$ might indicate a trend toward NO-limited conditions (see below), the increase of TrC-$O_3$ trends in this region might also indicate increasing trends of STE contribution (Li et al., 2024). However, TrC-HCHO trends are

consistent with that of TrC-$O_3$ in other regions, e.g., over the northeastern US and Europe.
Similarly, while $NO_2$ trends are slightly increasing over central and southern Australia, trends of
TrC-$O_3$ and TrC-HCHO are decreasing, which indicates a trend toward VOC-limited conditions
(see below).

**3.4.7.    HCHO/$NO_2$**
The ratio of TrC-HCHO/TrC-$NO_2$ observed from space (e.g., Martin et al., 2004) has been used
in a number of studies to give insights on the $O_3$ chemical regime, higher (resp. lower) TrC-
HCHO/TrC-$NO_2$ ratios indicate $NO_x$-limited (resp. $RO_x$-limited) regimes. Although imperfect
(e.g. Souri et al., 2023), this indicator yet provides some qualitative information on the evolution
of the $O_3$ regime over the last years (Nussbaumer et al., 2023). We note that this analysis does not
consider variations in the ratios and their trends with respect to season or altitude. The mean TrC-
HCHO/TrC-$NO_2$ over 2005-2019 are shown in Figure 20, and the trend results are in Figure 21
with a regional summary in Figure 22. The highest ratios are observed in the tropical regions due
to strong TrC-HCHO from biogenic sources and fire NMVOC emissions in tropical South America
and Africa combined with relatively low TrC-$NO_2$. Conversely, lower TrC-HCHO/TrC-$NO_2$ ratios
are observed across western Europe and north-eastern Asia, and to a lesser extent, the northeastern
US.

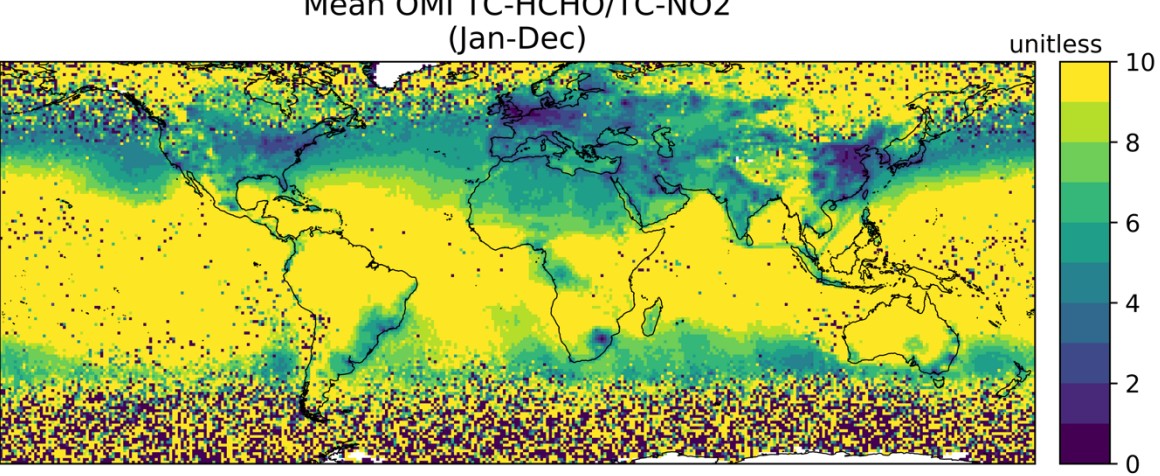

Figure 20: Global mean OMI HCHO/$NO_2$ tropospheric column ratio over 2005-2019.
At a global scale, the HC changes in TrC-HCHO/TrC-$NO_2$ trends (Figure 21-Figure 22) mostly
go in the direction of a reduction, with about 25% of the grid cells showing a median trend of -
0.52 yr$^{-1}$. (while only 5% of the cells show an HC increase of +0.03 yr$^{-1}$) as shown in Figure 22.
This suggests that these areas are evolving toward VOC-sensitive conditions (which does not
necessarily imply that they are already in this regime). This situation is observed over a large part
of Oceania (especially Polynesia) and specific parts of Africa, Asia, and South America. The
opposite HC trends, toward more NO-sensitive conditions, are mainly observed over Europe and
northern America, as well as South Asia. We note that the mean TrC-HCHO/TrC-$NO_2$ indicates
the mean status of the chemical regime over this period of time (2005-2019). However, the trends
of the TrC-HCHO/TrC-$NO_2$ ratio show the changing sensitivity of the chemical regime over this
period of time. For example, while the ratio in the Eastern US indicates VOC-sensitive conditions,
the trends of TrC-HCHO/TrC-NO$_2$ indicate a direction toward NO-sensitive conditions.

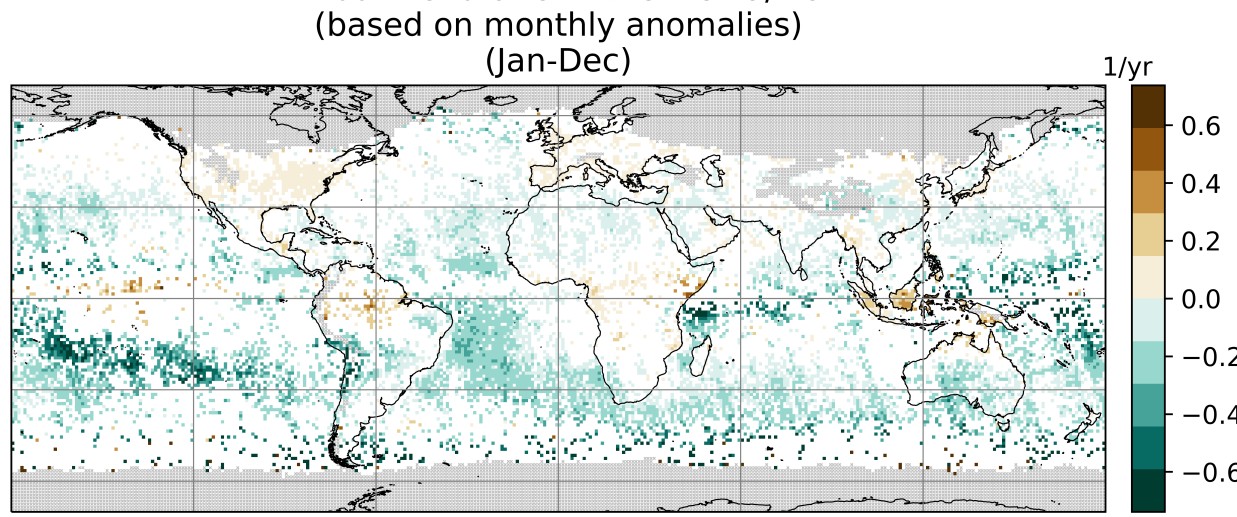

Figure 21: Global trends of OMI HCHO/NO$_2$ tropospheric column ratio over 2005-2019 (see
text for details on the calculation of the trends). Grey areas correspond to areas without enough
data, white areas correspond to regions where the trends remain statistically low-confidence at a
95% confidence level.
The trends on the TrC-HCHO/TrC-NO$_2$ ratio is mainly driven by specific trends on TrC-HCHO
and/or TrC-NO$_2$, depending on the region. The ratio increase in southern and western Europe and
southeast Asia appears primarily due to decreasing TrC-NO$_2$, since TrC-HCHO does not change
with HC. Over North America, observed TrC-HCHO values decrease but less than TrC-NO$_2$,
which thus drives the ratio toward an increase. Conversely, the increase of TrC-HCHO/TrC-NO$_2$
in equatorial Africa and Amazonians appears mainly driven by increasing TrC-HCHO. The
regions with HC decreasing TrC-HCHO/TrC-NO$_2$ ratio include Chile and Australia, due to both
decreasing TrC-HCHO and increasing TrC-NO$_2$ (Figure 22), indicating a trend towards a VOC-
limited regime. Note that over the US, Jin et al. (2020) demonstrated the reasonable ability of the
OMI-based TrC-HCHO/TrC-NO$_2$ trends to capture the transition from RO$_x$-limited to NO$_x$-limited
regimes over main US cities and found a relatively good consistency between observed changes
of the surface O$_3$ and space-based HCHO/NO$_2$ increasing trends.

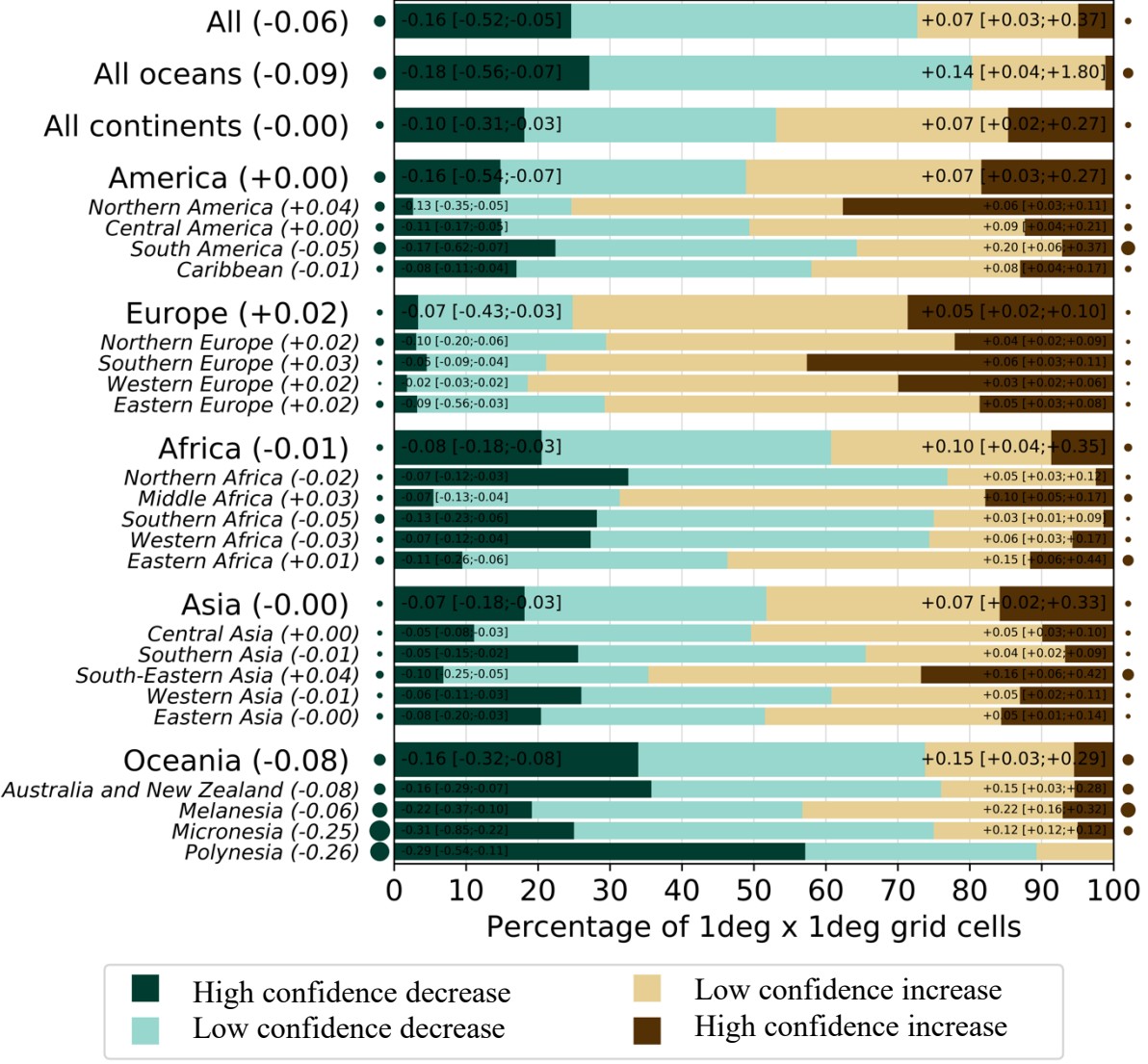

Figure 22: Summary of the statistically high- and low-confidence regional trends of OMI TrC-HCHO/TrC-NO₂ tropospheric column ratio trends over 2005-2019, at a 95% confidence level (see text for details on the calculation of the trends). For each region, the trends reported on the left (resp. right) represent the 50th[5th, and 95th] percentiles of the trends calculated over the different grid cells showing a high confidence TrC-HCHO/TrC-NO₂ increase or decrease.

778

### 3.5. Lightning NO$_x$ and Its Effects on Tropospheric NO$_x$ and O$_3$

779

Nitric oxide (NO) is produced in lightning flash channels and quickly comes into equilibrium with NO$_2$. Cloud-scale simulations of thunderstorms indicate that 55-75% of lightning NO$_x$ (LNO$_x$) is detrained above 8 km (Pickering et al., 1998) where it enhances upper tropospheric NO$_y$, OH, and O$_3$ (Labrador et al., 2005; Allen et al., 2010; Liaskos et al., 2015) and contributes to enhanced longwave radiative absorption by O$_3$ (Lacis et al., 1990; Finney et al., 2018). Enhanced OH leads to a decrease in CH$_4$ lifetime and decreased longwave radiative absorption (Fiore et al., 2006; Finney et al., 2018). The lifetime of NO$_x$ in the upper troposphere is controlled by the chemical cycling of NO$_x$ with reservoir species and is 10-20 days away from deep convection (Prather and Jacob, 1997) but only 2-12 hours in the vicinity of convection (Nault et al., 2016, 2017). This chemical recycling provides a source of NO$_x$ downwind of thunderstorms, which causes the ozone production efficiency of emitted NO$_x$ to be 4-20 times higher in the upper troposphere than at the surface. Thus, LNO$_x$ has a disproportionate impact on the tropospheric O$_3$ budget (Pickering et al., 1990; Grewe et al., 2001; Sauvage et al., 2007).

The distribution of lightning is fairly well known over much of the Earth due to remote sensing observations and an increase in the number and capability of ground-based lightning networks. However, the LNO$_x$ production efficiency (PE, mol fl$^{-1}$) is a continued source of uncertainty. Schumann and Huntrieser (2007) reviewed the literature on LNO$_x$ production, finding a best estimate of 250 moles per flash, with uncertainty factors ranging from 0.13 to 2.7. The PE can be estimated from theoretical and laboratory considerations (Price et al., 1997; Koshak et al., 2014), using thunderstorm anvil observations by aircraft (Ridley et al., 2004; Huntrieser et al., 2008, 2011; Pollack et al., 2016; Nault et al., 2017; Allen et al., 2021a), based on satellite data (Bucsela et al., 2010; Beirle et al., 2010; Pickering et al., 2016; Bucsela et al., 2019; Lapierre et al., 2020; Zhang et al., 2020; Allen et al., 2019, 2021b), or using cloud-resolved (e.g., DeCaria et al., 2000; 2005; Fehr et al., 2004; Ott et al., 2007, 2010; Cummings et al., 2013; Pickering et al., 2023) or global model simulations with chemistry (e.g. Martin, et al., 2007; Murray et al., 2012; Miyazaki et al., 2014; Marais et al., 2018). These various techniques have yielded PE estimates ranging from <50 to >1000 mol fl$^{-1}$, with most estimates in the 100-400 mol fl$^{-1}$ range. Miyazaki et al. (2014) assimilated OMI NO$_2$, MLS and TES O$_3$, and MOPITT CO into a chemical transport model to provide comprehensive constraints on the global LNO$_x$ source, resulting in an estimate of mean PE of 310 moles per flash. Marais et al. (2018) used cloud-sliced upper tropospheric NO$_2$ from OMI together with the GEOS-Chem model to estimate a mean LNO$_x$ PE of 280 moles per flash. Lightning is the dominant source of NO$_x$ in the tropical upper troposphere year-round and in the northern mid-latitudes in summer. Lightning is responsible for 10-15% of NO$_x$ emissions globally. Assuming 100-400 mol fl$^{-1}$, the global LNO$_x$ production is likely 2 – 8 Tg N a$^{-1}$ (Schumann and Huntrieser, 2007; Verma et al., 2021).LNO$_x$ impacts air quality and deposition (Kaynak et al., 2008; Allen et al., 2012). On average LNO$_x$ adds 1-2 ppbv to surface O$_3$ (Kang et al., 2019b), although contributions as large as 18 ppbv have been seen for individual events (Murray et al., 2016). Allen et al. found that the addition of LNO$_x$ to the Community Multiscale Air Quality (CMAQ) model increased wet deposition of oxidized nitrogen at National Atmospheric Deposition Program (NADP) sites by 43%, reducing low biases from 33% to near-zero. Kang et al. (2019b) found similar improvements for wet deposition and also found that including LNO$_x$ resulted in smaller biases with respect to ozonesondes and aircraft profiles taken during the NASA DISCOVER-AQ field campaign (Flynn et al., 2016). Thus, to accurately assess its impacts on air quality, it is critical that LNO$_x$-producing deep convection is accurately simulated.

Only in recent years with the advent of satellite observations of lightning flashes and improved coverage by ground-based lightning networks has there been sufficient data to make estimates of trends in the occurrence of lightning. However, it is unknown whether trends in $LNO_x$ production are similar to those of lightning itself. Lightning characteristics such as the ratio of intracloud (IC) flashes to cloud-to-ground (CG) flashes, the multiplicity (i.e., the number of strokes per flash), and the peak current or energy associated with flashes may vary over time. All of these lightning characteristics may have effects on the magnitude of LNOx production. We have insufficient data to take into account these possible effects on $LNO_x$ production over large spatial domains or over sufficiently long periods of time.

### 3.5.1. Global Historical Trends of Lightning

The first attempts at an examination of trends in thunderstorm activity were conducted in terms of thunder-days (in Japan by Kitagawa et al., 1989; in Brazil by Pinto et al., 2013). A more recent global analysis was conducted by Lavigne et al. (2019), who analyzed trends in thunder-days (number of days with audible thunder at weather observation stations) over 43 years and in flashes recorded by the Lightning Imaging Sensor (LIS) on the Tropical Rainfall Measuring Mission (TRMM) for 16 years. Thunder-days increased since the 1970s in the Amazon Basin, the Maritime Continent, India, Congo, Central America, and Argentina. Decreases in thunder-days were found in China, Australia, and the Sahel region of Africa. Lavigne et al. (2019) do not provide a global trend in thunder days, but an average trend computed over the nine primary lightning regions that they considered, weighted by the mean annual thunder days in each region, yields a near global estimate of +3.8% per decade. How well do thunder-days represent lightning flash rate? Lavigne et al. found a positive correlation between thunder-days and LIS flash rates in China, the Maritime Continent, South Africa and Argentina, but disagreement on the trend in India and West Africa.

Large-scale (±38° latitude) trends in lightning flashes have been examined in the data collected by the LIS on the TRMM satellite (January 1998 – December 2014) and on the International Space Station (February 2017 – December 2021). Füllekrug et al. (2022; see Figure SB2.1b) demonstrate that the annual mean deviations from the 1998 – 2021 mean are no more than ~5% except for ~-10% in 2020 and ~-8% in 2021. However, no long-term trend is evident from the LIS data. The possibility that these larger negative deviations in 2020 and 2021 are due to Covid-19 lockdowns and general declines in economic activity has been speculated. The link may be provided by changes in Aerosol Optical Depth (AOD) as suggested by Liu et al. (2021) who demonstrated 10-20% flash reductions in March – May 2020 relative to the 2018 – 2021 mean for those months from the GLD360 and WWLLN ground-based lightning networks. Regional lightning reductions were consistent with AOD reductions noted by Sanap (2021). Larger reductions in lightning were noted over Africa/Europe and Asia/Maritime Continent and lesser reductions over the Americas.

### 3.5.2. Regional Historical Trends of Lightning

Widely varying trends in lightning over China have been reported in the literature. To some extent, whether the trend in lightning is upward or downward depends on the particular region studied and on the period of time considered. Yang and Li (2014) were the first to report on lightning trends in China. They used lightning data from the TRMM/LIS sensor and human-observed thunderstorm day occurrence over the period 1990 to 2012 in southeastern China. Thunderstorms and lightning occurrence increased over the period as well as LIS precipitation radar echo tops heights. These increases were accompanied by decreases in visibility, indicating increases in pollution aerosol. Detailed work on lightning trends in China has been performed in relation to aerosols. Shi et al. (2020) correlated flashes from the TRMM/LIS Low-Resolution Monthly Time Series (2.5 deg. resolution) with AOD from MODIS-Terra V6.1 Level 3 over the period 2001 to 2014. For AOD

< 1.0, r = 0.64, indicating a likely microphysical effect on lightning flash rate. For AOD > 1.0, r =
-0.06, which could indicate that with higher aerosol concentration there is a radiation effect
stabilizing the atmosphere and/or a decrease in the number of graupel particles in the mixed-phase
region of the storms that is important for charging. Flashes were also correlated with surface
relative humidity and Convective Available Potential Energy (CAPE). As AOD generally
increased over much of the early portion of this time period and then decreased, lightning flash
rates followed similar trends. Wang et al. (2021) examined a 9-year record (2010- 2018) of CG
lightning from the China Lightning Detection Network in three polluted urban areas of China
(Chengdu, Wuhan, and Jinan). They found decreasing trends (see Wang et al., 2021) in CG
lightning and total AOD (from the MERRA-2 reanalysis). Annual mean lightning density in these
three regions decreased by 50 – 75% as annual mean AOD fell from 0.70 – 0.75 to 0.53 to 0.62.
Qie et al. (2022) analyzed the OTD/LIS record from 1996 through 2013, and found that lightning
increased over the eastern Tibetan Plateau by $0.072 \pm .069$ fl km$^2$ yr$^{-1}$. Over the 18 years, this
increase amounted to a total of 1.3 fl km$^2$ yr$^{-1}$, compared with a climatological value of 7.7 fl km$^2$
yr$^{-1}$, thereby indicating a HC increase. The ground-based World Wide Lightning Location Network
(WWLLN) also showed increased strokes in this region. The increase in lightning frequency in
this region was found to be due to an increase in thunderstorm frequency, not increased storm
intensity.
Koshak et al. (2015) analyzed National Lightning Detection Network (NLDN) CG flashes over
the contiguous United States (CONUS) from 2003 to 2012. The five-year mean flashes over 2008
to 2012 decreased by 12.8% from the five-year mean for 2003 to 2007 (Table 1). The CONUS
average wet bulb temperature also trended downward during this period, which may have led to
lesser or weaker storms. However, US Environmental Protection Agency air quality trends show
an 18% decrease in PM2.5 concentrations over CONUS between the two subperiods, which also
could have had an influence on the flash rates. A recent effort to update the Koshak et al. (2015)
analysis is underway. NLDN flashes have been reprocessed (Kenneth Cummins, personal
communication) from 2015 through 2021 to ensure that the classification of IC and CG flashes is
done consistently with data prior to 2015. Trend analysis of NLDN CG flashes from 2003 (a major
upgrade of the NLDN network hardware) through 2022 (William Koshak, personal
communication) shows a HC reduction in CG flashes over CONUS, comparing the mean CG
flashes over 2003-2004 with the mean over 2021 -2022. Within this period a major decrease
(~25%) in CONUS CG flashes occurred from 2011 to 2012. Flashes in 2013 remained low, but
recovered by 2014-2015. A major decrease (~27%) occurred from 2019 to 2020, with a small
increase in 2021. These results have been obtained from ongoing efforts by Dr. William Koshak
of the NASA Marshall Space Flight Center, and are presently part of a draft manuscript by lead
author Koshak that extends and refines the earlier work in Koshak et al. (2015). Details concerning
these trends will be contained in that manuscript.
A possible contributing factor to the CONUs decline in CG flashes over 2003 to 2021 is the
substantial decrease in aerosol. Surface annual average PM2.5 concentrations averaged over
CONUS decreased by 37% from 2000 to 2021 according to the EPA National Air Quality Trends
Report (https://www.epa.gov/air-trends/air-quality-national-summary). However, no decrease in
CONUS annual average PM2.5 was seen from 2019 to 2020. As mentioned previously, AOD may
be a better indicator of the aerosol amount that may become incorporated into thunderstorm clouds.
Sanap (2021) showed negative anomalies of AOD of ~0.1 in portions of CONUS in March and
April 2020 and 0.1 to 0.2 in May 2020. The major decrease in CONUS CG flashes from 2011 to
2012 has been related to drought conditions during Summer 2012 over the South Central and
Southeastern US (Koshak et al., 2015). The reason for the number of CONUS flashes remaining
lower in 2013 is uncertain. Koehler (2020) analyzed 26 years (1993 – 2018) of NLDN CG
lightning data to construct a thunder-day climatology for CONUS. Positive anomalies from the
26-year mean were found from Texas to Colorado during 2003 to 2007, and negative anomalies
in this region during 2008 to 2012. These anomalies were consistent with precipitation anomalies
associated with ENSO.

Holzworth et al. (2021) analyzed primarily CG lightning data from WWLLN for June, July, and
August for the years 2010 through 2020. The ratio of lightning strokes north of 65$^{\circ}$ N latitude to
the total global strokes increased by a factor of three over this period. This increase occurred as
the surface temperature anomaly in this region increased by 0.3$^{\circ}$C (see Holzworth et al., 2021).
These results suggest a substantial increase in upper tropospheric $NO_x$ and subsequent ozone
production at high northern latitudes.

### 3.5.3.    Future Lightning Trends

Parameterizations in global chemistry and climate models have been developed for
lightning flash rate. These schemes typically use kinematic, thermodynamic or microphysical
variables from the model as predictors. In some studies such predictors have simply been applied
to output from multiple climate models. This is the case with the Romps et al. (2014) work, which
showed that when a lightning parameterization scheme using CAPE x Precipitation Rate is applied
to 11 climate models an increase in CG lightning by 12 +/- 5% per degree Celsius of climate
warming was computed. This work simply used the 12-hour resolution time series of spatial means
of these variables over CONUS as input. Changes in IC lightning flashes were not considered. IC
flashes typically outnumber CG flashes by a factor of 3 averaged over CONUS. Therefore, the
result of this work is unknown with respect to the amount of change in $LNO_x$ emission. Romps et
al. (2018) updated their analysis using CAPE from 3-hourly North American Regional Reanalysis
(NARR) data and hourly precipitation from NOAA River Forecast Centers, finding that CAPE x
Precipitation Rate captures the spatial, seasonal, and diurnal variations of NLDN CG flash rate
over land, but does not predict the pronounced land-ocean contrast in flash rates. Therefore, these
analyses are of limited value in estimating trends of $LNO_x$ over broader-scale regions. Romps et
al (2019) tested four lightning proxies in a cloud-resolved 4-km resolution simulation over
CONUS with the Weather Research and Forecasting (WRF) model, and over the tropical oceans
with a Radiative Convective Equilibrium model. The proxies were CAPE x Precipitation Rate,
precipitation with vertical velocity > 10 m/s, vertical ice flux at the 260K isotherm, and vertical
integral of cloud ice and graupel product. The fractional change in proxy values per 1 degree
Celsius of warming over CONUS was +8 to +16%. Over the tropical oceans the changes in proxy
values per degree ranged from +12% for CAPE x Precipitation Rate to -1% for ice flux and -3%
for the cloud ice and graupel product. Therefore, over broad regions of the Earth, there is great
uncertainty on future trends in lightning.
Finney et al. (2016; 2018) compared lightning projections for 2100 using vertical ice flux
(Finney et al., 2014) and cloud-top height parameterizations for flash rate in the UK Chemistry
and Aerosols Model. They obtained -15% global change in total flash rate with ice flux under a
strong global warming scenario (see Finney et al., 2018), which was composed of a greater
decrease in the tropics and small increases in mid-latitudes. In terms of $LNO_x$ emissions this work
using the ice flux scheme produced -0.15 TgN K$^{-1}$ change over the years from 2000 to 2100,
implying less $O_3$ production. With the cloud-top height scheme they obtained +0.44 TgN K$^{-1}$ $LNO_x$
change, implying increased $O_3$ production. However, the ice flux scheme provided a more realistic
representation of global lightning for present day. Therefore, the negative LNO$_x$ emissions change
from this scheme may be more realistic. If indeed the ice flux scheme better represents the current
distribution of lightning, both the Romps and Finney results suggest LC  increase in LNO$_x$
emission in future climate, and possibly a small global decrease. Murray (2018) points out that the
ice flux scheme is a closer representation of the underlying charging mechanism, but this scheme
needs to be tested in multiple global chemistry and climate models.

### 3.5.4.        Recent findings concerning LNOx PE
Recent satellite-based estimates of LNO$_x$ production (Figure 23) have suggested a possible flash
rate dependence of LNO$_x$ production per flash (Bucsela et al., 2019; Allen et al., 2019; 2021).
Smaller values of LNO$_x$ PE in these studies were found to be associated with high flash rates,
likely due to smaller flashes in these conditions (Bruning and Thomas, 2015). Allen et al. (2021a)
noted positive correlations (Figure 23) of LNO$_x$ PE with flash energy and with flash multiplicity
(number of strokes per flash). Laboratory studies by Wang et al. (1998) found a positive correlation
between peak current and LNO$_x$ production. Koshak et al. (2015) found an 8% increase in peak
current from the 2003-2007 period to the 2008-2012 period that accompanied the 12.8% decrease
in CG flashes. These findings make it difficult to project future LNO$_x$ production given only a
prediction of future lightning flashes.

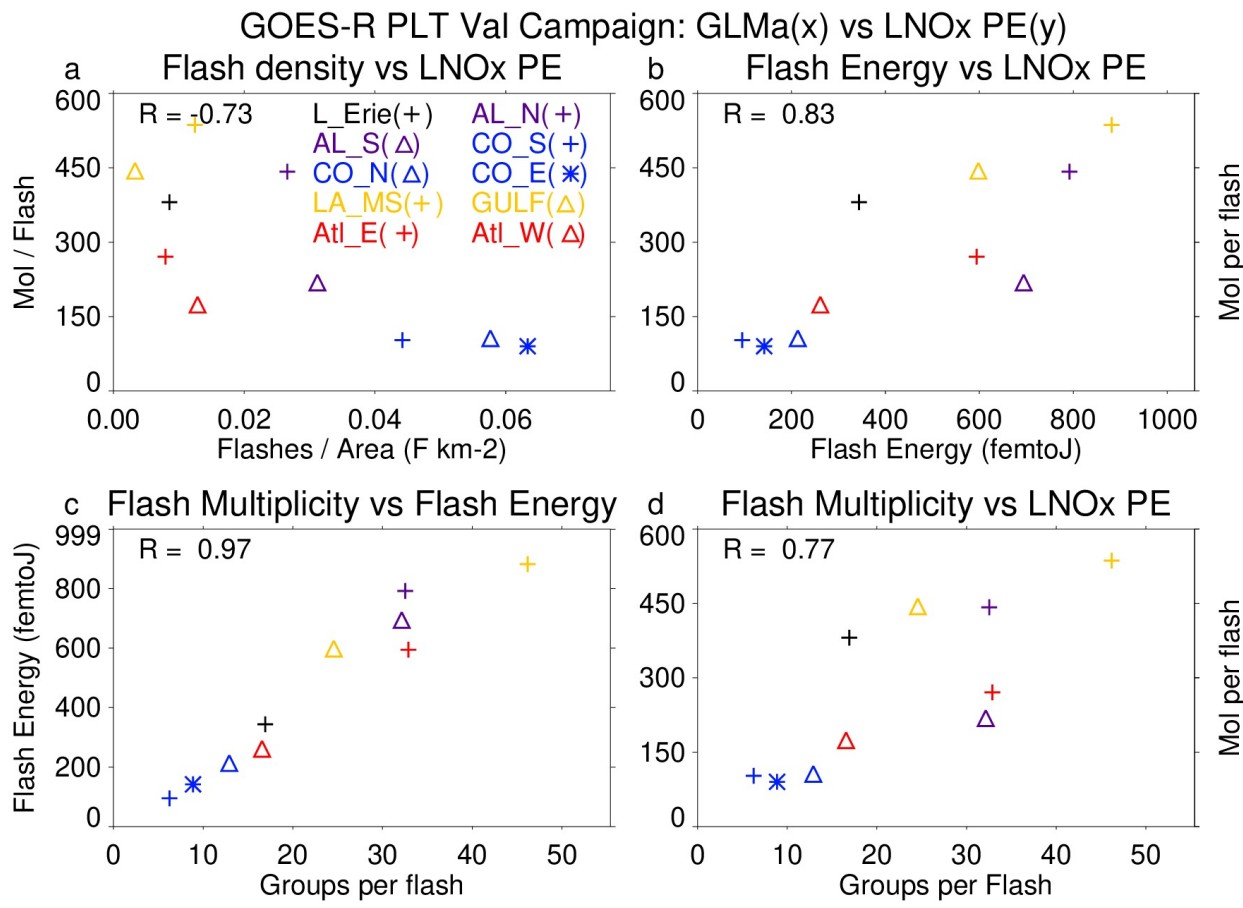


Figure 23. Scatterplots showing the GLMa-derived relationship between (a) LNO$_x$ PE (mol per
flash) and flash density (flashes km−2), (b) LNO$_x$ PE and flash energy (fJ), (c) flash energy and
flash multiplicity, and (d) LNO$_x$ PE and flash multiplicity. Colors are used to separate flight days
while symbols are used to separate system within each flight day. Correlations are shown in the
upper right. LNOx PE derived from airborne remote sensor, the Geo-CAPE Airborne Simulator
(GCAS) during the GOES-R Post-launch Test field campaign. GLMa indicates Geostationary
Lightning Mapper data adjusted for missing data. From Allen et al. (2021a).

### 3.5.5.     Impacts of LNOx on upper tropospheric O$_3$

The literature concerning the effects of lightning NO$_x$ production on upper tropospheric ozone
focuses on photochemical ozone production in storm outflow The STERAO-A storm simulation
by DeCaria et al. (2005) indicated that additional ozone production attributable to lightning NO
within the storm cloud during the lifetime of the storm was very small (~2 ppbv). However,
simulation of the photochemistry over the 24 hours following the storm showed that an additional
10 ppbv of ozone production in the upper troposphere can be attributed to lightning NO production.
Convective transport of HO$_x$ precursors led to the generation of a HO$_x$ plume, which substantially
aided the downstream ozone production. Ott et al. (2007) simulated the July 21, 1998 EULINOX
thunderstorm. During the storm, the inclusion of lightning NO$_x$ in the model combined with
convectively-transported boundary layer NO$_x$ from the Munich, Germany region resulted in
sufficiently large NO$_x$ mixing ratios to cause a small titration loss of ozone (on average less than
4 ppbv) at all model levels. Simulations of the chemical environment in the 24 hours following the
storm show on average a small increase in the net production of ozone at most levels resulting
from lightning NO$_x$, maximizing at approximately 5 ppbv per day at 5.5 km. Between 8 and 10.5
km, lightning NO$_x$ caused decreased net ozone production. Ren et al. (2008) found that net
tropospheric ozone production proceeded at a median rate of ~11 ppbv per day above 9 km in the
Intercontinental Transport Experiment (INTEX-A) in which the effects of frequent deep
convection over the United States dominated the upper troposphere. Apel et al. (2012) noted that
a box model calculation indicated a net ozone increase of ~10 ppbv over a few hours following
observed convection with lightning over Canada in the Arctic Research of the Composition of the
Troposphere from Aircraft and Satellite (ARCTAS) experiment. Apel et al. (2015) performed box
modeling of the chemistry downwind of two DC3 storms in northeast Colorado on June 22, 2012
finding greater ozone production over 2 days (14 ppbv) in the southern storm with more LNO$_x$
than in the northern storm (11 ppbv). Brune et al. (2018) studied ozone production in the outflow
of the June 21, 2012 DC3 mesoscale convective system. Their Box model calculations yielded a
13 ppbv increase in ozone over 5 hours, similar to the observed 14 ppbv increase. This rate of
increase is larger than others in the literature, perhaps because for a portion of the 5 hours the
outflow was in cirrus cloud, in which photolysis rates may have been larger than clear-sky values
due to multiple scattering. Using a regional chemistry model, Pickering et al. (2023) estimated that
net ozone production in the upper tropospheric outflow of a severe high flash rate storm observed
over Oklahoma proceeded at a rate of 10-11 ppbv day$^{-1}$ during the first 24 hours of downwind
transport. Downwind photochemical production of ozone due to LNO$_x$ accounted for much of the
recovery of upper tropospheric ozone following large reductions due to convective transport of
lower ozone boundary layer air.

### 3.5.6.     Summary of LNO$_x$

LNO$_x$ is responsible for the largest fraction of upper tropospheric ozone in the tropics year-round
and in the mid-latitudes in summer. Effects on longwave radiation due to ozone are most sensitive
due to the ozone near the tropopause. Therefore, it is of great importance to have knowledge of
the trends in ozone in this region that are due to changes in frequency and characteristics of
lightning flashes. Considerable uncertainty remains concerning trends in global thunder days. No
long-term trend in global flash rates has been found. However, regionally important trends have
been noted in CONUS and in China, which tend to be correlated to the decreasing atmospheric
aerosol content. An increasing trend at Arctic latitudes has been noted, as that region rapidly
warms. Future trends in flash rate also are uncertain, with conflicting predictions coming from
models with differing flash rate parameterizations. Flash characteristics (e.g., flash rate, flash
extent, flash energy or peak current, intracloud fraction) have been found to have important
implications for $LNO_x$ production per flash. Insufficient knowledge of these characteristics on a
global scale makes it highly uncertain to estimate changes in $LNO_x$ production, even with
knowledge of flash rate trends.

### 1043 3.6. Soil NO and HONO emissions and their impacts on $O_3$

Nitrous acid (HONO) is produced from microbial activity in soils with a similar mechanism and
strength as NO (Oswald et al., 2013). This emission source may partially account for the current
mismatch between observed and simulated HONO levels in the lower troposphere (Su et al.,
2011; Yang et al., 2020). Zhang et al. (2016) estimate a 29 % contribution of soil-HONO to the
HONO sources in China. This may also contribute substantially to OH production with important
implications for the $HO_x$ and $O_3$ budget. To account for this emission source and assess the
global potential for atmospheric pollution soil-HONO emissions have been parameterized based
on the HONO/NO emission ratio measured at multiple field samples (taken from different
regions of the world) and up-scale it to the 4 major land cover types applied to the whole globe.
The study estimates a global emission source of 7 TgN/yr from soil-HONO in 2009 (Emmerichs
et al., 2023). This is at the lower end of the estimated range of 7.4-12 TgN/yr presented by Wu et
al. (2022) for 2017 who employ an empirical and statistical model in combination with
observations. Due to the importance of NO and HONO soil emissions for the $O_3$ budget their
variability and historical and future trends are described here and linked to $O_3$. Additionally, we
discuss a modification of the soil NO emission scheme.

### 1059 3.6.1. Global modeling of reactive nitrogen emissions from soil

In this section, we present a short overview of the soil-NO emission algorithms and estimates for
regional and global emissions. The emission of nitrogen oxides (NO) from the soil is the major
source of $NO_x$ in unpolluted regions accounting for 15-25 % of global emissions (Weng et al.,
2020, Vinken et al., 2014). Thereby, NO is produced from the nitrification in soil (microbial
activity) and depends non-linearly on soil properties like pH, carbon and nutrient content,
temperature, and soil moisture (Gödde and Conrad 2000, Oswald et al. 2013). Model algorithms
estimate soil-NO emissions with a function dependent on biological and meteorological drivers.
The common empirical approach by Yienger and Levy (1995), which is used in the current
CMIP6 simulations (Szopa et al. 2022), is based on a biome-specific emission factor, soil
temperature, precipitation, and the canopy uptake reduction factor. The resulting global estimate
is in the range of 3.3-7.7 TgN/yr which is, however, only at the lower end of the more recent
model and observation-based estimates. The Yienge and Levy (1995) approach generally
underestimates soil NO for all landcover types except in the tundra and rainforest due to the
pulsing parameterization, which describes a large NOx release at the wetting of very dry soil and
the subsequent rapid decay (Steinkamp et al., 2009). This is accounted for in the more
mechanistic approach by Hudman et al. (2012) representing pulsing of the emissions following
dry spells and N-inputs from chemical fertilizer and atmospheric N-deposition. This approach
calculates spatial and temporal patterns of soil moisture, temperature, pulsing, fertilizer, manure
and atmospheric N deposition and biome overall replacing the emission factors by Yienger and
Levy (1995) which yields in comparison 34 % more annual global soil emissions of nitrogen
oxide (10.7 TgN/yr). Satellite top-down estimates range from 7.9 TgN/yr (Miyazaki et al., 2017:
2005-2014, assimilation of satellite data sets) to 16.7 TgN/yr (Vinken et al., 2014; GEOS-Chem
and OMI). The emission of soil-NO varies regionally with small sources in Australia (~0.5
TgN/yr), Europe, Russia and Southern Hemisphere (SH) Africa (0.7 TgN/yr, 0.8 TgN/yr),
America (0.9-1 TgN/yr) and high values in S.E. Asia and Northern Hemisphere (NH) Africa (2-
2.1 TgN/yr). The emission estimates (here for 0.25° lat. × 0.3125° lon.) increase with resolution
in some regions like Europe by 38 % (Weng et al., 2020).
Nitrous acid (HONO), a major OH source, is also produced from microbial activity in soils with
a similar mechanism and strength as NO (Oswald et al., 2013). This additional emission source
may account for the current mismatch between models and measurements representing HONO
levels in the lower troposphere (Su et al., 2011; Yang et al., 2020). Soil emissions of HONO play
a major role in the daytime-HONO concentrations in rural areas (in the lowest layers) where
traffic emissions and $NO_2$ heterogeneous reactions occur less than in urban areas (Wu et al.
2022). HONO photolysis is a main OH source and impacts the oxidation capacity of the
atmosphere (Zhang 2016, 2019). Therefore, this may also contribute significantly to OH
production with important implications for the $HO_x$ and $O_3$ budget.

### 3.6.2. Variability and trends of soil emissions of NO and HONO in the last 15 years

The magnitude of soil emissions varies strongly with season where the emissions rise from
January and July by a factor of 2.5 (Weng et al., 2020). This follows the meteorological
variability as for instance, heavy rainfall over dry grasslands/forests causes a pulse of soil NO
emissions coupled with the usage of fertilizer (Hudman et al., 2012). According to the CCMI
simulations by Jöckel et al. (2016) (following the future ('medium high') climate scenario
RCP6.0 the soil NO emissions show a positive trend since pre-industrial times with a steeper
increase of up to 0.3 TgN/decade from the year 2000. As soil emissions of HONO rely on the
same biogeochemical process with similar dependencies on temperature and water content as NO
also increased from 2000 to 2019.
For soil-HONO, however, the trend over 2005-2019 is much smaller, most pronounced in
Central Africa (Figure 25). Thereby, the highest positive monthly anomalies occur mainly in the
5 most recent years which is likely due to the more frequent heat wave occurrence, e.g. in Europe
and North America. Overall, Africa relates the most (~30%) to the global anomaly (Figure 24 -
Figure 25).

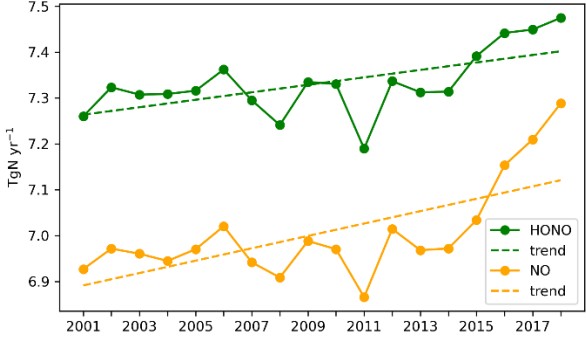
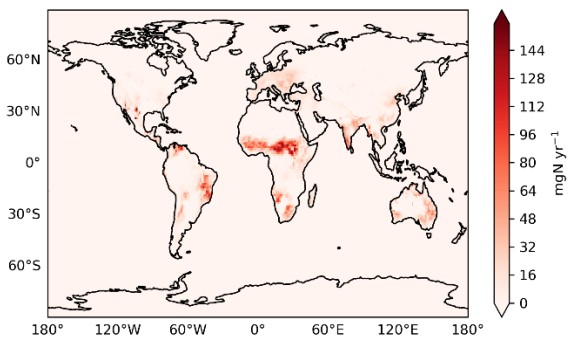


Figure 24: Time series of soil-HONO and soil-NO emissions and their trends (left) and the mean
global distribution of the soil-HONO emission trend for 2005-2019 based on monthly anomalies
(right).

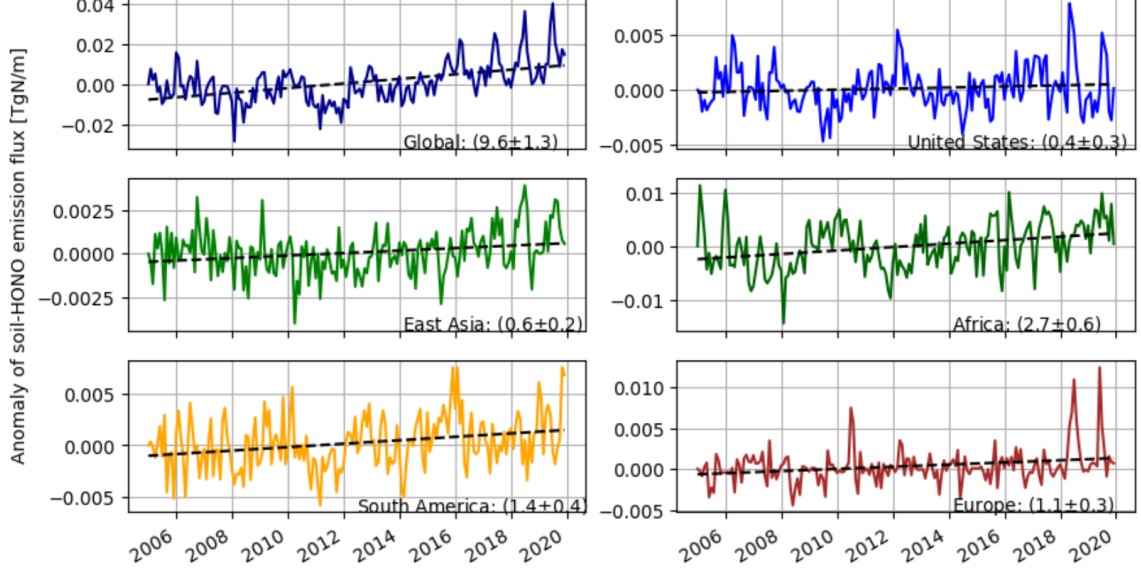


Figure 25: Monthly anomalies of HONO emissions from soil (de-seasonalized). The trend is
given in $10^{-5}$, including the uncertainty estimate (2*standard deviation).

### 3.6.3.    Canopy Reduction Factor

Most NO soil emission models (Yienger and Levy, 1995; Hudman et al., 2012) rely on an
empirical canopy reduction scheme which represents loss processes in plants as the diffusion of
$NO_2$ through the stomata and direct deposition to the cuticle. In particular, a large fraction of
$NO_x$ (and peroxyacyl nitrate) loss during the night may be only explainable by non-stomatal
processes (Delaria et al., 2020b). Mechanistically, the canopy reduction can be described by an
efficient NOx deposition to plants. Thus, Delaria et al. (2020a) points out that models already
represent the uptake by vegetation and do not need to use a canopy reduction scheme. The
potential change of NO soil emissions is shown by employing the global model
ECHAM/MESSy (1°x1°) with an explicit trace gas uptake at stomata and cuticle (Emmerichs et
al., 2021) for two different seasons in 2005 and 2006. Removing the canopy reduction factor in
the model leads to a HC increase of soil NO emissions highest over tropical forests (Figure 26).
The temporal variation follows the vegetational growth as in the Northern Hemisphere summer
50% higher emissions occur. These findings are reasonable as Hudman et al. (2012) estimated
that the canopy reduction scheme overall lowers the NO emissions by 10-15% in grasslands and
up to 85% over forests (GEOS-Chem at 2°×2.5° in 2006). Consequently, improper accounting
for the canopy reduction factor may imply a strong underestimation of the soil-N in densely
forested regions and globally by about 31% (2005-2006).

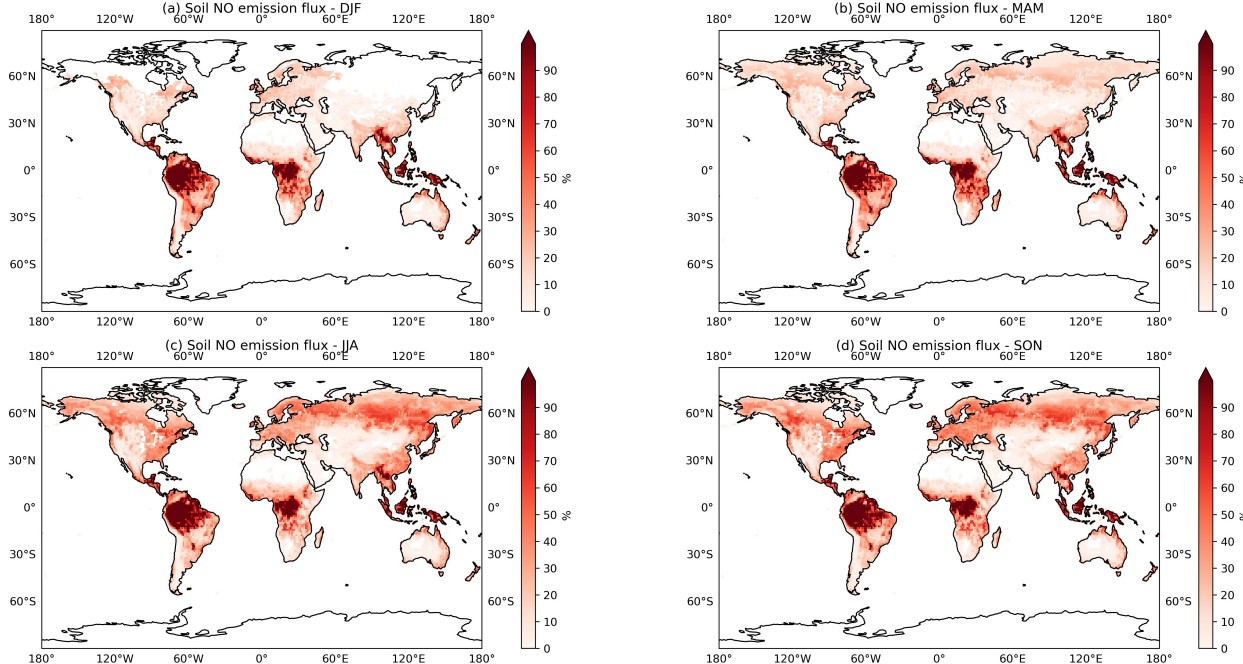

Figure 26: Relative difference Canopy Reduction soil HONO

### 3.6.4.    Projections of soil NO and variability in different climates

The future land use is predicted to change as a consequence of the growing demand for
nutrition and biofuels which implies an increasing use of fertilizer. Consequently, NO soil
emissions are estimated to rise by ~28% during the century to 11.5 TgN/yr at the end of 2100
(Fowler et al., 2015). Similarly, Liu et al. (2021) estimate an increasing soil NO emission of 8.9
TgN/yr by the year 2050 due to intensive nitrification processes.

An increase of LAI by 10 %, in contrast, would lead to 1% lower emissions. In addition, several
responses are expected from the changing climate. In fact, the 1°C higher temperature would
cause ~5% increase in emissions (Weng et al., 2020). Following the future ('medium high')
climate scenario RCP6.0 (Representative Concentration Pathway, 6 W/m2 radiative forcing until
2500, stabilization after 2150) used for the CMIP5 (Climate Model Intercomparison Project)
simulations. Jöckel et al. (2016) suggest an increase of ~15 % in soil NO emissions due to
increasing soil temperature (an increase of soil microbes) from present-day (2010) until 2100.
However, the most significant implications for large-scale denitrification activity are changing
rainfall and the regional hydrological cycles (Fowler et al., 2015). In general, soil $NO_x$ will play
a more important role in the global budget in the troposphere due to the decreasing
anthropogenic emissions in the future. Therefore, increasing NOx-soil emissions may slow down
the decrease of O3 in response to declining anthropogenic emissions (Wu et al. 2022).

### 3.6.5.    Next steps with biogeochemical models implemented in ESMs

Uncertainties of modeling soil nitrogen emissions are associated with the model input and
parameters (Wang and Chen 2012). Process-based biogeochemical models which also consider
the complexity of soil emission processes as DNDC (Denitrification–Decomposition) are needed
(Li et al., 2011). The capability to represent interactive biogeochemical cycles allows for

instance for the online calculation of crop nutrition from soil. Also, a model like CLM5
distinguishes between natural and agricultural soils which more accurately predicts the fertilizer
usage (Fung et al., 2022). Resolving the soil and litter biogeochemical dynamics vertically, in
addition, lead to a more efficient retainment and recycling of N by the ecosystem (Koven et al.,
2013). However, these models should be calibrated to multiple sites (Wang et al., 2019) which is
limited by the availability of measurement data, especially when it comes to global modeling.

### 4. Conclusion

In this article, we investigate temporal and spatial trends and variability of tropospheric ozone in
relation to its precursors using satellite products, ozonesonde measurements, and model
simulations. Our results show that ozone has positive trends at all latitudes and column depths
regardless of the tropopause height within $\pm100$ hPa. The positive trends in the 30-60ºN band are
due to increasing trends over Canada and Alaska and are slightly offset by the small negative
trends over the northeastern US and Europe. The lower trends in the bands 30-60ºN and 30-60ºS
are due to the offsetting impact of negative trends over Eastern US and Europe in the north, and
Australia and South Africa in the south, respectively. The decreasing trends of TrC-$O_3$ over parts
of the northeastern US and Europe are likely due to the decreasing trend of TrC-$NO_2$, which is
due to the effective measures applied over the last two decades to mitigate air pollution in these
regions. TrC-HCHO trends are decreasing in the Eastern US, some parts of northern and western
Africa, and western and northern Europe, and increasing in South Asia, central Africa, northern
Australia, and Brazil. TrC-HCHO trends are consistent with that of TrC-$O_3$ over northeastern US
and Europe. Simulated $O_3$ and its precursors are in good agreement with satellite measurements.
Considering different latitude bands, the TrC-$O_3$ highest trends are simulated between 30º S and
60º N, consistent with calculated trends based on satellite observations. The middle and upper
troposphere make the largest contributions to the simulated TrC-$O_3$ trend globally, with large
contributions from the upper troposphere driving the simulated TrC-$O_3$ trend at 30°S-30°N and
counteracting the negative TrC-$O_3$ trend in the southern midlatitudes.
We have also shed light on $NO_X$ lightning and its relation to ozone trends. $LNO_x$ is responsible
for the largest fraction of upper tropospheric ozone in the tropics year-round and in the mid-
latitudes in summer. Ozone Radiative forcing is due to the ozone near the tropopause. An
increasing trend of $LNO_x$ at Arctic latitudes has been noted, as that region rapidly warms.
However, future trends in flash rate are uncertain, with conflicting predictions coming from
models with differing flash rate parameterizations. Soil HONO emissions had their highest
positive monthly anomalies mainly in the 5 most recent years which is likely due to the more
frequent heat wave occurrence, e.g. in Europe and North America. Soil HONO trends are highest
in Africa accounting for ~30% of the global anomaly. Soil $NO_x$ emissions could play an
important role in the tropospheric $NO_x$ global budget due to the decreasing anthropogenic
emissions in the future. Therefore, the expected increase in $NO_x$-soil emissions may slow down
the decrease of $O_3$ in response to declining anthropogenic emissions. Overall, this study
presented a comprehensive overview of tropospheric ozone trends in relation to its precursors in
different spatial and temporal scales.
Competing interests: At least one of the (co-)authors is a member of the editorial board of
Atmospheric Chemistry and Physics
Author contribution: YE led the conceptualization, writing, and review of the article, JZ led the
OMI ozone satellite product and data analysis, SS led the GEOS 5 GMI data analysis, HP led the

sections on HCHO, $NO_2$, $HCHO/NO_2$ data analysis and contributed to the CO analyses, KM led the comparison of different satellite products, KP lead the lightning NOx section, HW and RB contributed to the CO analysis, DT and TE led the section on HONO soil emission, all authors contribute to the writing and review of the article.

**Acknowledgment**

This study was partially funded by the NSF AGS, grant number 1900795, USF Creative Scholarship Grant 2022. A part of the research was conducted at the Jet Propulsion Laboratory, California Institute of Technology, under a contract with NASA. HP has received funding from the Ministerio de Ciencia e Innovación through the MITIGATE project (grant no. PID2020-113840RA-I00 funded by MCIN/AEI/10.13039/501100011033) and the Ramon y Cajal grant (RYC2021-034511-I, MCIN / AEI / 10.13039/501100011033 and European Union NextGenerationEU/PRTR). The GEOS-GMI simulation was supported by the NASA's Making Earth System Data Records for Use in Research Environments (MEaSUREs) program and the high-performance computing resources for GEOS-GMI were provided by the NASA Center for Climate Simulation (NCCS). A part of the research was supported by the NSF National Center for Atmospheric Research, which is a major facility sponsored by the U.S. National Science Foundation under Cooperative Agreement No. 1852977. We acknowledge the support of the National Aeronautics and Space Administration (NASA) Atmospheric Composition: Aura Science Team Program (19-AURAST19-0044), Atmospheric Composition Modeling and Analysis Program (22-ACMAP22-0013), NASA Earth Science U.S. Participating Investigator program (22-EUSPI22-0005).

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
