# Peer review of "Tropospheric Ozone Precursors: Global and Regional Distributions, Trends, and Variability"

_EGUsphere, 2024_

## Community Comment (CC1)

Comments by Owen R. Cooper (TOAR Scientific Coordinator of the Community Special Issue) on:

**Tropospheric Ozone Precursors: Global and Regional Distributions, Trends and Variability**

Yasin Elshorbany, Jerald Ziemke, Sarah Strode, Hervé Petetin, Kazuyuki Miyazaki, Isabelle De Smedt, Kenneth Pickering, Rodrigo Seguel, Helen Worden, Tamara Emmerichs, Domenico Taraborrelli, Maria Cazorla, Suvarna Fadnavis, Rebecca Buchholz, Benjamin Gaubert, Néstor Rojas, Thiago Nogueira, Thérèse Salameh, and Min Huang

EGUsphere [preprint], https://doi.org/10.5194/egusphere-2024-720, 2024.
Discussion started: 21 March 2024;  Discussion closes 2 May, 2024

This review is by Owen Cooper, TOAR Scientific Coordinator of the TOAR-II Community Special Issue. I, or a member of the TOAR-II Steering Committee, will post comments on all papers submitted to the TOAR-II Community Special Issue, which is an inter-journal special issue accommodating submissions to six Copernicus journals:  ACP (lead journal), AMT, GMD, ESSD, ASCMO and BG. The primary purpose of these reviews is to identify any discrepancies across the TOAR-II submissions, and to allow the author teams time to address the discrepancies.  Additional comments may be included with the reviews. While O. Cooper and members of the TOAR Steering Committee may post open comments on papers submitted to the TOAR-II Community Special Issue, they are not involved with the decision to accept or reject a paper for publication, which is entirely handled by the journal's editorial team.

**General Comments:**

TOAR-II has produced two guidance documents to help authors develop their manuscripts so that results can be consistently compared across the wide range of studies that will be written for the TOAR-II Community Special Issue.  Both guidance documents can be found on the TOAR-II webpage: https://igacproject.org/activities/TOAR/TOAR-II

*The TOAR-II Community Special Issue Guidelines*:  In the spirit of collaboration and to allow TOAR-II findings to be directly comparable across publications, the TOAR-II Steering Committee has issued this set of guidelines regarding style, units, plotting scales, regional and tropospheric column comparisons, tropopause definitions and best statistical practices.

*Guidance note on best statistical for TOAR analyses*:  The aim of this guidance note is to provide recommendations on best statistical practices and to ensure consistent communication of statistical analysis and associated uncertainty across TOAR publications. The scope includes approaches for reporting trends, a discussion of strengths and weaknesses of commonly used techniques, and calibrated language for the communication of uncertainty. Table 3 of the TOAR-II statistical guidelines provides calibrated language for describing trends and uncertainty, similar to the approach of IPCC, which allows trends to be discussed without having to use the problematic expression, "statistically significant".

Recommendation:  **Major revision**

**Major Comments:**

1) As stated on the first line of the abstract, methane is an important ozone precursor, but this paper does not address the distribution and trends of methane. Why has methane been omitted? Methane should be addressed as studies have shown its impact on recent ozone increases (Zhang et al., 2016), and as shown in Chapter 6 of IPCC AR6 (Szopa et al., 2021), the only future scenario with an increasing tropospheric ozone burden is SSP3-7.0, which is driven by increasing methane. NOAA GML observations of methane ([https://gml.noaa.gov/ccgg/trends_ch4/](https://gml.noaa.gov/ccgg/trends_ch4/) ) show that methane concentrations in the atmosphere have increased sharply since 2005 (an 8% increase from 2005 to 2023).

2) Lines 66-68
When summarizing global tropospheric ozone trends, the best reference is Section 2.2.5.3 in Chapter 2 of IPCC AR6 (Gulev et al., 2021).  While observations in the Southern Hemisphere are limited compared to the northern hemisphere, the available in situ and satellite observations do indicate an increase of ozone since the late 20$^{th}$ century:  "Observations in the SH are limited, but indicate average tropospheric column ozone increases of 2–12% (1–5 ppbv) per decade in the tropics (Figure 2.8c), and weak tropospheric column ozone increases (<5%, <1 ppbv per decade) at mid-latitudes (Cooper et al., 2020). Above Antarctica, mid-tropospheric ozone has increased since the late 20$^{th}$ century (Oltmans et al., 2013)."

3) As stated in Section 5 of the 'Guidance note on best statistical for TOAR analyses':
"One of the most critical components of statistical analysis is to acknowledge the uncertainty. Every estimation must be accompanied by a quantification of the associated uncertainty (or error bar), which is used to assess the reliability of the (trend) estimate and is considered to be as equally important as the estimate". According to the guidance note, all trends need to be reported with the 95% confidence intervals and p-values.  Basically, a trend value without an uncertainty estimate is meaningless.  For example, on lines 463-466 model trends are compared to OMI/MLS trends. But because the model trends have no uncertainty range, the trend value is meaningless and no conclusions can be drawn from this comparison.

4) Another important piece of advice from the 'Guidance note on best statistical for TOAR analyses' is that all TOAR analyses should abandon the use of the phases "statistically insignificant" or "statistically significant".  Compelling arguments for this policy are provided by the highly influential paper by Wasserstein et al., 2019.  The submitted paper has many instances of the phrases "statistically insignificant" or "statistically significant".  These phrases need to be removed, and they can be replaced by statements from the authors regarding their confidence in the reported trend values. Advice is given in the Guidance Note, and this advice can also be applied to figures such as Figure 13.

5) Section 3.5
I found the section on LNOx to be too long and it lacks clear statements on lightning trends.  While the section cited previous work that thunderstorm days have increased in some regions (and decreased in others), no number were given, so it's not clear by how much thunderstorm days have increased. In terms of flash rate, some regions showed increases and some showed decreases, but there was no summary statement that lets the reader know if lightning has clearly increased or decreased on the global scale. Line 714 states that lightning contributes to positive ozone radiative forcing, but it's not clear to me that this is really the case. In the UT ozone has a strong longwave radiative effect (i.e. it absorbs outgoing longwave radiation) and of course LNOx can affect ozone in the UT and therefore affect ozone's longwave radiative effect.  But are there any studies that have shown that LNOx impacts ozone's radiative forcing (as opposed to ozone's longwave radiative effect)?  IPCC defines radiative forcing as the change in the Earth's radiative balance since 1750.  If lightning is impacting radiative

forcing then there must be conclusive evidence that lightning frequency has increased on the global scale.  If there is no clear evidence for a global increase (or decrease) of lightning then the link to radiative forcing cannot be established.

6) lines 391-393
Regarding the number of ozone profiles required to accurately detect a trend, several studies over the years have shown that once-per-week sampling is often inadequate for accurate trend detection. A paper recently accepted for publication in the TOAR-II Community Special Issue (Chang et al., 2024) addresses this issue, and the paper's conclusions need to be considered when interpreting ozone trends based on sparse sampling.

7) Line 362
Here it is claimed that the pandemic period led to increases in emissions and therefore an increase in the ozone rate of change, but no convincing references are provided to support this claim. The paper by Oleribe et al. 2021 has nothing to do with atmospheric chemistry, and the paper by Matandirotya et al., 2023 only looks at 3 cities in South Africa. This statement seems like speculation and it should be removed.

8) Section 3.4.2
This section seems to only review ozone trends from previous studies by Wang et al., 2022 and by Christiansen et al., 2022.  Does this paper actually calculate updated trends from available ozonesonde records? Section 2.2.2. in the Methods section lists ozonesondes as a data source, but I see no new data analysis.

**Minor Comments:**

line 66
The stated radiative forcing for ozone (0.34) is incorrect. As reported in Section 7.3.2.5 of Chapter 7 of IPCC AR6 (Forster et al., 2021), ozone has an assessed effective radiative forcing of 0.47 [0.24 to 0.70] W m–2.

Line 76
More context needs to be given regarding the ozone increase of 40 ppb. Is this at the surface or in the free troposphere? Over which continent?  If stating the extreme ozone increase, it would help to also provide the average ozone increase. A useful number is the approximate 45% increase in the tropospheric ozone burden.

A recent paper published in ACP (Nussbaumer et al., 2023) is highly relevant to this submission and some discussion of their conclusions is warranted.

Line 350
When discussing the impact of COVID-19 on tropospheric ozone there are some key papers that should be cited:  Steinbrecht et al., 2021; Chang et al., 2022; Putero et al., 2023

Line 400
Very strong ozone trends above Japan since 2010 was not a major conclusion of Christiansen et al. (2022). They only show the higher ozone values after 2010 in the supplement, and they recommend that these data sets be treated with caution because the instruments changed from carbon-iodide to ECC after 2010; these time series have not been homogenized to correct for the change in instruments.

**Figure 6**
The color table is not aligned correctly with the ozone trend values.  Weak positive trends of 0.0 to 0.5 DU per decade are shaded light blue instead of light orange. This makes negative trends appear to be far more prominent in the maps than they truly are. Please correct the color table.

**Line 326**
Are the small changes in tropopause height enough to have an impact on ozone trends?  According to Figure 3 the tropopause height at northern mid-latitudes has increased at a rate of about 30 m per decade. Over the period 2004-2021 (the OMI/MLS record), this equals a total increase of the tropopause height of 51 meters. Assuming a typical tropopause height of 12,000 m, this means that the depth of the troposphere has increased by less than one half of one percent.  I'm skeptical that this very small change has a meaningful impact on the ozone trend over such a short period.

**Line 343**
Here it is claimed that positive ozone trends are driven by oceanic emissions. Oceanic emissions of what? Please provide references that demonstrate that this is a known and valid explanation.

**Figure 14**
Please correct "United Utates"
Also, please remove the trend lines that extend beyond the bounds of the SCIAMACHY record.

**Lines 355-357**
"The trends during the time period (2005-2021) show a significant decline in O3 column trends in the northern hemisphere but a slightly increasing trend in the southern hemisphere."
This sentence is not phrased well. I think you are trying to make the point that when the years 2020-2021 are added to 2005-2019, then there is a change in the ozone rate of change. Please rephrase so that this point is clear.

**References:**

Chang, K.-L., et al. (2022), Impact of the COVID-19 economic downturn on tropospheric ozone trends: an uncertainty weighted data synthesis for quantifying regional anomalies above western North America and Europe, *AGU Advances, 3*, e2021AV000542. https://doi.org/10.1029/2021AV000542

Chang, K.-L., Cooper, O. R., Gaudel, A., Petropavlovskikh, I., Effertz, P., Morris, G., and McDonald, B. C.: Technical note: Challenges of detecting free tropospheric ozone trends in a sparsely sampled environment, EGUsphere [preprint], https://doi.org/10.5194/egusphere-2023-2739, 2024.

Forster, P., T. Storelvmo, K. Armour, W. Collins, J.-L. Dufresne, D. Frame, D.J. Lunt, T. Mauritsen, M.D. Palmer, M. Watanabe, M. Wild, and H. Zhang, 2021: The Earth's Energy Budget, Climate Feedbacks, and Climate Sensitivity. In Climate Change 2021: The Physical Science Basis. Contribution of Working Group I to the Sixth Assessment Report of the Intergovernmental Panel on Climate Change [Masson-Delmotte, V., P. Zhai, A. Pirani, S.L. Connors, C. Péan, S. Berger, N. Caud, Y. Chen, L. Goldfarb, M.I. Gomis, M. Huang, K. Leitzell, E. Lonnoy, J.B.R. Matthews, T.K. Maycock, T. Waterfield, O. Yelekçi, R. Yu, and B. Zhou (eds.)]. Cambridge University Press, Cambridge, United Kingdom and New York, NY, USA, pp. 923–1054, doi:10.1017/9781009157896.009.

Gulev, S.K., P.W. Thorne, J. Ahn, F.J. Dentener, C.M. Domingues, S. Gerland, D. Gong, D.S. Kaufman, H.C. Nnamchi, J. Quaas, J.A. Rivera, S. Sathyendranath, S.L. Smith, B. Trewin, K. von Schuckmann, and R.S. Vose, 2021: Changing State of the Climate System. In Climate Change 2021: The Physical Science Basis. Contribution of Working Group I to the Sixth Assessment Report of the Intergovernmental Panel on

Climate Change [Masson-Delmotte, V., P. Zhai, A. Pirani, S.L. Connors, C. Péan, S. Berger, N. Caud, Y. Chen, L. Goldfarb, M.I. Gomis, M. Huang, K. Leitzell, E. Lonnoy, J.B.R. Matthews, T.K. Maycock, T. Waterfield, O. Yelekçi, R. Yu, and B. Zhou (eds.)]. Cambridge University Press, Cambridge, United Kingdom and New York, NY, USA, pp. 287–422, doi:10.1017/9781009157896.004

Nussbaumer, C. M., Fischer, H., Lelieveld, J., and Pozzer, A.: What controls ozone sensitivity in the upper tropical troposphere?, Atmos. Chem. Phys., 23, 12651–12669, https://doi.org/10.5194/acp-23-12651-2023, 2023.

Putero, D., et al. (2023), Fingerprints of the COVID-19 economic downturn and recovery on ozone anomalies at high-elevation sites in North America and western Europe, Atmos. Chem. Phys., 23, 15693–15709, https://doi.org/10.5194/acp-23-15693-2023

Steinbrecht, wt al. (2021), COVID-19 Crisis Reduces Free Tropospheric Ozone Across the Northern Hemisphere, *Geophysical Research Letters, 48*, e2020GL091987. https://doi.org/10.1029/2020GL091987

Szopa, S., V. Naik, B. Adhikary, P. Artaxo, T. Berntsen, W.D. Collins, S. Fuzzi, L. Gallardo, A. Kiendler-Scharr, Z. Klimont, H. Liao, N. Unger, and P. Zanis, 2021: Short-Lived Climate Forcers. In Climate Change 2021: The Physical Science Basis. Contribution of Working Group I to the Sixth Assessment Report of the Intergovernmental Panel on Climate Change [Masson-Delmotte, V., P. Zhai, A. Pirani, S.L. Connors, C. Péan, S. Berger, N. Caud, Y. Chen, L. Goldfarb, M.I. Gomis, M. Huang, K. Leitzell, E. Lonnoy, J.B.R. Matthews, T.K. Maycock, T. Waterfield, O. Yelekçi, R. Yu, and B. Zhou (eds.)]. Cambridge University Press, Cambridge, United Kingdom and New York, NY, USA, pp. 817–922, doi:10.1017/9781009157896.008

Wang, H., et al. (2022), Global tropospheric ozone trends, attributions, and radiative impacts in 1995–2017: an integrated analysis using aircraft (IAGOS) observations, ozonesonde, and multi-decadal chemical model simulations, Atmos. Chem. Phys., 22, 13753–13782, https://doi.org/10.5194/acp-22-13753-2022

Wasserstein, R.L., Schirm, A.L. and Lazar, N.A., 2019. Moving to a world beyond "p< 0.05". The American Statistician, 73(sup1), pp.1-19 https://www.tandfonline.com/doi/pdf/10.1080/00031305.2019.1583913

Zanis et al., 2022, Climate change penalty and benefit on surface ozone: a global perspective based on CMIP6 Earth System Models, Environmental Research Letters, 17

Zhang, Y., et al. (2016), Tropospheric ozone change from 1980 to 2010 dominated by equatorward redistribution of emissions, *Nature Geoscience*, 9(12), p.875, doi: 10.1038/NGEO2827

---

## Author Comment (AC1)

**RC1**: ['Comment on egusphere-2024-720'](), Anonymous Referee #1, 18 Apr 2024, **Citation**:

https://doi.org/10.5194/egusphere-2024-720-RC1

This is a very detailed paper using a large amount of satellite, ozonesonde and modelling data to investigate the precursor drivers to the trends in total column tropospheric ozone between 2005-2019 on regional and global scales. The use of $NO_2$, VOC, HCHO tropospheric column data is novel for this type of analysis, is within the scope of ACP and should be published subject to a slight restructure and few minor corrections.

We thank the reviewer for his comments and recommendations. Our responses are in blue following each comment.

**General comments**

It is unclear to me the relevance of the model analysis for different parts of the troposphere (Figure 5) if it isn't referred to in the later interpretation of the measurements or in the conclusion.

The simulated tropospheric ozone and its precursors in the lower, middle and higher troposphere are now mentioned in the discussion (e.g., secs. 3.3, and 3.43) and in the conclusion.

There is some clear parallel in the model analysis where the contribution and trends are separated by region (Figure 11) but can this be brought into the discussion more and perhaps section 3,3. (and Figure 5,11) be moved to after the measurements are presented (3.4.4, 3.4.5, 3.4.6, 3.4.7 ). A summary could then also include a discussion about which parts of the column are driving the trends in different regions of the globe, otherwise why include the model information?

We thank the reviewer for this comment. We have added this discussion to the conclusions as suggested. Figures 5 show the simulated mean partial column of tropospheric O3, CO, NO2, and HCHO and we believe it is suitable here in section 3.3 which addresses the model simulation. Figure 11 is located in section 3.4.3 for regional trends and likewise is suitable there since it addresses trends for several regions and latitudinal bands. We have included now a discussion in the conclusion regarding which parts of the columns are driving the trends.

Are the model trends inline with the measurements?  Please include some reference to the model findings in the precursor measurement discussions and conclusions.

Comparisons to measurements are already in the text, for example, in section 3.4.3, as stated "Simulated TrC-O$_3$ trends are also quite comparable to those observed by OMI/MLS within the measurement model uncertainty (see **Error! Reference source not found.** and **Error! Reference source not found.**). Over Australia, the OMI/MLS trend of 1.05 DU/decade is higher than the model trend of about 0.2 DU/decade (see **Error! Reference source not found.**). However, since OMI/MLS trend has a calculated uncertainty (2σ) of 1.44 DU/decade, both the model and OMI/MLS for Australia are not statistically different."

We have also added related discussion to the conclusion.

The figures are a little confusing.  It would make sense to use similar figures for each of the ozone precursors to allow easier interpretation.  The discussion on CO trends includes more plots (including anomaly trend plots) for which the other precursor species there is a summary (also nice but maybe don't need both).  Can the plots used be the same for all species in sections 3.4.4, 3.4.5, 3.4.6, 3.4.7,  either anomaly time-series or summary of anomalies?

We have added new CO figures for the trend summaries, similar to NO2 and HCHO as the reviewer suggested.

As stated above Figure 5 perhaps relates more to the O$_3$ discussion and should sit with Figure 11 as they are related (contribution to each part of the column from model and actual trend).  Also perhaps Figure 4 should sit later with Figure 12 (see later specific points).

As we mentioned earlier, Figure 5 presents the burdens of each precursor in while Figure 11 presents the trends for each region. We believe that Figure 5 sits in the correct place (sec. 3.3) following figure 4 that shows the burdens of all precursors globally. Figure 11 shows the simulated trends sitting in section 3.4 (tropospheric trends)

I think it is just a misunderstanding on my part but there should be a consistency with the time period that you are discussing, for the most part the time period 2005-2019 is shown but the main $O_3$ trend figure 6 is possibly until 2021. There is also some reference to COVID years and this seems a bit out of place in the paper unless related to the above point?

All figures are from 2005 to 2019 except for Figure 8b (2005-2021) which is shown only to demonstrate the effect of COVID-19. Figure 6 is also from 2005-2019 as shown on the legends on top of each panel. However, there was a typo in the caption that says 2004-2021, which is corrected now. We thank the reviewer for mentioning that typo.

**Specific comments**

Figure 2: Please include time range (2005-2019) in the figure caption.

Done!

Line 263:  Use 'peroxyl' not 'proxy'.

Done!

Line 272: Self referencing, can you choose other references for these fundamental reactions?

Done!

Lines 281-282 and Figure 4: $NO_2$ column is stated to be decreasing in North America, Europe and Australia.  Please include a reference for this relating to air pollution controls or alternatively Figure 4 should show the trend of $NO_2$ rather than the mean if you are referring to it? The trends are shown later in Figure 12 so discussion about them should be there unless you have another reference for it?

Done! A reference has been added.

Section 3.2, lines 290-295 generally needs more references when discussing sources from certain regions, not just 'e.g.' in brackets.

Done!

Figure 5: Model shows lower, middle, upper differences in precursor contribution.  At this point the reader is interested in the trends within the column and whether any specific part of the troposphere has been increasing/decreasing over this time and in which regions, actually shown later in Figure 11.  Can these be put together, perhaps in a later discussion?

Done!

Figure 6: What timescale are the trends over, 2005-2019 or 2005-2021?

Done! It is 2005-2019, the caption is corrected.

Figure 7: Does the right plot show the 'zonal mean trend' of column depths or should the caption say 'mean $O_3$' by latitude for different column depths for 2005-2019?

Done! The caption of the right panel has been further clarified "zonal mean of different column depths (right) from 2005-2019"

Figure 9: Can latitude of the observations be included on this plot?

Done! Table S 1 includes a list of the coordinates of ozonesonde stations used in the study. This is now mentioned in the caption as well.

Line 431: What is a partial column, is it defined?

The partial columns are labeled on Figure 11 as (lower, middle, and upper troposphere) in addition to the full troposphere. They are also defined in section 3.3: "lower (up to 700hPa), middle (700-400hPa), and upper (400hPa to tropopause) portions of the troposphere". We have further defined them in the text in section 3.4.3.

Line 547: Include reference for biomass burning activity

Done!

Figure 16: This is a nice figure showing average CO anomalies in column mean 2005-2021 but with trends only until 2019 which is good. Can we have similar plots for $NO_2$?

The trends are calculated only until 2019 so that it is consistent with other species and model simulations. As per the reviewer's earlier suggestion, we have included additional figures for Trend summaries similar to that of NO2 and HCHO.

Figure 18, Summary of trends, for HCHO, why don't we have this type of plot for CO?

As per the reviewer's earlier suggestion, we have included additional figures for Trend summaries similar to that of NO2 and HCHO. Figure 24: Monthly anomalies of HONO from soil by region, please include trend data on these.

Done. The figure is now updated with the trend values and their uncertainties.

---

## Author Comment (AC2)

**Community Comments, CC1: ['Comment on egusphere-2024-720'](), Owen Cooper, 02 May 2024**

**Major Comments:**

1) As stated on the first line of the abstract, methane is an important ozone precursor, but this paper does not address the distribution and trends of methane. Why has methane been omitted? Methane should be addressed as studies have shown its impact on recent ozone increases (Zhang et al., 2016), and as shown in Chapter 6 of IPCC AR6 (Szopa et al., 2021), the only future scenario with an increasing tropospheric ozone burden is SSP3-7.0, which is driven by increasing methane. NOAA GML observations of methane (https://gml.noaa.gov/ccgg/trends_ch4/ ) show that methane concentrations in the atmosphere have increased sharply since 2005 (an 8% increase from 2005 to 2023).

Answer: While oxidation of methane and NMHCs was mentioned in the introductory statement in the article, it was not investigated in this study since we focused in this article more on the reactive species, NO2, CH2O, and CO. Methane is an important precursor and we also think future assessments should focus on its analysis to include methane and other precursors.

We have included the following paragraph in section 3.2 to address methane contribution to tropospheric ozone: While this paper focuses on ozone precursors with higher reactivity, we note that methane, with an assessed total atmospheric lifetime of $9.1 \pm 0.9$ years (Szopa et al., 2021), is also a crucial driver (Fiore et al., 2002; Isaksen et al., 2014), given its accelerated growing rate of $7.6 \pm 2.7$ nmol mol$^{-1}$ yr$^{-1}$ between 2010 and 2019 (Canadell et al., 2021), largely driven by anthropogenic activities (Szopa et al., 2021).

2) Lines 66-68 When summarizing global tropospheric ozone trends, the best reference is Section 2.2.5.3 in Chapter 2 of IPCC AR6 (Gulev et al., 2021). While observations in the Southern Hemisphere are limited compared to the northern hemisphere, the available in situ and satellite observations do indicate an increase of ozone since the late 20th century: "Observations in the SH are limited, but indicate average tropospheric column ozone increases of 2–12% (1–5 ppbv) per decade in the tropics (Figure 2.8c), and weak tropospheric column ozone increases (<5%, <1 ppbv per decade) at mid-latitudes (Cooper et al., 2020). Above Antarctica, mid-tropospheric ozone has increased since the late 20th century (Oltmans et al., 2013)."

Answer: We thank Owen for this comment and we have updated this sentence as follows:

"…..radiative forcing of $(0.47^{+0.23}_{-0.23})$ W m$^{-2}$; Forster et al., 2021). Since the mid-1990s, free tropospheric ozone trends based on in situ measurement and satellite retrievals have increased with high confidence by 1-4 nmol mol$^{-1}$ decade$^{-1}$ across the northern mid-latitudes and 1-5 nmol mol$^{-1}$ decade$^{-1}$ within the tropics (Guleb et al., 2021). In the Southern Hemisphere, with more limited observation coverage compared with the Northern Hemisphere, the tropospheric column ozone shows an increase since the mid-1990s by less than 1 nmol mol$^{-1}$ decade$^{-1}$ with *medium confidence* at southern mid-latitudes (Gulev et al., 2021, Cooper at al., 2020). Tropospheric O$_3$ short- and long-term..".

3) As stated in Section 5 of the 'Guidance note on best statistical for TOAR analyses': "One of the most critical components of statistical analysis is to acknowledge the uncertainty. Every estimation must be accompanied by a quantification of the associated uncertainty (or error bar), which is used to assess the reliability of the (trend) estimate and is considered to be as equally important as the estimate". According to the guidance note, all trends need to be reported with the 95% confidence intervals and p-values. Basically, a trend value without an uncertainty

estimate is meaningless.  For example, on lines 463-466 model trends are compared to OMI/MLS trends. But because the model trends have no uncertainty range, the trend value is meaningless and no conclusions can be drawn from this comparison.

Answer: We added error bars to the model trends in Figure 11 and clarified the model trend discussion.  All trends are now shown with their corresponding error.

4) Another important piece of advice from the 'Guidance note on best statistical for TOAR analyses' is that all TOAR analyses should abandon the use of the phases "statistically insignificant" or "statistically significant".  Compelling arguments for this policy are provided by the highly influential paper by Wasserstein et al., 2019.  The submitted paper has many instances of the phrases "statistically insignificant" or "statistically significant".  These phrases need to be removed, and they can be replaced by statements from the authors regarding their confidence in the reported trend values. Advice is given in the Guidance Note, and this advice can also be applied to figures such as Figure 13.

Answer: We appreciate the advice, and we will certainly consider in future submissions.

5) Section 3.5 I found the section on LNOx to be too long and it lacks clear statements on lightning trends.  While the section cited previous work that thunderstorm days have increased in some regions (and decreased in others), no number were given, so it's not clear by how much thunderstorm days have increased. In terms of flash rate, some regions showed increases and some showed decreases, but there was no summary statement that lets the reader know if lightning has clearly increased or decreased on the global scale. Line 714 states that lightning contributes to positive ozone radiative forcing, but it's not clear to me that this is really the case. In the UT ozone has a strong longwave radiative effect (i.e. it absorbs outgoing longwave radiation) and of course LNOx can affect ozone in the UT and therefore affect ozone's longwave

radiative effect.  But are there any studies that have shown that LNOx impacts ozone's radiative forcing (as opposed to ozone's longwave radiative effect)?  IPCC defines radiative forcing as the change in the Earth's radiative balance since 1750.  If lightning is impacting radiative forcing then there must be conclusive evidence that lightning frequency has increased on the global scale.  If there is no clear evidence for a global increase (or decrease) of lightning then the link to radiative forcing cannot be established.

Answer:  Section 3.5 has been shortened and a table has been removed.  Section 3.5.1 now has a rough estimate of the global trend in thunder days, and makes it clear that no long-term trend in global flash rates has been detected.  All mentions of "radiative forcing" have been removed, and instead, the terminology, "effects by ozone on longwave radiation absorption" is now used.

6) lines 391-393 Regarding the number of ozone profiles required to accurately detect a trend, several studies over the years have shown that once-per-week sampling is often inadequate for accurate trend detection. A paper recently accepted for publication in the TOAR-II Community Special Issue (Chang et al., 2024) addresses this issue, and the paper's conclusions need to be considered when interpreting ozone trends based on sparse sampling.

Answer:  We acknowledge this limitation, and we further clarified this aspect in the text including two references (Chang et al., 2020 and Chang et al., 2024). However, as mentioned by Chang et al (2024) only 3 European stations have achieved such high sampling frequency (Hohenpeissenberg, Germany; Payerne, Switzerland, and Uccle, Belgium). In the meantime, despite the lower sampling frequency at the rest of the global stations, ozonesonde observations continue to be the gold standard against which satellite data are validated. Furthermore, ozonesonde climatologies are critical to provide feedback to satellite observations for which they continue to be used with this long-term purpose. Thus, with proper acknowledgment of the

frequency limitation, we believe that ozonesonde trends published in previously peer-reviewed studies are valuable information for the ozone community.

7) Line 362 Here it is claimed that the pandemic period led to increases in emissions and therefore an increase in the ozone rate of change, but no convincing references are provided to support this claim. The paper by Oleribe et al. 2021 has nothing to do with atmospheric chemistry, and the paper by Matandirotya et al., 2023 only looks at 3 cities in South Africa. This statement seems like speculation and it should be removed.

Answer: It is not mentioned in any part of the article that "the pandemic period led to increases in emissions and therefore an increase in the ozone rate of change"

We mentioned that the higher $O_3$ trends in the southern hemisphere is due to the lesser impact of the pandemic and we cited Oleribe et al. (2021) since they explained "why Sub-Saharan Africa Experienced a less severe COVID-19 Pandemic in 2020". Similarly, Matandirotya et al., 2023 assessed the $NO_2$ atmospheric air pollution over three cities in South Africa during 2020 COVID-19 pandemic, and both articles cover important regions in the southern hemisphere. However, we agree that these references do not provide concrete evidence as to why $O_3$ trends in the southern hemisphere were higher, therefore we remove this sentence as suggested.

8) Section 3.4.2 This section seems to only review ozone trends from previous studies by Wang et al., 2022 and by Christiansen et al., 2022. Does this paper actually calculate updated trends from available ozonesonde records? Section 2.2.2. in the Methods section lists ozonesondes as a data source, but I see no new data analysis.

Answer: Figure 9 was prepared by the authors for this paper, using trends from Wang et al 2022. As pointed out, this portion of the manuscript is based on a review of global ozonesonde trends

calculated and published in previous studies (Wang et al., 2022 and by Christiansen et al., 2022). This is indicated in the caption of Figure 9. However, we added a sentence at the bottom of the corresponding methods section for further clarification.

**Minor Comments:**

line 66 The stated radiative forcing for ozone (0.34) is incorrect. As reported in Section 7.3.2.5 of Chapter 7 of IPCC AR6 (Forster et al., 2021), ozone has an assessed effective radiative forcing of 0.47 [0.24 to 0.70] W m–2.

Answer: The numbers we showed are correct for the global average Radiative Forcing (RF) of $(0.34^{+0.09}_{-0.06}$ W m$^{-2}$; IPCC, 2023) for clear sky conditions based on the *Ramaswamy et al., 2018 (IPCC AR6)*. Page 363 of the IPCC AR6 report states that "For total sky conditions, the range in globally and annual averaged tropospheric O3 forcing from all of these models is from 0.28 to 0.43 Wm−2,… The tropospheric O3 forcing constrained by the observational climatology is 0.32 Wm−2 for globally averaged, total sky conditions."

The numbers in the comment are for Effective Radiative Forcings (ERF), which include all tropospheric and land surface adjustments, particularly aerosol-cloud interactions to aerosol forcing, which is highly uncertain itself (Smith et al., 2020, https://doi.org/10.5194/acp-20-9591-2020). Both numbers are correct but reflect different things and are nevertheless very close (within the uncertainty range). However, since the reference Forster et al., 2021 is newer and include the latest increase of GHG as well as improved cloud parameterization for RF, we will update our number to that reference.

Line 76 More context needs to be given regarding the ozone increase of 40 ppb. Is this at the surface or in the free troposphere? Over which continent?  If stating the extreme ozone increase,

it would help to also provide the average ozone increase. A useful number is the approximate 45% increase in the tropospheric ozone burden. A recent paper published in ACP (Nussbaumer et al., 2023) is highly relevant to this submission and some discussion of their conclusions is warranted.

Answer: The sentence has been updated as follows: CMIP6 models simulate large increasing trends of surface concentrations of $O_3$ and $PM_{2.5}$ in East and South Asia with an annual mean increase of up to 40 ppb and 12 $\mu gm^{-3}$, respectively, over the historical periods (1850-2014; Turnock et al., 2020). The reference is useful, and we have cited it properly.

Line 350 When discussing the impact of COVID-19 on tropospheric ozone there are some key papers that should be cited: Steinbrecht et al., 2021; Chang et al., 2022; Putero et al., 2023

Answer: References are cited now.

Line 400 Very strong ozone trends above Japan since 2010 was not a major conclusion of Christiansen et al. (2022). They only show the higher ozone values after 2010 in the supplement, and they recommend that these data sets be treated with caution because the instruments changed from carbon-iodide to ECC after 2010; these time series have not been homogenized to correct for the change in instruments.

Answer: We updated the sentence to "For example, ozone in East Asia (Japan) has been increasing at a cautious rate of 3.5 to 5 ppbv/decade, particularly since 2010 (Christiansen et al., 2022)."

---

## Author Comment (AC3)

**C2:** ['Comment on egusphere-2024-720'](https://doi.org/10.5194/egusphere-2024-720-RC2)**, Anonymous Referee #2, 22 Apr 2024, Citation**:

https://doi.org/10.5194/egusphere-2024-720-RC2

We thank the reviewer for his comments. Our responses are in blue following each comment.

This manuscript describes global and regional trends of tropospheric ozone and its precursors (NO2, CO, and HCHO) and aims to investigate the spatio-temporal characteristics of the ozone formation regime. Satellite observations of the tropospheric column have been used as the main data source, while ozone sonde data and GEOS-GMI model simulations have also been analyzed to study the sub-columns. The topic is central to the scope of the journal, and the trend analysis from OMI/MLS seems sufficiently valid.

However, particularly for the part analyzing ozone formation regimes, I find the authors' discussion rough and even flawed: my main argument is that without considering 1) the trends of "long-range transported" ozone, 2) the seasonality, or 3) the major components from lower/middle/upper tropospheric sub-columns that drive the trends (although treated in the model for Figure 11), the assessment of the regimes is not correct. For example, in lines 652-653, it is doubtful to conclude that most regions in the southern hemisphere are VOC sensitive regions (lines 365 and 652), simply from the fact that the trends of O3 and HCHO are decreasing while the trend with NO2 is increasing. In the atmospheric chemistry theory, VOC-limited conditions must occur where NOx is abundant (and thus OH loss is controlled by its reaction with NO2), which is unlikely for "most regions in the southern hemisphere".

Answer: We agree with the reviewer that other factors contribute to the burdens of the tropospheric ozone column such as stratosphere-troposphere exchange (STE), regional and longrange transport of ozone precursors, reactivity, and seasonality. We have now included a discussion about possible contributions from STE as it has been recently highlighted as an important contributor to tropospheric ozone column especially in the middle and upper troposphere, and particularly in midlatitudes. We also noted in section 3.4.7 that this analysis does not consider variations of the ratios and their trends with respect to season or altitude. Regarding the chemical regimes, we have clarified that the trends of the HCHO/NO$_2$ ratio indicate moving towards a VOC or NOx sensitivity regime rather than already in a VOC or NOx-sensitive regime.

Another crude statement is made in lines 343-344, that the positive trends in the 30-60degS band are mainly driven by oceanic emissions, without any supporting results. All other parts discussing regimes need to be reviewed and reconsidered.

The statement has been further clarified (… and the positive trends in this band are contributed mainly by oceanic regions (see **Error! Reference source not found.**).).

The discussion on the TrC-HCHO/TrC-NO ratio (section 3.4.7) seems to be the opposite. If the ratio decreases, the chemical status must be becoming more VOC sensitive (rather than NOx sensitive, line 675). The increasing trend should indicate more NOx sensitive conditions (rather than ROx sensitive, line 678).

As explained in section 3.4.7, there are two important aspects to differentiate between, 1) the HCHO/NO ratio, 2) the trend in HCHO/NO2. The mean HCHO/NO2 ratio is shown in Figure 19. We explain in lines 657 – 665 that higher HCHO/NO2 ratio is related to NO sensitive condition as the reviewer pointed out. There is no discrepancy here.

We also added the following sentence to clarify the different interpretations of the HCHO/NO2 trends. "For example, over the eastern US and Europe, the HCHO/NO2 ratio (Figure 19) is low but the HCHO/NO2 trend is showing a slightly increasing trend indicating a direction towards the opposite, NO sensitive conditions.

However, there is a typo mistake in lines 675 and 678 that was made in the final stage of the article and was not detected before the submission. We thank the reviewer for mentioning this typo, which we have corrected.

I also did not understand why the positive emission trends with soil HONO in the southern hemisphere lead to a decrease in O3 (line 1074).

This is a typo and the sentence has been removed.

When presenting the satellite data used in Section 2.2.1, the authors need to describe what kind of data screening was applied, in particular with respect to cloud fraction and solar zenith angle. It is also necessary to describe which emission inventory was used for the GEOS-GMI model simulations has to be described (leading to an erroneous positive CO trend over East Asia).

The Satellite data products used in this article are already published and their references are listed in Table 1 and cited as appropriate in the text. The same applies to the GEOS-GMI setup and simulation results are described. In addition, we include now in section 2.2.3. a discussion and references for the emissions used in GEOS-GMI simulations.

In section 3.2, only the first authors papers are cited. It is needed to provide a more balanced citation.

More references are now cited.

Considering the importance of understanding the chemical regimes and providing valid information for the abatement strategy (line 141), I do not recommend publication of this manuscript in its current form.

The article has been revised including the reviewer's comments.

---

## Referee Report (RR1)

Review of Elshorbany 2024
Tropospheric Ozone Precursors: Global and Regional Distributions, Trends and Variability

The revised paper is much improved, shorter and more evidenced. On the whole the three reviewer's comments have been addressed. I would however recommend that the authors address the below outstanding comments before it can be recommended for publication.

Review 1

Mostly addressed. References are missing in the responses to the reviewer.

Review 2

1) Addressed sufficiently
2) Needs a reference for the response to the comment around ocean emissions/regions. Issues with this statement were also highlighted within Reviewer 3's comments and this needs more substance and discussion in the text.
3) Addressed sufficiently
4) Addressed sufficiently
5) Addressed sufficiently
6) Addressed sufficiently

Review 3

1) Based on the importance of methane to future tropospheric ozone burden highlighted by Cooper's review (in particular with respect to the IPCC AR6 reference); this needs more attention before considering the impact of the reactive species. Discussion of methane distribution and trends and impact on ozone with reference to other studies should be included in the introduction and also the emphasis of the new paragraph should be the other way around. Suggest discussing methane importance first and then move the focus for the paper to higher reactivity precursors as this is then the new work.
2) Addressed sufficiently
3) Addressed sufficiently
4) As this is a paper discussing trends for TOAR-II and contributing to the Community special issue then the guidance should be observed and as advised the authors should try and change the wording they have used from 'significant' to statements about 'confidence'.
5) Addressed sufficiently
6) Addressed sufficiently
7) Addressed sufficiently
8) Addressed sufficiently

Minor comments

Mostly addressed but some detail is missing in the responses, *if* changes have been made- and it doesn't look like it- they are not described in the response. Authors need to respond to the comments around Figure 6, line 326, line 343, Figure 14, lines 355-357.

---

## Author Response (AR2)

**Responses to Report#1**

We thank the reviewer for his comments and recommendations. Our responses are in blue following each comment.

Re-review of Elshorbany et al.

Although several of the previously identified fatal points have been removed, there are still several important issues that remain unaddressed, as follows:

1. Section 2.2.1: I have to repeat that it is fundamentally important to describe satellite data usage criteria, e.g., cloud fraction and data quality flag.

Answer: We have added the data usage criteria for the used satellite products.

2. In Line 784 (ATC-1.pdf), the authors added a statement that variations and trends with respect to season or altitude are not considered. However, I strongly suggest conducting seasonally-separated trend analysis for NO2, CO, HCHO, and the HCHO/NO2 ratio. With data shown in Figure S6, I believe the authors could address this issue. The chemical regime analysis is only meaningful for the ozone production season.

Answer:

We have conducted a separate seasonal and latitudinal analysis of ozone photochemical regimes, including O3 and its precursors, $NO_2$, CO, and HCHO, and will be submitted within two weeks to the same special issue (Fadnavis et al., 2024).

In addition, we now show the seasonal variability of TrC-$O_3$ trends in Figure S7 as also outlined in section 3.4.1.

We think that this will address the reviewers' comments and also avoid confusing the readers with too many analyses.

"

I am also still concerned about the altitude dependence, about the applicability of the HCHO/NO2 COLUMN ratio to infer chemical regime and its change, beyond the strong ozone production region (i.e. non-urban region). The small trends in NO2 and HCHO might be from different altitudes, particularly in the global-scale analysis, about the background atmosphere (lines 796ff). In my opinion, the term "regime" should be only used for the urban or photochemically-active region and the relevant season, where near-ground ozone production is obvious and dominating the column quantities.

Answer:

Ozone regimes can usually be explicitly determined using ozone sensitivity studies (e.g., Fadnavis et al., 2024, this special issue). However, indicators, such as $HCHO/NO_2$ which uses remote sensing products of NO2 and HCHO can still provide valuable insights into the chemical regimes. We mentioned in section 3.4.7 that the HCHO/NO2 is not an optimal indicator of zone sensitivity, which would require a sensitivity analysis "Although imperfect (e.g. Souri et al., 2023), this indicator yet provides some qualitative information on the evolution of the $O_3$ regime over the last years (Nussbaumer et al., 2023). We note that this analysis does not consider variations in the ratios and their trends with respect to season or altitude.". In addition, our results based on HCHO/NO2 are consistent with our ozone sensitivity analysis from our sensitivity paper (e.g., Fadnavis et al., 2024, this special issue) and support our findings.

The word "regime" indicates the prevailing conditions in a certain region(s) and, as the reviewer suggested, we didn't use it on a global scale or to indicate general conditions.

3. When trends in TrC-NO2 or HCHO are analyzed in combination to TrC-O3, not only STE (newly stated in lines 588, 591, and 763 of ATC1.pdf), but also baseline trends as well as increased/decreased long-range transport of ozone are equally or even more important. These points have to be mentioned in the Abstract too. I do not find Li et al., 2024 in the references.

Answer: We thank the reviewer for this comment, and we have now included long-range transport (LRT) throughout the text and in the abstract.

4. Concerns about the analysis over Australia. Why more points appear in Figure 21 (HCHO/NO2 ratio) than Figure 18 (TrC-HCHO) over Australia? What is the NOx concentration levels over the central Australia, showing an increasing trend? What does the "shift toward VOC-sensitive conditions with ozone production" mean over such region?

Answer: This comment is comparing two different quantities. Figure 18 shows the HCHO trends only whereas Figure 21 shows the HCHO/NO2 ratio annual trends. The caption in both figures read "white areas correspond to regions where the trends remain statistically insignificant at a 95% confidence level.". Therefore, the different number of points for the two different quantities is due to the different significance (confidence) of the different quantities.

As shown in Figure 4 "Mean (2005-2019) of TrC-O$_3$, TrC-NO$_2$, TrC-HCHO, and TC-CO", TrC-NO2 levels over central Australia is 0.5 Pmolec/cm2, and mean (2005-2019) HCHO/NO2 ratio is very high (10) indicating mean (2005-2019) NO sensitive conditions. However, since the HCHO/NO2 ratio shows a small trend, the system is slowly moving toward VOC-sensitive conditions. As mentioned in the last sentence of the last paragraph of section 3.4.6, this is due to increasing NO2 trends but decreasing HCHO trends "Similarly, while NO$_2$ trends are slightly increasing over central and southern Australia, trends of TrC-O$_3$ and TrC-HCHO are decreasing, which indicates a trend toward VOC-limited conditions"

5. While the authors state that HCHO trends are inconsistent with that of O3 in regions including southeastern US (after revision, Lines 761-765), it is said consistent in line 1291-1299, in the Eastern US. Some clarification is needed.

Answer: The sentences in lines 761-765 refer to the trends in some regions, with an example over the southeastern US while in lines 1291-1299 in the conclusion we refer to the trends in the northeastern US. The reviewer is specifically referring to the following sentences in lines 761-765 "HCHO trends are inconsistent with that of $O_3$ (sec. 3.4.1) **in some regions** which might be due to several factors, such as their different sensitivity to $NO_x$ and hydrocarbons (Luecken et al., 2018) but also possible STE contribution to tropospheric ozone levels, especially in midlatitudes (Willimas et al., 2019; Li et al., 2024). For example, while TrC-$O_3$ is increasing in the **southeastern US**, TrC-$NO_2$, TC-CO, and TrC-HCHO are decreasing, which, in addition to the local chemistry, might indicate a STE signal." and lines"1291-1299 "The decreasing trends of TrC-$O_3$ over parts of the **northeastern US** and Europe are likely due to the decreasing trend of TrC-$NO_2$, which is due to the effective measures applied over the last two decades to mitigate air pollution in these regions. TrC-HCHO trends are decreasing in the Eastern US, some parts of northern and western Africa, and western and northern Europe, and increasing in South Asia, central Africa, northern Australia, and Brazil. TrC-HCHO trends are consistent with that of TrC-$O_3$ over Eastern US and Europe."

For further clarification, we have modified the sentences in line 761-765 to be "HCHO trends **varies** with that of $O_3$ (sec. 3.4.1) which might be due to several factors, such as their different sensitivity to $NO_x$ and hydrocarbons (Luecken et al., 2018) but also possible STE contribution to tropospheric ozone levels, especially in midlatitudes (Willimas et al., 2019; Li et al., 2024). For example, while TrC-$O_3$ is increasing in the southeastern US, TrC-$NO_2$, TC-CO, and TrC-HCHO

are decreasing, which, in addition to the local chemistry, might indicate a STE signal. TrC-$NO_2$ trends are decreasing over the northern coast of Australia while those of TrC-$O_3$ and TrC-HCHO are increasing. While the increase of HCHO/$NO_2$ might indicate a trend toward NO-limited conditions (see below), the increase of TrC-$O_3$ trends in this region might also indicate increasing trends of STE contribution (Li et al., 2024). **However, TrC-HCHO trends are consistent with that of TrC-$O_3$ in other regions, e.g., over the northeastern US and Europe.** Similarly, while $NO_2$ trends are slightly increasing over central and southern Australia, trends of TrC-$O_3$ and TrC-HCHO are decreasing, which indicates a trend toward VOC-limited conditions (see below)."

Overall, the manuscript needs major revision in the areas listed above.

We thank the reviewer for his comments, which we have addressed as outlined above.

**Responses to Report#2**

We thank the reviewer for his comments and recommendations. Our responses are in blue following each comment.

Review of Elshorbany 2024
Tropospheric Ozone Precursors: Global and Regional Distributions, Trends and Variability

The revised paper is much improved, shorter, and more evidenced. On the whole the three reviewer's comments have been addressed. I would however recommend that the authors address the below outstanding comments before it can be recommended for publication.

We thank the reviewer for his comments, and we have addressed all the remaining issues.

Review 1
Mostly addressed. References are missing in the responses to the reviewer.
Answer: The mentioned reference is included in the body of the text.
Review 2
1) Addressed sufficiently
2) Needs a reference for the response to the comment around ocean emissions/regions. Issues with this statement were also highlighted within Reviewer 3's comments and this needs more substance and discussion in the text.
Answer: The mentioned reference is included in the body of the text. We have also clarified this sentence as per our response to the reviewer.
3) Addressed sufficiently
4) Addressed sufficiently
5) Addressed sufficiently
6) Addressed sufficiently
Review 3
1) Based on the importance of methane to future tropospheric ozone burden highlighted by Cooper's review (in particular with respect to the IPCC AR6 reference); this needs more attention before considering the impact of the reactive species. Discussion of methane distribution and trends and impact on ozone with reference to other studies should be included in the introduction and also the emphasis of the new paragraph should be the other way around. Suggest discussing methane importance first and then move the focus for the paper to higher reactivity precursors as this is then the new work.
Answer:
Per the reviewer's suggestions, we have added the following paragraph to the introduction:
"Methane, with an assessed total atmospheric lifetime of $9.1 \pm 0.9$ years (Szopa et al., 2021), is also a crucial driver of tropospheric ozone(Fiore et al., 2002; Isaksen et al., 2014). Its accelerated growth rate of $7.6 \pm 2.7$ nmol mol$^{-1}$ yr$^{-1}$ between 2010 and 2019 (Canadell et al., 2021) is largely driven by anthropogenic activities (Szopa et al., 2021). NOAA GML observations of methane (NOAA, 2024) show that methane concentrations in the atmosphere have increased sharply since 2005 (an 8% increase from 2005 to 2023). Future scenarios show that emission control measures can influence future changes to air pollutants. Although the global increases in $CH_4$ abundance may offset benefits to surface $O_3$ from local emission reductions (Fiore et al., 2002; Shindell et

al., 2012; Wild et al., 2012; Szopa et al., 2021), recent reports (e.g., Itahashi et al., 2020; Zanis et al., 2022), showed the dominant role of precursor emission changes in projecting surface ozone concentrations under future climate change scenarios. In this study, we investigate the relation between ozone trends and the trends of its precursors, with a focus on $NO_2$, CO, and HCHO."

2) Addressed sufficiently
3) Addressed sufficiently
4) As this is a paper discussing trends for TOAR-II and contributing to the Community special issue then the guidance should be observed and as advised the authors should try and change the wording they have used from 'significant' to statements about 'confidence'.
Answer:
We have changed the significance levels to the corresponding confidence level.

5) Addressed sufficiently
6) Addressed sufficiently
7) Addressed sufficiently

8) Addressed sufficiently

Minor comments

Mostly addressed but some detail is missing in the responses, if changes have been made- and it

doesn't look like it- they are not described in the response. Authors need to respond to the

comments around Figure 6, line 326, line 343, Figure 14, lines 355-357.

Answer:
We have addressed all the comments related to the figures. The comment on Figure 6 was regarding the time period in the caption, which has been corrected. There was no comment on Figure 14 or the mentioned lines in the revies or track changes versions.

---

## Author Response (AR3)

Responses to the Editor

Dear Authors,

Thank you for the responses to the reviewers' comments. I am still not convinced that you have addressed the first point regarding the altitude dependence and the applicability of the HCHO/NO2 column ratio to infer chemical regimes and its change beyond the strong ozone production region. You mentioned that you have conducted additional ozone sensitivity analysis, which is to be presented in another paper that is being prepared. I would encourage you to include the main results of that analysis in the manuscript to support your statement. 'In preparation' papers cannot be cited - unless available as preprints through an archival repository. Please add the salient analysis from the manuscript under preparation in the supplementary text to support this.

**We thank the editor for his time and effort in reviewing our article. Please find below our answers in blue following each comment.**

Answer: Regarding the reviewer's comment on the altitude dependencies of the HCHO/NO2 ratio, we didn't look into that. So, we can't comment on that from the measurement point of view. However, from the model simulation in Figure 5, the mean (2005-2019) percent contribution from the upper versus lower troposphere differs between HCHO and $NO_2$, both globally and in the tropical band, leading to a lower HCHO/NO2 ratio in the upper troposphere relative to the middle and lower troposphere. The lower HCHO/NO2 ratio in the upper troposphere is due to the lower photochemically formed HCHO mixing ratios, consistent with other studies (Souri et al., 2023; Müller et al., 2024). However, delving into the altitude dependence of the HCHO/NO2 would require an additional significant space/effort and is not among the objectives of this global study and it has been addressed in many other regional studies (e.g., Souri et al., 2023; Chong et al., 2024; Müller et al., 2024).

Regarding the regional trends in HCHO/NO2 ratio, our analysis in section 3.4.7 is indeed limited to ozone production sites, e.g., in Eastern Asia, North America, Europe, and the tropics.

The sensitivity paper will be submitted very soon, but the citation might take a couple of weeks to appear, therefore, we removed the citation to this article from the references list. The sensitivity paper was mentioned in our response to the reviewers but not in the body of this article. However, I have included here a confidential version of a draft of this article for your reference.

Another point is the format of the references. Please use the Copernicus reference formatting style throughout the reference list - there are many instances of wrong formatting being used.

Answer: The format of the references is fixed now.

**References:**

Chong, K., Wang, Y., Liu, C., Gao, Y., Boersma, K. F., Tang, J., & Wang, X. (2024). Remote sensing measurements at a rural site in China: Implications for satellite NO2 and HCHO measurement uncertainty and emissions from fires. Journal of Geophysical Research: Atmospheres, 129, e2023JD039310. https://doi.org/10.1029/2023JD039310, 2024.

Müller, J.-F., Stavrakou, T., Oomen, G.-M., Opacka, B., De Smedt, I., Guenther, A., Vigouroux, C., Langerock, B., Aquino, C. A. B., Grutter, M., Hannigan, J., Hase, F., Kivi, R., Lutsch, E., Mahieu, E., Makarova, M., Metzger, J.-M., Morino, I., Murata, I., Nagahama, T.,

Notholt, J., Ortega, I., Palm, M., Röhling, A., Stremme, W., Strong, K., Sussmann, R., Té, Y., and Fried, A.: Bias correction of OMI HCHO columns based on FTIR and aircraft measurements and impact on top-down emission estimates, Atmos. Chem. Phys., 24, 2207–2237, https://doi.org/10.5194/acp-24-2207-2024, 2024.

Souri, A. H., Johnson, M. S., Wolfe, G. M., Crawford, J. H., Fried, A., Wisthaler, A., Brune, W. H., Blake, D. R., Weinheimer, A. J., Verhoelst, T., Compernolle, S., Pinardi, G., Vigouroux, C., Langerock, B., Choi, S., Lamsal, L., Zhu, L., Sun, S., Cohen, R. C., Min, K.-E., Cho, C., Philip, S., Liu, X., and Chance, K.: Characterization of errors in satellite-based HCHO$/$NO$_2$ tropospheric column ratios with respect to chemistry, column-to-PBL translation, spatial representation, and retrieval uncertainties, Atmos. Chem. Phys., 23, 1963–1986, https://doi.org/10.5194/acp-23-1963-2023, 2023.

Wang, Y., Dörner, S., Donner, S., Böhnke, S., De Smedt, I., Dickerson, R. R., Dong, Z., He, H., Li, Z., Li, Z., Li, D., Liu, D., Ren, X., Theys, N., Wang, Y., Wang, Y., Wang, Z., Xu, H., Xu, J., and Wagner, T.: Vertical profiles of NO$_2$, SO$_2$, HONO, HCHO, CHOCHO and aerosols derived from MAX-DOAS measurements at a rural site in the central western North China Plain and their relation to emission sources and effects of regional transport, Atmos. Chem. Phys., 19, 5417–5449, https://doi.org/10.5194/acp-19-5417-2019, 2019.

**Influence of nitrogen oxides and volatile organic compounds emission changes on**

**tropospheric ozone variability, trends and radiative forcing**

Suvarna Fadnavis[1], Yasin Elshorbany[2], Brice Barret[3], Jerald Ziemke[4], Alexandru Rap[5], Satheesh Chandran PR[1], Richard J. Pope[5], Vijay Sagar[1], Domenico Taraborrelli[6], Eric Le Flochmoe[3], Juan Cuesta[7], Catherine Wespes[8], Folkert Boersma[9,10], Isolde Glissenaar[9], Isabelle De Smedt[11], Michel Van Roozendael[11], Hervé Petetin[12],

[1]Center for Climate Change Research, Indian Institute of Tropical Meteorology, MoES, Pune, India
[2]School of Geosciences, College of Arts and Sciences, University of South Florida, St. Petersburg, FL, USA
[3]LAERO/OMP, Université Paul Sabatier, Université de Toulouse-CNRS, Toulouse, France
[4]NASA Goddard Space Flight Center, Greenbelt, Maryland, USA
[5]School of Earth and Environment, University of Leeds, Leeds, UK
[6]Institute of Energy and Climate Research, IEK-8: Troposphere, Forschungszentrum Jülich GmbH, Jülich, Germany,
[7]University Paris Est Creteil and Université Paris Cité, CNRS, LISA, F-94010 Créteil, France
[8]Université libre de Bruxelles (ULB), Spectroscopy, Quantum Chemistry and Atmospheric Remote Sensing, Brussels, Belgium
[9]Royal Netherlands Meteorological Institute (KNMI), De Bilt, The Netherlands
[10]Wageningen University, Environmental Sciences Group, Wageningen, The Netherlands
[11]Belgian Institute for Space Aeronomy
[12]Barcelona Supercomputing Center, Barcelona, Spain

**Corresponding author email: suvarna@tropmet.res.in**

**Abstract:**

Ozone in the troposphere is a prominent pollutant whose production is sensitive to the emissions of nitrogen oxides (NOx) and volatile organic compounds (VOC). In this study, we assess the variation of tropospheric ozone levels, trends, ozone photochemical regimes and radiative effect using the ECHAM6-HAMMOZ chemistry-climate model which is validated against satellite measurements. Further, we investigate the impacts of doubling/halving $NO_X$ (DNOx/HNOx) and VOC (DVOC/HVOC) emissions on ozone levels, trends, ozone photochemical regimes, and radiative effect. Our analysis shows that the enhancement in global mean tropospheric column ozone (TCO) is sixteen times higher in DNOx (mean:11.7, [minimum: 5.6; maximum:29.3 ppb]) than in DVOC (0.7 [-0.5; 3.6] ppb) simulations. The decrease (increase) in surface ozone with the increase in NOx (VOC) in DNOx (DVOC) simulation indicates the prevalence of VOC-limited regime over Indo-Gangetic Plains, Eastern China, Eastern US, and Europe.

The estimated global mean trends in TCO show increasing tendencies in OMI/MLS (1.4 [-2.1; 4.5] ppb.decade$^{-1}$ from January 2005 to December 2020), IASI-SOFRID (0.08 [-3.9; 7.1] from 2008 to 2020), and ECHAM6-HAMMOZ model simulations (0.9 [-0.9; 4.5] ppb.decade$^{-1}$ from 1998 to 2020). The global mean trend in TCO is increasing in DNOx (2.2 [-1.5; 6.2] ppb.decade$^{-1}$) and DVOC (1.4 [-0.2; 5.8] ppb.decade$^{-1}$) compared to control (CTL) simulations. However, trends are negative in DNOx-CTL over India and China (-1.2 to 4.7 ppb.decade$^{-1}$) and in the DVOC-CTL simulation over Africa and Europe (0.7 to 4.2 sppb.decade$^{-1}$).

The model simulations show nonlinear enhancement in surface ozone over Australia, Amazon Argentina, and the southwest part of Indo-China in response to changes in NOx and VOC emissions. The simulations show VOC-limited regimes over Indo-Gangetic Plains, Eastern China, Western Europe and the eastern part of the US in DNOx, while in HNOx, America and Asia become NOx-limited. In DVOC simulations, spatial extent of VOC-limited regimes decreased over Eastern China and Western Europe.

Further, we provide estimates of tropospheric ozone radiative effects (TO3RE). Estimated global mean TO3RE is 1.21 $W.m^{-2,}$ during 1998-2020 which is enhanced in DNOX-CTL (by 0.36 $W.m^{-2}$) and DVOC-CTL (by 0.01 and reduced in HNOx (by -0.12 $W.m^{-2}$) and HVOC-CTL (by -0.03 $W.m^{-2}$) simulations. We show that the global troposphere in the last 25 years is mostly NOx-limited, and show increasing trends. The anthropogenic NOx emissions have a higher impact on radiative forcing than VOC emissions globally.

**1. Introduction**

Tropospheric ozone, a major air pollutant, has been a pressing issue in recent decades due to its detrimental effect on human and ecosystem productivity and as a short-term climate forcer (IPCC, 2021; Wang et al., 2022). Considering these harmful impacts, the assessment of tropospheric ozone levels and trends is being conducted frequently (Mills et al., 2018; Gaudel et al., 2018, Tarasick et al., 2019). Ozone trends are being assessed from surface observations, in-situ and ground-based measurements, satellite retrievals, and model simulations (Cooper et al., 2014; Tarasick et al., 2018; Cohen et al., 2018). The numbers of Tropospheric Ozone Assessment Reports (TOAR) (Cooper et al 2014; Schultz et al., 2017; Young et al., 2018; Fleming et al., 2018; Lefohn et al.,

2018; Gaudel et al., 2018; Mills et al., 2019; Tarasick et al., 2019) have documented global increases of tropospheric column ozone (TCO) over the course of the 20[th] century. Increasing tropospheric trends are explained by enhanced anthropogenic emissions (Cooper et al., 2014, Zhang et al., 2016) and modulation by climate variability (Lin et al 2014, Lu et al., 2018). Several studies documented an increase in trends in TCO e.g 2%–7% per decade in the northern mid-latitudes, 2%–12% in the tropics (Gulev et al 2021), 5–20 % during 1970 to 1995 over Canada (Tarasick et al., 2019), $2.7 \pm 1.7$ and $1.9 \pm 1.7$ ppb per decade during 1995–2017 over globe from multiple observations (Global Observing System database (IAGOS), ozone sondes), and a multi-decadal GEOS-Chem chemical model simulation (Wang et al., 2022), 0.6 to 2.5 ppb.decade$^{-1}$ during 1950-2014 from the IAGOS measurements and CESM2-WACCM6 model (Fiore et al., 2022). Trends in TCO are stronger in the Northern Hemisphere (NH) than Southern Hemisphere (SH) due to larger anthropogenic emissions (Monks et al., 2015). Ozone Monitoring Instrument (OMI) and Microwave Limb Sounder (MLS) observations from 2005 until 2010 show annual TCO trends averaged over the NH exceed the SH average by 4% at low (0°–25°), by 12% at mid (25°–50°), and by 18% at high (50°–60°) latitudes (Cooper et al., 2014).

The trends in surface ozone have grown during the last century, however, a few locations show decreasing trends (Cooper et al., 2014). The UKESM1 model simulations show that global mean surface ozone increased by ~28% throughout the twentieth century. The set of lower tropospheric and surface ozone measurements in the NH shows an increase in ozone by 30%–70% since the middle of the 20[th] Century (Gulev et al., 2021). Recent observations from UV-absorption analyzers in Southwestern Europe from 2000–2021 show an increase in ozone trends of $2.2 \pm 0.3$

ppb.decade$^{-1}$ (Adame et al., 2022). However, some of the station data showed negative trends in surface ozone during 1980–2001, for example, Goose Bay, Labrador (−0.7±0.4 ppb.decade$^{-1}$),

Churchill, Manitoba, (−0.6±0.4 ppb.decade$^{-1}$), Edmonton, Alberta (−1.4±0.7 ppb.decade$^{-1}$)

(Adame et al.2022). Cooper et al., (2014) reported that surface ozone trends have varied over different regions from 1990 until 2010. In Western Europe, ozone concentration increased in the

1990s, followed by a leveling off or decrease since 2000. Analysis of worldwide monthly surface ozone anomaly data from 2000 to 2018 shows the strongest negative trend of −2.8 ± 1.1

ppb.decade$^{-1}$ at Gothic station (41°N, 2.1°E) and strongest positive trend of 2.2±0.9 ppb.decade$^{-1}$

at American Samoa (14°S, 171°W) (Cooper et al., 2020). Lu et al. (2019) reported surface ozone trends varying between 0.17 % to 0.81 % in the SH from 1990 to 2015.

Ozone trends are influenced by the emission of its precursor gasses. In order to comprehensively address the observed ozone trends and photochemical regimes, it is imperative to gain a deeper understanding of the levels of major ozone precursor gasses, namely nitrogen oxide (NOx) and Volatile Organic Compounds (VOCs). The levels of these ozone precursor gasses vary due to changes in economic activities, natural variability, and due to various pollution control strategies being implemented. In Europe, a reduction of emissions of ozone precursor (HCHO,

CO, $NO_2$) by 27-32% in the past decade (2002-2011) has led to an overall decline in surface ozone levels (Guerreiro et al., 2014). Over the US, a decline in NOx by 38%, VOC by 17%, and CO by

67% during the period 2000-2022 has resulted in a reduction in the national average surface ozone levels by ~17% (EPA 2023). Conversely, Southeast Asia has witnessed a decrease in NOx by 3%

and VOCs by 0.3% during the period 2013-2021 (Ren et al., 2022). Despite these control strategies,

Southeast Asia experienced a rise in surface ozone levels. The prevalence of VOC-limited photochemistry in this region led to reduced NOx titration, resulting in an ozone enhancement (Souri et al., 2017; Lefohn et al., 2017). NOx or VOC are the major precursors that define the ozone photochemical regimes when radiation levels are sufficiently high. The information of ozone photochemical regimes is of utmost importance to control ozone pollution. However, the non-linearity in the $O_3$-NOx-VOC chemistry has always posed a significant challenge in identifying photochemical regimes. The regime is called NOx-limited if the ozone production is directly related to change in NOx, with no impact from VOC perturbations. Whereas the region where ozone production is regulated by the ambient availability of VOCs, it is called VOC-limited (Sillman et al., 1990; Kleinman, 1994). The ratios such as $O_3$/(NOy-NOx), HCHO/NOy,

HCHO/$NO_2$, $H_2O_2$/$HNO_3$ are adopted to diagnose the ozone photochemical regimes (e.g., Sillman,

1995; Martin et al., 2004; Duncan et al., 2010). Among these, the most widely used indicator to identify regimes is the Formaldehyde (HCHO) to Nitrogen dioxide ($NO_2$) Ratio (FNR) (Martin et al., 2004; Duncan et al., 2010). In our study, we adopt FNR to identify NOx-limited or VOC- limited regimes. On par with the current effort to mitigate ozone pollution, it is important to understand how the changes in emissions of NOx and VOC affect the ozone photochemical regimes and trends (Jin et al., 2017, 2020).

Ozone is the third strongest anthropogenic greenhouse gas forcer, also called a near-term climate forcer (Myhre et al., 2013). The Intergovernmental Panel on Climate Change (IPCC) in the fifth assessment report documented ozone changes in the troposphere during the industrial era from 1750 to 2011 exerted a RF of 0.40 (0.20–0.60) W.m$^{-2}$ with a 5%–95 % confidence interval.

The increase in ozone during 1750–2011 has a global radiative forcing of +0.35 W.m$^{-2}$ (Myhre et al., 2013). The CMIP6 model from 1850 up to the present day estimated an ozone RF of

0.39 W.m$^{-2}$ [0.27–0.51] (Skeie et al., 2020). The knowledge of ozone radiative forcing due to changes in anthropogenic emissions of  NOx and VOC will be helpful to assess climate change.

In this study, we also assess the impacts of enhanced or reduced emissions of NOx and VOC on ozone radiative forcing in addition to ozone trends and photochemical regimes. To achieve this, we conducted sensitivity experiments by doubling and halving global NOx and VOC emissions using the state-of-the-art chemistry-climate model ECHAM6-HAMMOZ. The paper is outlined as follows: satellite data and the model experimental setup are given in section 2, results are given in section 3 that includes comparison of simulated tropospheric column ozone with satellite data and estimated ozone trends. Discussions on ozone photochemical regimes and their trends are made in section 4 and 5. Estimates of ozone radiative effects are given in section 6. Conclusions are made in section 7.

**2.  Satellite data and model experiments**

**2.1. OMI Satellite Data**.

We include OMI/MLS tropospheric column ozone (TCO) for October 2004–December 2020 and

OMI NO$_2$, HCHO data for latitude range 60°S–60°N (**Ziemke et al., 2006**). OMI/MLS TCO Yis determined by subtracting MLS stratospheric column ozone (SCO) from OMI TCO each day at each grid point. Tropopause pressure used to determine the SCO invoked the WMO a 2 K.km$^{-1}$

lapse-rate definition from the NCEP reanalyses. The MLS data used to obtain SCO were derived from the MLS v4.2 ozone profiles.  We estimate 1σ precision for the OMI/MLS monthly-mean gridded TCO product to be about 1.3 DU.  Adjustments for drift calibration and other issues (e.g.

OMI row anomaly) affecting OMI/MLS TCO are discussed by Ziemke et al. (2019) and Gaudel et al. (2024).

OMI $NO_2$ Monthly Mean Level 3 dataset consists of the monthly averaged tropospheric $NO_2$

column density as measured by the OMI from October 2004 to March 2021. The data were first spatially averaged in a 1 by 1 degree grid, using a minimum spatial coverage threshold of 30% and then temporally averaged (with a minimum temporal coverage of 10%). We have applied the averaging kernel to model data for comparison with the model.

**2.2  IASI-SOFRID**

The Software for a Fast Retrieval of Infrared Atmospheric Sounding Interferometer (IASI) data (SOFRID) retrieves global ozone profiles from IASI radiances (Barret et al., 2011, 2021). It is based on the RTTOV (Radiative Transfer for TOVS) operational radiative transfer model jointly developed by ECMWF, Meteo-France, UKMO and KNMI within the NWPSAF (Saunders et al.,

1999; Matricardi et al., 2004). The RTTOV regression coefficients are based on line-by-line computations performed using the HITRAN2004 spectroscopic database (Rothman et al., 2005), and the land surface emissivity is computed with the RTTOV UW-IRemis module (Borbas et al.,

2010). The IASI-SOFRID ozone for the study period (2008 to 2020) is obtained from METOP-A

(2008-2018) and METOP-B (2019-2020).

We use the SOFRID version 3.5 data presented and validated in **Barret et al. (2021)**, which uses dynamical a priori profiles from an $O_3$ profile tropopause-based climatology according to tropopause height, month, and latitude (Sofieva et al., 2014). The use of such an a priori has largely improved the retrievals, especially in the southern hemisphere where the previous version was significantly biased. The retrievals are performed for clear-sky conditions (cloud cover fraction <

20%). IASI-SOFRID ozone retrievals provide independent pieces of information in the troposphere, the UTLS (300–150 hPa), and the stratosphere (150-25 hPa) (**Barret et al., 2021**).

SOFRID TCO absolute biases relative to ozonesondes are lower than 8 % with root mean square error (RMSE) values lower than 18 % across the six 30° latitude bands (see Barret et al., 2021).

Importantly, Barret et al. (2021) have shown that relative to ozonesondes, TCO from IASI-

SOFRID  display no significant drifts (<2.1 % decade$^{-1}$) for latitudes lower than 60°N and in  the

SH for latitudes larger than 30° (<3.7 % decade$^{-1}$).  But significant drifts are observed in the SH

tropics (-5.2% decade$^{-1}$) and in the NH at high latitudes (12.8% decade$^{-1}$). We have applied the averaging kernel to model data for comparison with the model.

**2.3 IASI+GOME2**

IASI+GOME2 is a multispectral approach to retrieve the vertical profile of ozone and its abundance in several partial columns. It is based on the synergy of IASI and GOME2 spectral measurements in the thermal infrared and ultraviolet spectral regions, respectively, which are jointly used to improve the sensitivity of the retrieval for the lowest tropospheric ozone (below 3

km above sea level, see **Cuesta et al., 2013**). Studies over Europe and East Asia have shown particularly good capabilities for capturing near-surface ozone variability compared to surface in situ ozone measurements (**Cuesta et al. 2018; 2022; Okamoto et al., 2023**). TCOs from

IASI+GOME2 also show good agreement with several datasets of in-situ measurements for a four- year period in the tropics, with almost negligible biases and high correlations (**Gaudel et al., 2024**).

This ozone product provides global coverage for low cloud fraction conditions (below 30%) for

12 km diameter pixels spaced 25 km apart (at nadir). The IASI+GOME2 global dataset is publicly available through the French AERIS data center, with data from 2017 to the present (available at (O3 (IASI+GOME2) – IASI portal, 2024) and covers the 90° S-90° N latitude band. For this study, we use the monthly TCO data between the surface and the tropopause for 2017 - 2022 for different latitude bands. We have applied the averaging kernel to model data for comparison with the model.

**2.4 TROPOMI**

The TROPOspheric Monitoring Instrument (TROPOMI) is the sole payload on the Copernicus

Sentinel-5 Precursor (Sentinel-5P or S5P) satellite, which provides measurements of multiple atmospheric trace species, including $NO_2$ and HCHO, at high spatial and temporal resolutions (Veefkind et al., 2012).  TROPOMI has a 108° field of view and spans the ultraviolet-visible (270–

495 nm), near-infrared (675–775 nm), and shortwave infrared (2305–2385 nm) wavelength ranges at the nadir view. It has a daily global coverage with a spatial resolution of $5.5 \times 3.5$ km$^2$ at nadir since a long-track pixel size reduction on 6 August 2019. We have used the Tropospheric column of $NO_2$ and HCHO data for our study. We have applied the TROPOMI averaging kernel to model data for comparison with the model.

**a.    2.5 The ECHAM6-HAMMOZ model experiments**

The ECHAM6.3-HAM2.3-MOZ1.0 aerosol chemistry–climate model (Schultz et al., 2018) used in the present study comprises the general circulation model ECHAM6 (Stevens et al., 2013), the tropospheric chemistry module, MOZ (Stevenson et al., 2006) and the aerosol module, Hamburg Aerosol Model (HAM) (Vignati et al., 2004). The gas phase chemistry is represented by the Jülich Atmospheric Mechanism (JAM) v002b mechanism (Schultz et al., 2018). This scheme is an update and an extension of terpenes and aromatics oxidation based on the MOZART-4 model (Emmons et al., 2010) chemical scheme. Tropospheric heterogeneous chemistry relevant to ozone is also included (Stadtler et al., 2018). MOZ uses the same chemical preprocessor as CAM-Chem (Lamarque et al., 2012) and WACCM (Kinnison et al., 2007) to generate a FORTRAN code containing the chemical solver for a specific chemical mechanism. Land surface processes are modeled with JSBACH (Reick et al., 2013). Biogeninc VOC emissions are modeled with the MEGAN algorithm (Guenther et al., 2012) which has been coupled to JSBACH (Henrot et al., 2017). The lightning NOx emissions are parameterized in the ECHAM6-HAMMOZ as per Rast et al. (2014). The lightning parameterization is the same in all the simulations. The model simulations were performed for the period 1998 to 2020 using Atmospheric Chemistry and Climate Model Intercomparison Project (ACCMIP) (Lamarque et al., 2010, van Vuuren et al., 2011) emission inventory. ACCMIP emission inventory includes emissions from agriculture and waste burning, forest and grassland fires, aircraft, domestic fuel use, energy generation, including fossil fuel extraction, industry, ship traffic, solvent use, transportation, and waste management. We used the high emission scenario Representative Concentration Pathway (RCP) 8.5 emissions (van Vuuren et al., 2011) to show their impact on ozone variability and trend.

The model is run at a T63 spectral resolution corresponding to about 1.8° × 1.8° in the horizontal dimension and 47 vertical hybrid σ–p levels from the surface up to 0.001 hPa. The details of model parameterizations and validation are described by Fadnavis et al. (2019a,b; 2021a,b; 2022, 2023).

We performed five experiments: (1) control and four emission sensitivity experiments: (2)

doubling anthropogenic emission of $NO_X$ globally (DNOx), (3) reduce anthropogenic emissions of $NO_X$ by 50 % globally (HNOx), (4) doubling anthropogenic emissions of all VOCs globally (DVOC), (5) reducing anthropogenic emissions of all VOCs by 50 % globally (HVOC).  We performed each experiment from 1998 to 2020 after a spin-up of one year. In each experiment, the monthly varying AMIP-II Sea surface temperature and Sea ice representative of the period 1998–

2020 were specified as a lower boundary condition. VOCs considered in this study are listed in the supplementary table-S1.

The tropospheric column ozone (TCO) is computed from the satellite data and model simulations by averaging $O_3$ amounts from the surface up to the tropopausIe. Tropopause is as per WMO

thermal tropopause; the lowest level at which the temperature lapse rate decreases to 2 $K.km^{-1}$ or less (Maddox and Mullendore, 2018). For comparison of the model with satellite datasets, e.g.

IASI-SOFRID, OMI/MLS, we use model and satellite data for the same period, also, apply an averaging kernel of each satellite on the model data during the respective comparison.

**2.5 Tropospheric ozone radiative effects**

The tropospheric ozone radiative effect (TO3RE) is calculated as in Pope et al (2024). While the radiative forcing calculated in ECHAM6–HAMMOZ also includes impacts of aerosols and dynamical effects, here we isolate TO3RE by using the Rap et al. (2015) tropospheric ozone radiative kernel derived from the SOCRATES offline radiative transfer model (Edwards and

Slingo, 1996), including stratospheric temperature adjustments. To calculate the TO3RE, the monthly averaged ECHAM6-HAMMOZ simulated ozone field is multiplied by the offline radiative kernel (at every grid box). It is then summed from surface to the tropopause. The simulated ozone data are mapped onto the spatial resolution of the radiative kernel and then interpolated vertically onto its pressure grid. The equation for each grid box is

$\text{TO3RE} = \sum_{i=\text{surf}}^{\text{trop}} RK_i \times O_{3i} \times dp_i / 100$ (1)

where TO3RE is the tropospheric ozone radiative effect (W.m$^{-2}$), RK is the radiative kernel (W.m$^{-2}$.ppbv$^{-1}$.100 hPa$^{-1}$), O$_3$ is the simulated ozone grid box value (ppbv), dp is the pressure difference between vertical levels (hPa), and i is the grid box index between the surface pressure level and the tropopause pressure. The tropopause pressure is based on the World Meteorological

Organization (WMO) definition of "the lowest level at which the temperature lapse rate decreases to 2 K.km$^{-1}$ or less" (WMO, 1957). Several past studies have used this approach of using the

SOCRATES offline radiative kernel with output from model simulations to derive the TO3RE

(Pope et al., 2014, Rap et al., 2015, Scott et al. 2018, and Rowlinson et al. 2020).

**3. Results**

**3.1 Comparison of Latitudinal Variation Seasonal Cycle in TCO Satellites Retrievals**

In this section, we compare the estimated TCO from the model (CTL) simulation with OMI/MLS (2006-2020), IASI-SOFRID (2008-2020), and IASI-GOME2 (2017-2020) satellite retrievals. We compared simulated TCO for the same period as individual satellite retrievals and applied an averaging kernel of that satellite to the model. The comparison of monthly mean TCO is made for 20° latitude bins in Figure 1. In the northern tropics (0°-20°N) (Fig 1a), the OMI/MLS data exhibits an annual cycle with a peak in April, whereas the model indicates a peak in January. Both datasets show a minimum in August. The model underestimates TCO by 1.8 to 4 ppb during March to October. In the 21-40 °N and 41-60 °N latitude bands (Fig. 1 b-c), the model shows a one-month lead in the peak of the annual cycle compared to OMI/MLS. Within these bands, the model underestimates OMI/MLS TCO by (2.8 - 6.1 ppb) during the summer months (May-August), while it overestimates TCO by 0.6 - 8.4 ppb during November to February. The 41-60 °N latitude band exhibits an overall underestimation (1.5 - 3.2 ppb) in June-July while it overestimates (0.7 - 5.9 ppb) rest of the year. In the Southern Hemisphere 9SH), OMI/MLS and the model show a similar pattern in the seasonal cycle. There is a consistent underestimation in the model for all months by 9 to 16.9 ppb in the 0-20 °S; 15.9 - 25.3 ppb in 21-40 °S and 41-60 °S. The comparison of TCO from IASI-SOFRID with the model shows features similar to those in the OMI/MLS. In the 0-20 °N latitude band, the model underestimates the TCO by about 3.6 to 7.5 ppb during April to October and in the 21-60 °N latitude band by 1.8 - 11 ppb in summer (May-August). In the SH, the model shows a closer association in TCO with IASI-SOFRID compared to OMI/MLS. During the SHc winter (June-August), the model overestimates TCO by 10 - 19.2 ppbv in the latitude range of 0-40 °S. Conversely, it underestimates TCO by 17.7 - 23.4 ppbv in the 41-60 °S

throughout the year, which is less compared to other satellite datasets. IASI-SOFRID is known to suffer from negative drifts in the SH.

Interestingly, the model exhibits a fair agreement with IASI-GOME2 retrieved TCO during the summer months (May-August) in the entire Northern Hemisphere (NH). During the winter months, the estimated TCO shows a large overestimation of 8.3 - 11.7 ppb in the NH (0-40 °N).

In the SH, a fairly good agreement is observed between the model and IASI-GOME2 TCO, especially in the 0-40 °S latitude band. The model overestimates the TCO by 7.4 - 8.8 ppb in the

0 - 20 °S during SH winter and underestimates by 4.7 - 6.7 ppb in the 21 - 40 °S band during SH

summer (December-January-February). An overall underestimation of about 7 - 11.2 ppb in TCO

is noted in the 41-60 °S throughout the year. Figure 1 shows that a peak in the seasonal cycle, in the model is earlier than the three satellite data between 40°N and 40°S. In general, the model underestimates TCO in summer in the NH and overestimates in winter relative to OMI-MLS, and

IASI-SOFRID. In the SH, the model underestimates TCO throughout the year compared to OMI-

MLS, IASI-SOFRID and IASI-GOME2. This underestimation is large for OMI/MLS, while IASI-

SOFRID and IASI-GOME2 are larger than the model in the latitude band 0-40 °S during the SH

winter. Although the model-satellite comparison is done for the same time period and averaging kernel of each of the satellites are applied on the model during comparison, the differences in sampling between the model and satellite measurements may cause these differences.  It should be noted that there are differences among the satellites. The resolution of each of the data (CTL

1.8°×1.8°), IASI-SOFRID (5°×5°), IASI-GOME2 (..), and OMI/MLS ( 5°×5°) is different. This may be causing differences among them.

[Figure]

Figure 1: Monthly mean time series of TC ozone (ppb) averaged for 20° wide latitude bins from
(a-f) OMI-/MLS (blue) and ECHAM6-HAMMOZ CTL simulations (black) for the time period
October 2004-December 2020. (g-l) same as (a-f) but for IASI-SOFRID (blue) and ECHAM6-
HAMMOZ CTL simulations (black) for the time period January 2008- December 2020, and (m-r)
same as (a-f) but for IASI+GOME2 (blue) for time period January 2017 - December 2022 and
ECHAM6-HAMMOZ CTL simulations (black) for the time period January 2017 - December
2020. The vertical bars in the figures represent 1σ standard deviation.

Further, we compare the simulated tropospheric column $NO_2$ and HCHO with the ESA CCI+

monthly averaged TROPOMI and OMI data (Glissenaar et al., 2024) (Fig. 2). The simulated $NO_2$

reproduces the seasonal cycle but overestimates in the NH and SH tropics in TROPOMI and OMI

(Fig. 2 a-f) by 0.12 to 0.4 $\times 10^{15}$ molecules.cm$^{-2}$ in the NH, and by 0.01 to 0.1 $\times 10^{15}$

molecules.cm$^{-2}$ in the SH. Simulated $NO_2$ shows a underestimation in the 21-60°N latitude belts in the NH (0.8x$10^{15}$ molecules.cm$^{-2}$) except in summer season and fairly good agreement in the

SH 21-60°N (0.6 to 1.5$\times 10^{15}$ molecules.cm$^{-2}$).

Simulated HCHO (Fig. 2 g-l) is overestimated in 0-20° N belt during January-February and July-

December (by 0.2-1 x $10^{15}$ molecules.cm$^{-2}$) compared to TROPOMI. However, it shows underestimation compared to OMI by 2-3 x $10^{15}$ molecules.cm$^{-2}$. The simulated HCHO shows underestimation over all the latitude bands except 0-20°N compared to TROPOMI and OMI. It should be noted that TROPOMI/OMI monthly means are valid for clear-sky situations, whereas the model simulations are 31-day all-sky averages. In previous studies (Boersma et al., 2016 and references therein), it was shown that $NO_2$ is typically 15-20% lower on clear-sky days than under cloudy situations due to lower photolysis rates, and slower chemical loss of $NO_2$. This effect likely explains part of the model overestimate compared to TROPOMI $NO_2$. For HCHO the effect is smaller because HCHO is both produced and destroyed by OH (see Fig. 4 in Boersma et al. 2016).

Considering these differences we proceed for the analysis of TCO trends, ozone photochemical regimes, and ozone radiative effects. The overestimation/underestimation of ozone will be more or less the same in CTL, DNOx, DVOC, HNOx and HVOC simulations. The anomalies DNOx -

CTL, DVOC - CTL, HNOx - CTL, and HVOC - CTL will be less impacted by the overestimation/underestimation of TCO in the model.

[Figure]

Figure 2: Monthly mean time series of TC NO₂ (ppb) averaged for 20° wide latitude bins from ECHAM6-HAMMOZ CTL simulations (black) for the time period same as TROPOMI satellite for (a-f) (May 2018 - December 2020), (g-l) same as a-f but for HCHO. The vertical bars in the figures represent 1σ standard deviation. The 1σ standard deviation is estimated from data within the latitudinal belt during the period of observations.

**3.2. Impacts of emission changes on the spatial distribution of ozone**

Figure 3 shows the spatial distribution of the simulated surface (Fig. 3a-e) and tropospheric column ozone (TCO) (Fig. 3f-j) concentration from ECHAM-CTL simulations and the differences in DNOx - CTL, DVOC - CTL, HNOx - CTL, and HVOC - CTL simulations for the period 1998-

2020. The CTL simulation shows high surface ozone levels (19 - 61.1 ppb) between 10-40 °N (Fig.

3a). Doubling of NOx emission (DNOx) causes a global mean enhancement of surface ozone by

4.1 ppb. Surface ozone increases by 5-20 ppb compared to the control scenario across most of the globe, excluding highly urbanized regions like the Indo-Gangetic plains (IGP), Southeast China,

Northeastern US, and Europe. (Fig. 3b). Over these regions, a large reduction (8 to 20 ppb) in surface ozone is noticed in response to DNOx conditions indicating ozone titration by $NO_X$. While an increase in surface ozone concentrations is observed globally for DVOC (0.92 ppb), its magnitude is less than that of the DNOx condition (Fig. 3c). The largest increase in surface ozone concentration for DVOC is observed over Indo-Gangetic Plains, Eastern China and the Eastern

United States (3-6 ppb). Interestingly, these are the same regions where a decrease in ozone is observed in the DNOx simulation. The decrease (increase) in ozone with an increase in NOx (VOC) indicates that these regions could be NOx-saturated or VOC-limited. Reduction of $NO_X$

emissions (HNOx-CTL) simulations show a reduction in surface ozone globally (-2.53 ppb) except over North-Eastern China (Fig. 3d). Earlier, Souri et al. (2017) reported that eastern Asia has witnessed a rise in surface ozone levels despite NOx control strategies, indicating the prevalence of VOC-limited photochemistry over this region (details in section 4 and 5). However, the absence of such an increase over other VOC-limited regions points towards the possibility of nonlinear ozone chemistry. While HVOC stimulation causes a reduction in surface ozone globally (-0.44

ppb), an increase is observed in South America, some parts of the US, Australia, and the Indo-

China peninsula (Fig. 3e). This increase could be due to a reduction in the radical destruction of ozone caused by aromatic hydrocarbons in low NOx conditions observed in HVOC simulations compared to CTL simulations in these regions (Taraborrelli et al., 2021).

On the other hand, the estimated global mean TCO from the ECHAM-CTL simulation from 1998 to 2020 is 39.45 ppb. CTL simulations show higher amounts of TCO in the Northern

Hemisphere (NH) within the latitudinal band of 20° to 40°N (40.9 to 68.8 ppb). These concentrations are pronounced over South and East Asia, spanning from the Mediterranean region to eastern China (Fig. 3f). Doubling of anthropogenic $NO_X$ emissions (DNOx) enhances the global

TCO by 11.7 ppb compared to CTL (Fig. 3g). The ozone enhancement exceeds the CTL by 6.1 -

29.3 ppb between 20°- 40°N, particularly over South Asia. Interestingly, in highly urbanized areas such as the Indo-Gangetic plains, South-East China, Northeast US, and Europe, there is only a marginal increase in ozone levels (~5 ppb). This suggests the existence of a distinct ozone photochemical regime in these regions. Further exploration of this aspect will be conducted in sections 4 and 5.

The impact of the doubling of VOC emissions (DVOC-CTL) on ozone is depicted in Figure 3h.

An increase in global mean TCO by 1 ppb is observed in this emission scenario. The spatial distribution of TCO anomalies shows that enhancement in TCO is ten times less than that of doubling $NO_X$ condition (-0.8 - 3.6 ppb in DVOC compared to 5.6 - 29.3 ppb in DNOx) (Fig. 3g and 3h). Large values of TCO (1.5-2) are observed in the high latitudes (north of 60°N) and South and East Asia, with the largest values of more than 2.5 ppb over east China (e.g., Beijing).

Interestingly, in the tropical regions slight decreases in TCO are simulated. This is consistent with the recent finding that aromatics, especially benzene, can lead to efficient ozone destruction in the tropical UTLS (Rosanka et al., 2021). The TCO anomalies in response to the reduction of $NO_X$

emission by 50% (HNOx-CTL) show negative TCO anomalies all over the globe (Fig. 3i). The global mean TCO is reduced by -3.73 ppb. Large decreases in TCO are observed over Arabia,

South and East Asian regions (2.6 - 12.8  ppb). Reducing VOCs by 50 % (HVOC) causes an overall decrease in TCO of 0.27 ppb (Fig. 3j). A small enhancement is noted in the TCO by 0.5 - 1 ppb in the southern tropics and South polar region, while a decrease of -2.3 to 0.3 ppb is observed in the

Northern Hemisphere. (Fig. 3j). Figure 3 clearly portrays that the TCO response to NOx emission change is larger than that of VOCs and shows a spatially distinct distribution associated with the region-specific ozone photochemical regimes (more discussion on the ozone photochemical regimes will be detailed in section 4 and 5).

[Figure]

Figure 3: Spatial distribution of surface ozone (ppb) for (a) from CTL, (b) anomalies from DNOx - CTL, (c) anomalies from DVOC - CTL, (d) anomalies from HNOx - CTL, (e) anomalies from HVOC - CTL. Figures (f) to (j) are the same as those of figures (a) to (e) but for TCO. The stippled regions in the figures indicate anomalies significant at 95% confidence. The tropopause considered is WMO-defined lapse rate tropopause (WMO, 1957).

**3.3. Spatial distribution of trends in TCO, NO₂ and HCHO**

The trends in TCO estimated from ECHAM-CTL simulations are compared with satellite-retrieved TCO to gain more confidence in the model-derived trends. Since IASI-GOME2 has a short observation period and IASI-SOFRID has negative drift in the southern hemisphere, only OMI/MLS (October 2004 to December 2020) is considered for trend estimation and is shown in Figure 4. The spatial pattern of trends from OMI/MLS closely aligns with model simulations for the period October 2004 to December 2020. OMI/MLS show a slightly lower trend of 1.41 [-2.1; 4.5] ppb.decade$^{-1}$ than model simulation 1.28 [-1.5;3.7] ppb.decade$^{-1}$. Both datasets reveal pronounced trends, ranging from 3-4 ppb.decade$^{-1}$, across regions such as South Asia, East Asia, the western Pacific, and the Southern Hemisphere between 0°-30°S. OMI/MLS show negative trends over parts of Africa, South America, Australia and the South-eastern Pacific (Fig. 4b), which is not simulated in ECHAM6-HAMMOZ. This may be due to the model's tendency to underestimate ozone levels and disparities in the seasonal cycle compared to OMI/MLS data. (see Fig. 1).

The CESM2-WACCM6 simulation from 1950 to 2014 also shows the largest estimated trends at 20–30° N of 0.8 Tg.decade$^{-1}$ **(Fiore et al 2022)**. Recently, **Wang et al (2021)** reported TCO trends varying between 2.55 to 5.53 ppb.decade$^{-1}$ during 1955-2017 over South and East Asia using IAGOS, ozonesonde observations, and GEOS-Chem simulations. Further, a large positive trend of ~2.5 ppb.decade$^{-1}$ observed near 50°S in OMI/MLS is not simulated by the model (Fig. 4a-b). TCO trends analyzed from the Total Ozone Mapping Spectrometer (TOMS) indicate a consistent absence of trend over the tropical Pacific Ocean, with notable positive trends (5–9% per.decade$^{-1}$)

seen in the mid-latitude Pacific regions of both hemispheres. This pattern is consistent across the

ECHAM6-HAMMOZ and OMI/MLS data, although their magnitude differs (Fig. 4 a-b). TOMS

data also showed trends of ~2–9% decade$^{-1}$ across broad regions of the tropical South Atlantic,

India, Southeast Asia, Indonesia, and the tropical/subtropical regions downwind of China during

1979–2003 (Ziemke et al., 2005, Beig and Singh 2007).

[Figure]

Figure 4: Trend of tropospheric column Ozone (TCO) (ppb/decade) from (a) ECHAM
CTL, and (b) OMI/MLS satellite for the period January 2005 to December 2020. Stippled regions
in the figures indicate trends significant at 95% confidence. The tropopause considered is WMO
defined lapse rate tropopause.

Figure 5 shows the spatial distribution of estimated trends in surface ozone and TCO from

ECHAM simulation for the period 1998-2020 from CTL, DNOx - CTL, DVOC-CTL, HNOx-

CTL, and HVOC - CTL. The surface ozone trend in the CTL simulation shows a pronounced global increase, particularly notable over South Asia and the Middle East (Fig. 5a). This rise is also seen in the TCO trend (Fig. 5f). However, the negative trends in surface ozone over Mexico, certain parts of the US, and East China are barely discernible in the TCO data. This discrepancy may stem from the interplay of mixing and transport processes, which are crucial when assessing ozone levels across the total column. For a double-NOx emission condition, both surface ozone and TCO exhibit large negative trends over India, China and Australia (-0.4 to -2 ppb.decade$^{-1}$), while trends are positive over Europe, the US, some parts of Africa and South America (Fig. 5b and 5g). A global mean increase in TCO trend by 1.2 [-5.3; 5.8] ppb.decade$^{-1}$ is seen in a DNOx simulation. Interestingly, a positive trend in TCO is seen over the oceanic regions downwind of major continental regions like Africa, China and the US, which was absent in the surface ozone trend, indicating the potential contribution of transport in the TCO trend (Fig. 5g). When global emissions of VOCs are doubled, a decreasing trend (-0.8 to -1.9 ppb.decade$^{-1}$) in surface ozone is noted over Europe, Africa and some parts of the US, while strong positive trends (1.6 to 2

ppb.decade$^{-1}$) are seen over India and China (Fig. 5c). Though the surface trends are faintly captured in the TCO trend, an enhancement over South Asia, China, parts of the Indian Ocean, and the western Pacific are noted (Fig. 5h). Compared to DNOx, the enhancement in the TCO

trend for DVOC is only marginal but shows a global mean trend of 0.5 [-0.85;1.93] ppb.decade$^{-1}$.

Reducing NO$_X$ and VOC emissions by 50% (HNOx and HVOC) decreases the global mean TCO

trends by 0.38 [-1.96;1.64] ppb.decade$^{-1}$ and 0.42 [-0.96;1.61] ppb.decade$^{-1}$ respectively. Though a large positive trend is observed in some parts of China, India, US for both HNOx and HVOC

simulations, it is not that evident in their TCO trend.

[Figure]

Figure 5: (a) Trend of surface ozone (ppb.decade$^{-1}$) from (a) CTL, (b) DNOx, (c) DVOC, (d) HNOx, (e) HVOC simulation. (f) to (j), same as that of figures (a)-(e) but for the tropospheric column ozone (TCO) trend. The stippled regions in the figures indicate significance at 95% confidence. The tropopause considered is WMO defined lapse rate tropopause.

**3.4. Trends in emission and tropospheric column of NO₂ and HCHO**

We show mean emissions of NOx and HCHO over urban/semi-urban regions; US, Brazil, Europe, Africa, India, China, Australia in Figure 6. Figure 6 clearly portrays high emissions of VOCs and NOx in India and China. Furthermore, VOCs emissions are noted to be higher than NOx over all the regions. They are higher by a factor of 3.3 in the US, 11.3 in Brazil, 4.8 in Europe, 10.5 in Africa, 10.8 in India, 6.1 in China, and 6.7 in Australia.

[Figure]

Fig. 6: Box and whisker plot illustrating the NOx and VOCs emission over the regions US (85°W - 110°W, 35°N - 44°N), Brazil (34°W- 49°W, 24°S - 3°S), European Union (9°W - 45°E, 35°N - 55°N), Central Africa (14°W - 45°E, 0° - 14°N), India (75°E - 90°E, 8°N - 30°N), China (110°E - 125°E, 30°N - 42°N), South Australia (134°E - 154°E, 38°S - 28°S). The box represents the 25 and 75 percentile, and the whisker represents the 5 and 95 percentile. The plus marker represents the mean and the horizontal bar represents the 1 and 99 percentile.

The trends in ozone are partly modulated by the change in the emission of its precursors and partly by meteorology (e.g., Verstraeten et al., 2015). Further, we show trends in emissions and tropospheric column amounts of ozone precursors NO₂ and HCHO, from ECHAM-CTL and OMI satellite retrievals in Figure 7. NO₂ and HCHO are considered here because column concentration of these will be used to identify the ozone photochemical regimes discussed later in

Sections 4-5. Emission and tropospheric columns of HCHO and $NO_2$ from ECHAM-CTL show large positive trends over the South and East Asian regions (Figure 7a-d). These regions show large positive ozone trends in both model and OMI satellite data (see Figures 4a-b and 5a,f). Over

Europe and the US, the emission trend in both HCHO and $NO_2$ is negative. Though a similar negative trend in tropospheric column $NO_2$ is seen over these regions, a marginal positive trend (insignificant) is noted for HCHO (Figures 7c-d). The positive trend in column HCHO could be due to secondary production pathways from biogenic emissions or methane oxidation and transport (e.g., Alvarado et al., 2020; Anderson et al., 2017). The positive trend in ozone (Figures 4a-b and

5a,f) along with a negative trend in $NO_2$ and HCHO (Figure 7a-d) over Europe indicates that ozone production over this region has been initially controlled by VOCs (i.e., VOC-limited regime; detailed discussed in section 4). However, a large decreasing trend in $NO_2$ compared to that of

HCHO over this region might have decreased the NOx titration effect, resulting in an increase in ozone.  On the contrary, a negative trend in surface ozone (Figure 5a) along with negative trends in $NO_2$ and HCHO are seen over the US (Figure 7a-b). The decrease in both $NO_2$ and HCHO

would have resulted in a decreasing trend in surface ozone over this region. This also indicates that the US might have been in a NOx-sensitive regime before and the large negative trend in $NO_2$

might have resulted in the decreasing trend in ozone (discussed further in section 4).

Further we compared the simulated trends in column HCHO and $NO_2$ with the OMI retrievals for the period 2005-2020 (Figures 7e-h). OMI shows a positive trend in tropospheric column HCHO

over South Asia, parts of eastern China, the Iranian Plateau, the Amazon and central Africa. The model simulated trends show reasonable agreement with OMI, except for some areas in central

Africa. Additionally, differences are seen in regions such as the US, Northern Africa, Australia, and Argentina, where OMI indicates a negative trend, while the model suggests a marginal positive trend. Both OMI and ECHAM CTL show a good agreement in the tropospheric column $NO_2$ trend.

Both datasets show negative trends over the US and Europe, and positive trends over the Middle

East, South Asia, and Eastern China. Thus, Figures 4, 5, and 7 clearly indicate the impact of ozone precursors on the spatial distribution of ozone trends. Figure 7 further indicates the prevalence of different ozone photochemical regimes associated with the availability of HCHO and $NO_2$. This warrants a detailed discussion on the spatial distribution of ozone precursors and their impact on ozone production sensitive regimes, which will be presented in the next section.

[Figure]

Figure 7: Trend in (a) anthropogenic emission of HCHO (kg.m$^2$.s$^{-1}$.decade$^{-1}$) (b) anthropogenic emission of NO$_2$ (kg.m$^2$.s$^{-1}$.decade$^{-1}$) (c) tropospheric column HCHO (molecules.cm$^{-2}$.decade$^{-1}$), and (d) tropospheric column NO$_2$ (molecules.cm$^{-2}$.decade$^{-1}$) from ECHAM CTL simulation for the period 1998-2020. (e) and (f) Trend in tropospheric column HCHO from OMI and ECHAM CTL simulation respectively for the period 2005-2020. (g) and (h) is the same as that of (e) and (f) but for tropospheric column NO$_2$. The stippled regions in the figures indicate data significance at 95% confidence. The tropopause considered for column estimate is WMO defined lapse rate tropopause.

   **3.  Influence of NOx and VOCs emissions on Formaldehyde to Nitrogen dioxide Ratio**

In this section, we diagnosed the spatial distribution of ozone production sensitivity regimes (NOx- limited/VOC-limited) associated with different simulations of emission changes by using formaldehyde to nitrogen dioxide ratio (FNR). We estimate the FNR thresholds from ECHAM6-

HAMMOZ model simulations adhering to the methodology outlined by Jin et al. (2017). The method to obtain FNR involves two steps: (1) obtaining the ozone response from emission sensitivity simulations (here, HNOx and HVOC simulations) and plotting it as a function of FNR

(Fig. 8a), (2) calculating cumulative probability from this data for the conditions $d[O_3]/dE_{NOx} < 0$

(NOx limited) and ($d[O_3]/dE_{NOx} > d[O_3]/dE_{VOC} > 0$) (VOC-limited) (Fig. 8b). This approach is applied to estimate FNR thresholds to distinctly delineate the various ozone photochemical regimes as NOx or VOC-limited over major urban and semi-urban regions over the globe. The regions considered for estimating the FNR are shown in Figure 9.

[Figure]

Figure 8: (a) Typical example of a normalized surface ozone sensitivity to a 50% reduction in
global NOx (HNOx) and VOC (HVOC) emissions versus tropospheric column HCHO/NO₂ ratio
derived from ECHAM6-HAMMOZ model simulation over China for the period 1998-2020 (b)
Cumulative probability (CP) of VOC-sensitive ($d[O_3]/dE_{NOx} < 0$) and NOx-sensitive ($d[O_3]/dE_{NOx}$
$> d[O_3]/dE_{VOC} > 0$) conditions, as a function of tropospheric column HCHO/NO₂ as simulated by
the ECHAM6-HAMMOZ model. The horizontal dashed line represents the 95% CP, and the
vertical dashed lines represent the  HCHO/NO₂ ratio corresponding to 95% CP for both the VOC-
sensitive and NOx-sensitive curve demarcating the  VOC-sensitive, NOx-sensitive, and transition
regimes.

[Figure]

Figure 9: The rectangular box marks indicate the regions considered for estimating the
HCHO/NO₂ ratio (FNR).

Table 3 presents FNR thresholds across the regions outlined in Figure 8. Based on ECHAM6-

HAMMOZ simulations, our analysis closely mirrors the threshold ranges documented in prior research. For instance, during summer in the USA, many studies report FNR thresholds within the

0.8-2 range (Roberts et al., 2022; Chang et al., 2016; Jin et al., 2017), while our simulations indicate a range of 0.3 to 1.05. Similarly, across China, previous studies (e.g., Lee et al., 2022;

Chen et al., 2023) have reported FNR thresholds spanning 1-2/0.6-3, aligning closely with our simulated range of 0.6-1.45. It is interesting to note that the transition region exhibits a very narrower range in the US, Europe, and China, indicating that the transition from VOC-limited to

NOx-limited can happen suddenly in response to changes in the emission of NOx/VOC. Whereas the transition region is wider in Central Africa.

Table 3. Estimated values of the tropospheric HCHO/NO$_2$ columns threshold ratios from
ECHAM6-HAMMOZ model control simulation to identify the NOx and VOC sensitive regimes
across various regions. The FNR less than the lower limit indicates VOC-limited, and that higher
than the upper limit indicates NOx-limited regimes.

| Sr. No. | Regions | Transition limits | |
|---|---|---|---|
| 1 | US (85°W - 110°W, 35°N - 44°N) | 0.44 | 1.14 |
| 2 | Brazil (34°W- 49°W, 24°S - 3°S) | 3.04 | 7.53 |
| 3 | European Union (9°W - 45°E, 35°N - 55°N) | 0.3 | 1.37 |
| 4 | Central Africa (14°W - 45°E, 0° - 14°N) | 3.24 | 7.19 |
| 5 | India (75°E - 90°E, 8°N - 30°N) | 2.27 | 4.63 |
| 6 | China (110°E - 125°E, 30°N - 42°N) | 1.14 | 1.91 |
| 7 | South Australia (134°E - 154°E, 38°S - 28°S) | 1.03 | 3.28 |

[Figure]

Figure 10: Spatial distribution of mean tropospheric column HCHO/NO$_2$ (FNR) obtained from
ECHAM6-HAMMOZ CTL simulations (2005-2020) and OMI (2005-2020).

To enhance our confidence in the model estimations, we compared the model-estimated FNR with the OMI-derived FNR for the period 2005-2020. Figure 10 illustrates the comparison of FNR

estimated from ECHAM6-HAMMOZ CTL simulations with OMI. The spatial map of FNR shows fairly good agreement between OMI and the model. Over the urbanized regions (e.g., South Asia,

Europe, the US, and China) both the model and OMI show FNR < 4. In contrast, regions like North

Canada, South America, central Africa, Australia, and Siberia exhibit high FNR values >9.

Although there is good agreement of the model simulations with OMI, some minor differences are seen between the model and OMI FNR over the west coast of South America, South Africa, the

Tibetan Plateau, and western Australia. These differences could be due to the underestimation of

HCHO in the model over these regions. Considering the fair performance of the model in comparison with OMI, we further analyzed the influence of NOx and VOC emissions on the FNR

based on the model simulations, which are discussed in the subsequent sections.

Fig. 11 shows the spatial distribution of estimates of FNRs from CTL, DNOx, DVOC,

HNOx, and HVOC simulations. In the control simulation for the period 1998-2020, most of the polluted cities/industrialized areas in the US, Canada, Europe, west Russia, East China, Korea and

Japan are VOC limited (FNRs <2). The NOx-limited regimes (largest FNR values >5) are found over the tropical rainforest, savanna, and arid climates clearly reflect the rural or unpolluted background regions where large biogenic emissions of VOCs are high (e.g., Shen et al., 2019;

Millet et al., 2008) (see Table 3 and central Africa in Fig. 10f). The DNOx simulation yields a significant shift in the spatial extent of VOC-limited regimes (Fig. 10bS). Regions across the NH

exhibit VOC-limited regimes, except central Africa, Amazonia, and north Australia. Notably, the

SH exhibits minimal change in the spatial extent of VOC-limited regimes with consistent occurrences over the western coastlines of South America, Argentina, Brazil, South Africa, and southern Australia.

The DVOC simulations show (Fig. 11c) a persistent occurrence of VOC-limited regimes over

Western Europe (e.g., the UK). The moderate FNR values (1-6) prevail across most of the NH, indicating a transition or NOx-limited regime. The spatial distribution of FNR in the SH is similar to that of the control simulation. In Figure 3b-c, the increase in ozone in response to a decrease in

NOx and an increase in VOC is attributed to the existence of a VOC-limited regime over these regions. The Indo-Gangetic Plains, Eastern China and the eastern United States clearly indicate the VOC-limited condition. The comparison of CTL and HNOx simulation (Fig. 10d) shows the transition from VOC-limited regimes to NOx-limited regimes occurring globally.

The FNR distribution for HVOC simulations is similar to CTL (as depicted in Fig. 10e) without any notable change in the spatial pattern. This suggests that ozone photochemistry exhibits less sensitivity to halved VOC emissions. Figure 11 clearly depicts that DNOx and HNOx simulations greatly impact the shift in ozone photochemical regimes compared with DVOC and HVOC

simulations. This indicates that ozone photochemistry is highly sensitive to changes in NOx emissions globally.

[Figure]

Figure 11: Spatial distribution of monthly mean tropospheric column HCHO/NO$_2$ (FNR) obtained from ECHAM6-HAMMOZ simulations (1998-2020) for (a) CTL, (b) DNOx, (c) DVOC, (d) HNOx, and (e) HVOC simulations. (f) Box and whisker plot illustrating the long-term average FNR over the regions depicted in Fig.7. Box represents 25 and 75 percentile and whisker represents 5 and 95 percentile. The black spherical marker represents the mean and the horizontal bar represents the 1 and 99 percentile.

**5. **Seasonal variation of Formaldehyde to Nitrogen dioxide Ratio**

Since the emission of HCHO and NO$_2$ varies significantly with the seasons across the globe (e.g.,

Smedt et al., 2015; Kumar et al., 2020; Goldberg et al., 2021; Wang et al., 2017; Surl et al., 2018;

Guan et al., 2021), understanding the seasonal changes in FNR is also crucial for comprehending shifts in ozone photochemical regimes. In this regard, using the methodology described in Section

4, we extracted the seasonal changes in transition limits for the major urban and semi-urban regions shown in Figure 8 and summarized in Table 4. Figure 12 illustrates the seasonal variation of estimated FNR from both OMI data and model simulations across these key urban regions. In general, all regions exhibit distinct seasonal variations in transition limits (Table 4). Previously reported transition limits over the US (2-5 : Johnson et al., 2024; 1.1-4 : Schroeder et al., 2017)

and China .6-1.5/1.25-2.39 (Chen et al., 2023) during summer season are also compared with our model estimates. The estimated FNR values from the ECHAM6-HAMOZ simulations shows fair agreement over both the locations (0.4-4.6 at US and 0.58-2.56 at China) with some minor differences. These minor discrepancy in the estimated FNR could be due to difference in the chosen location, time period and dataset used. Chen et al. (2023) has also reported that the transition limits significantly depends on the region considered for the analysis.

Table 4. Seasonal mean estimated values of the tropospheric $HCHO/NO_2$ columns threshold ratios from ECHAM6-HAMMOZ model control simulation to identify the NOx and VOC sensitive regimes across regions mentioned in Figure 8. The FNR less than the lower limit indicates VOC- limited, and that higher than the upper limit indicates NOx-limited regimes.

| Sr. No. | Regions | Transition limits | | | | | | | |
|---------|---------|------|------|------|------|------|------|------|------|
| | | DJF | | MAM | | JJA | | SON | |
| 1 | US | 0.48 | 1.04 | 0.49 | 1.15 | 0.49 | 4.69 | 0.45 | 1.39 |
| 2 | Brazil | 2.93 | 7.79 | 2.93 | 6.66 | 2.93 | 6.02 | 3.12 | 8.44 |
| 3 | European Union | 0.33 | 1.13 | 0.33 | 1.17 | 0.33 | 3.32 | 0.3 | 1.45 |
| 4 | Central Africa | 2.95 | 7.26 | 2.92 | 5.66 | 2.93 | 6.56 | 3.14 | 7.06 |
| 5 | India | 2.23 | 3.91 | 2.22 | 9.19 | 2.22 | 5.76 | 2.27 | 5.29 |
| 6 | China | 0.56 | 1.85 | 0.57 | 1.86 | 0.58 | 2.56 | 1.14 | 2.01 |
| 7 | South Australia | 1.1 | 5.54 | 1.09 | 2.3 | 1.09 | 1.82 | 1.12 | 3.93 |

Based on the threshold values depicted in Table 4 and the mean FNR in Figure 12, the seasonal change in ozone photochemical regimes over the key regions associated with the different emission scenarios are assessed. In the CTL simulation (Figure 12e-h), the US, Europe, and China are found to be in the transition regime, while all other regions are NOx-limited during winter. In spring every region except India remains NOx-limited, with India transitioning into the transition regime.

During summer and autumn, all regions shift to a NOx-limited condition. We further compared the model-estimated regional FNR from the CTL simulation with the OMI-derived FNR shown in

Figure 12a-d. The ozone photochemical regimes inferred from both OMI and the model show consistent results except during winter. During winter, US, Europe and China are NOx limited in

OMI and our model shows them as the transition regimes.

Doubling NOx (DNOx) leads to a shift to a VOC-limited regime in all regions except Africa and

Australia during winter, spring, and autumn (Figure 11i-l). The relatively high VOC contributions in Africa and Australia likely keep these regions in the transition regime. During summer, the US,

Europe, Africa and Australia transform to the transition regimes, while all other regions remain

VOC-limited. In both the DVOC and HNOx scenarios (Figure 11m-t), ozone photochemical regimes show no seasonality. All regions consistently exhibit a NOx-limited regime throughout all seasons. In the HVOC simulation (Figure 11u-x), the US, Europe, and China are in transition regimes, while all other regions become NOx-limited during winter. India remains in a transition regime during all other seasons, whereas other regions consistently exhibit NOx-limited conditions.

[Figure]

Figure 12: Box and whisker plot illustrating the long-term seasonal average FNR over the regions depicted in Fig.7. Box represents 25 and 75 percentile and whisker represents 5 and 95 percentile. The plus marker represents the mean and the horizontal bar represents the 1 and 99 percentile.

**6. Influence of NOx and VOCs emissions on trends of Formaldehyde to Nitrogen dioxide Ratio**

To understand the temporal evolution of ozone photochemical regimes associated with different emission scenarios, trend analysis is carried out on FNR. Figure 13 illustrates trends of FNR during the period 1998-2020 from CTL, DNOx, DVOC, HNOx, and HVOC simulations. In CTL

simulation, decreasing (negative) trends in FNR are seen over the Asian region (0.4-1.2 decade$^{-1}$)

Sand an increasing (positive) trend in Europe (0.2 decade$^{-1}$) and the US (0.8-1.4 decade$^{-1}$) (Fig.

13a). These observed trends in FNR are mainly driven by the region-specific trends on HCHO and

NO$_2$ (Figure 6). Figure 6 shows a higher positive trend in NO$_2$ than in HCHO in the Asia region, causing an overall decreasing trend in FNR, indicating a tendency towards VOC-limited regimes.

Whereas, over the US and Europe, there is a higher negative trend in NO$_2$ than HCHO, causing a positive trend in FNR. indicating a tendency towards a NOx-limited regime. A recent study by

Elshorbany et al. (2024) also reported a significant positive trend over Europe and the US and a negative trend over Asia using the OMI-based tropospheric column - HCHO/NO$_2$ ratio. Further, lLong-term column measurements of HCHO and NO$_2$ from OMI over India and China have revealed an increasing trend in NO$_2$ compared to that of HCHO, causing a decreasing trend in FNR

over these regions (Mahajan et al., 2015; Jin and Holloway, 2015).

DNOx simulation (Figure 13b) shows a similar pattern in spatial trend as that of CTL simulation (Fig 13a). However, the magnitude of this trend is less than that of the CTL. For example, a weak positive trend is noted in the US and Europe (0.2-0.4 decade$^{-1}$), while trends over India, and China are negative (0.2 - 0.4 decade$^{-1}$) in DNOx than CTL. (Fig. 12b). On the contrary, the magnitude of positive trend over Canada and negative trend over central Africa increased in DNOx emission.

This indicates that Canada and central Africa have a tendency to become NOx-limited and VOC- limited, respectively.

In DVOC simulations, trends are increasing over the US, Canada, and Europe compared to the

CTL (Fig. 13a and 13c). A notable change is observed over the Middle East and Amazon, where trends become more negative and positive, respectively, compared to CTL. While, the negative trends over Australia in the CTL become positive in the DVOC simulation. In HNOx simulations (Fig. 13d), the positive trends are higher over the US, Europe and Amazon, while negative trends prevail over India, China and Australia. The observed global trends are relatively stronger in the

HNOx simulation compared to all other simulations (Fig. 13). Meanwhile, in HVOC simulation, marginal changes are noted globally compared to CTL. The most pronounced change in the FNR

trend is observed over West Australia, where the negative trend in CTL becomes increasingly positive in HVOC (Fig. 13e). Figure 11f clearly shows that the trend in FNR is always negative over India and China for all the simulations, indicating that these regions have a tendency to become VOC-limited, while the positive trends over Europe, North America and Amazon show a tendency to become more NOx-limited. Further, Figures 5 and 12 show that the relation between trends in FNR and ozone exhibits a nonlinearity. For example, even though FNR shows a negative trend over India and China for all the simulations, the TCO trend depends on the specific emission scenario.

[Figure]

Figure 13: Trends in the lower tropospheric (surface - 700 hPa) HCHO/NO$_2$ ratio during 1998 -

2020 from ECHAM6-HAMMOZ simulations for (a) for CTL, (b) DNOx, (c) DVOC, (d) HNOx, (e) HVOC simulations. The stippled region indicates the trend significant at 95% confidence. (f)

scatter plot illustrating the long-term trend and standard deviation over the regions depicted in

Fig.9.

6. **Tropospheric ozone radiative effects**

The impact of emission changes on the tropospheric ozone radiative effect (TO3RE) is estimated using the ECHAM6 model output and a radiative kernel method (see data and model experiments).

The estimated TO3RE for different model simulations are shown in Figure 14. In the CTL

simulations (Fig. 14a), high TO3RE is noted over North Africa and the Middle East region in NH

(2.2 $W.m^{-2}$, while in SH, it is over Australia and South Africa (1.2 $W.m^{-2}$). The global mean area weighted average TO3RE estimated from the CTL simulation is 1.21 $W.m^{-2}$ (1998-2020, WMO

tropopause). TO3RE estimates from TES measurements (2005-2009) also show a peak of 1.0

$W.m^{-2}$ in northern Africa, the Mediterranean, and the Middle East in June–July–August (Bowman et al. 2013). Recently, Pope et al. (2024) have reported TO3RE estimates from IASI-SOFID, IASI-

FORLI, and IASI-IMS for the period 2008 - 2017. The values reported by Pope et al (2024) are comparable with our CTL simulation (e.g. IASI-FORLI: 1.23 $W.m^{-2}$, IASI-SOFRID: $1.21 W.m^{-2}$,

IASI-IMS: 1.21 $W.m^{-2}$). The minor differences in the estimated global mean TO3RE from the model and satellites are due to different time periods of observations/simulations.

The anomalies of TO3RE from DNOx-CTL simulations are shown in Figure 14b. Doubling of

NOx emission causes an enhancement in TO3RE by 0.36 $W.m^{-2}$ compared to the CTL simulation.

It shows a peak over the Middle East and adjacent North Africa (0.7 $W.m^{-2}$). A similar peak over this region is also seen in the CTL simulation. Doubling of VOC emissions enhances global mean

TO3RE by 0.01 $W.m^{-2}$, which is smaller than the doubling of NOx (Fig. 14b and 14c). TCO

enhancement for doubling NOx is also higher than doubling VOC (see Fig.3). DVOC-CTL

simulations (Fig. 14 c) show a peak over the Arctic (0.02 $W.m^{-2}$). The TO3RE anomalies are negative between 30°N-30°S. The negative anomalies in TO3RE between 30°S-30°N (Fig. 14c)

can be attributed to negative anomalies of TCO (Fig. 3h).

The reduction of NOx emission by 50% reduced global mean TO3RE by -0.12 $W.m^{-2}$ than CTL

(see table 3). The anomalies in TO3RE from HNOx-CTL simulations (Fig. 14d) show negative anomalies all over the globe, with a strong decrease over the Middle East and adjacent North Africa (-0.25 $W.m^{-2}$). Figures 14b and 14d show that the effect of enhancement/reduction of NOx emission is high over the Middle East and adjacent North Africa. The reduction of VOC emission by 50% reduced global mean TO3RE by -0.03 $W.m^{-2}$ than CTL simulations (Fig. 14e). HVOC -

CTL simulations show negative anomalies of TO3RE between 40°S - 40°N and positive 0.015

$W.m^{-2}$ (low confidence) over mid-high latitudes in NH and SH. From Figure 14, it is interesting to note that the magnitude of TO3RE and its response to emission change is pronounced over the

Middle East compared to all other regions. Figure 14f indicated that impacts of NOx emission changes are larger than VOC.

[Figure]

Figure 14: Tropospheric Ozone radiative effects (TO3RE) $(W.m^{-2})$ for (a) CTL, (b) anomalies from DNOx - CTL, (c) anomalies from DVOC - CTL, (d) anomalies from HNOx-CTL,(e) anomalies from HVOC - CTL simulations. Stippled regions in Figures (b-e) indicate RE significant at 95 % confidence level, (f) zonal mean TO3RE $(W.m^{-2})$ from CTL, DNOx - CTL, DVOC - CTL, HNOx-CTL, HVOC - CTL, shades indicates standard deviation.

**7 Conclusions**

In this study we report variation of tropospheric ozone levels, trends, photochemical regimes and radiative effect using the state-of-the-art ECHAM6-HAMMOZ chemistry-climate model simulations from 1998 to 2020. The model simulations are validated against multiple satellite observations. Our analysis shows that (1) The model underestimates global mean TCO by 15.3 ppb than OMI/MLS, by 1.7 ppb than IASI-SOFRID.

(2) The estimated global mean trend in TCO from CTL for the period 1998-2020 is 0.94 [-0.91;4.5] ppb.decade$^{-1}$. Trend estimates from OMI/MLS (1.41 [-2.15 4.54] ppb.decade$^{-1}$) for the period Oct 2004 to Dec 2020 show good agreement with CTL (1.86 [-2.7;4.6] ppb.decade$^{-1}$) for the same period. It has to be noted that IASI-SOFRID documents slightly negative trends (0.01 [-3.9 ;7.1] ppb.decade$^{-1}$) over the globe. The trends discrepancy between UV-Vis (mostly positive trends) and IR sensors (negative trends) was already documented in Gaudel et al. (2018).

(3) DNOx-CTL simulations show positive trends are seen over Europe, the US, Africa, and South America, with a global mean increase in TCO trend by 1.23 [-5.32; 5.76] ppb.decade$^{-1}$ and negative trends in surface ozone and TCO (-0.4 to -2 ppb.decade$^{-1}$) over India, China, and Australia.

(4) Compared to DNOx-CTL, DVOC-CTL simulations show a marginal enhancement in TCO

global mean trend (by 0.5 [-0.85; 1.93] ppb.decade$^{-1}$). HNOx - CTL and HVOC - CTL

simulations show decreases in the global mean TCO trends by 0.38 [-1.96; 1.64] ppb.decade$^{-1}$

and 0.42 [-0.96;1.61] ppb.decade$^{-1}$, respectively.

(5) The spatial distribution of ozone anomalies shows that enhancement in ozone is nearly 16

times less in DVOC simulation than that of doubling NO$_X$ simulation. The largest increase in surface ozone concentration for DVOC is observed over Indo-Gangetic Plains, Eastern China and the eastern United States (4-6 ppb), where a decrease in ozone is observed in the DNOx simulation. This decrease (increase) in ozone with an increase in NOx (VOC) indicates that these regions are VOC-limited.

(6) The FNR over the major urban and semi-urban regions shows that the transition from

VOC-limited to NOx-limited happens suddenly in response to changes in the emission of

NOx/VOC over the US and China. Whereas this transition region is wider in Central Africa.

Most polluted cities/industrialized areas in the US, Canada, Europe, west Russia, East China,

Korea and Japan are identified with a low FNR, indicating VOC limited (FNRs <2).

Meanwhile, NOx-limited regimes (largest FNR values >5) are primarily found in tropical rainforests, savannas, and arid climates.

(7) The DNOx simulation shows a notable change in the spatial extent of VOC-limited regimes, particularly in the Northern Hemisphere (NH). While the southern hemisphere (SH)

exhibits minimal change in the spatial extent of VOC-limited regimes.

(8) DVOC simulations reveal persistent VOC-limited regimes over Western Europe, with moderate FNR values indicating a transition to NOx-limited regimes across most of the

Northern Hemisphere. Comparing CTL and HNOx simulations globally shows a shift from

VOC to NOx-limited regimes.

(9) Comparison of all the emission simulations, DNOx and HNOx simulations significantly influence the shift in ozone photochemical regimes compared to DVOC and HVOC

simulations, highlighting the global sensitivity of ozone photochemistry to NOx emissions changes.

(10) Trends estimated from modeled FNR are negative over India and China in all the simulations, indicating that these regions have a tendency to become VOC-limited, while the positive trends over Europe, North America and Amazon, indicating a tendency to become more NOx-limited.

(11) The trends in FNR are negative over India and China in all simulations. However, the trends in TCO are positive in DVOC - CTL and HVOC - CTL simulations and negative in

DNOx - CTL and HNOx - CTL .

(12) The tropospheric ozone radiative effects (TO3RE) in DNOx - CTL and DVOC - CTL

show an increase in TO3RE by 0.36 $W.m^{-2}$ and 0.01 $W.m^{-2}$ respectively. However, HNOx-

CTL and HVOC-CTL show reduction in the global mean TO3RE by -0.12 $W.m^{-2}$ and -0.03

$W.m^{-2,}$ respectively.

(13) We show that anthropogenic NOx emissions have a higher impact on tropospheric ozone levels, trends and radiative effect than VOC emissions globally.

**Author's contribution**: SF and YE initiated the manuscript. SF made the model simulations.

VS and SC did analysis. satellite data sets are provided by JZ, BB, EF, IG, ID,  MR, IS. All authors contributed to writing.

**Data availability**

Available from the TOAR FTP server

**Code availability**

Available from the corresponding author upon reasonable request.

**References:**

Acdan, J. J. M., Pierce, R. B., Dickens, A. F., Adelman, Z., and Nergui, T.: Examining TROPOMI

formaldehyde to nitrogen dioxide ratios in the Lake Michigan region: implications for ozone exceedances, Atmos. Chem. Phys., 23, 7867–7885, https://doi.org/10.5194/acp-23-7867-2023,

2023.

Adame J.A., Gutierrez-Alvarez I., Cristofanelli P. and Notario A. and  Bogeat J.A.and Bolivar

J.P.and  Yela M., Surface ozone trends at El Arenosillo observatory from a new perspective,

Environmental Research, Volume 214, Part 1, November 2022, 113887, DOI:

10.1016/j.envres.2022.113887

Alvarado, L. M. A., Richter, A., Vrekoussis, M., Hilboll, A., Kalisz Hedegaard, A. B., Schneising,

O., and Burrows, J. P.: Unexpected long-range transport of glyoxal and formaldehyde observed from the Copernicus Sentinel-5 Precursor satellite during the 2018 Canadian wildfires, Atmos.

Chem. Phys., 20, 2057–2072, https://doi.org/10.5194/acp-20-2057-2020, 2020.

Anderson, D. C., Nicely, J. M., Wolfe, G. M., Hanisco, T. F., Salawitch, R. J., Canty, T. P., Zeng,

G. (2017). Formaldehyde in the tropical western Pacific: Chemical sources and sinks, convective transport, and representation in CAM-Chem and the CCMI models. *Journal of Geophysical*

*Research: Atmospheres*, 122, 11,201–11,226. https://doi.org/10.1002/2016JD026121

Barret, B., Gouzenes, Y., Le Flochmoen, E., & Ferrant, S. (2021). Retrieval of Metop-A/IASI N2O
profiles and validation with NDACC FTIR data. Atmosphere, 12(2), 219.

Barret, B., Le Flochmoen, E., Sauvage, B., Pavelin, E., Matricardi, M., & Cammas, J. P. (2011).
The detection of post-monsoon tropospheric ozone variability over south Asia using IASI data.
Atmospheric Chemistry and Physics, 11(18), 9533-9548.

Beig, G., & Singh, V. (2007). Trends in tropical tropospheric column ozone from satellite data and
MOZART model. Geophysical research letters, 34(17).

Borbas, E. E. and B. C. Ruston, 2010. The RTTOV UWiremis IR land surface emissivity module.
Mission Report EUMETSAT NWPSAF-MO-VS-042. http://nwpsaf.eu/vs_reports/nwpsaf-mo-vs-
042.pdf

Bowman, K. W., Shindell, D. T., Worden, H. M., Lamarque, J. F., Young, P. J., Stevenson, D. S.,
Qu, Z., de la Torre, M., Bergmann, D., Cameron-Smith, P. J., Collins, W. J., Doherty, R., Dalsøren,
S. B., Faluvegi, G., Folberth, G., Horowitz, L. W., Josse, B. M., Lee, Y. H., MacKenzie, I. A.,
Myhre, G., Nagashima, T., Naik, V., Plummer, D. A., Rumbold, S. T., Skeie, R. B., Strode, S. A.,
Sudo, K., Szopa, S., Voulgarakis, A., Zeng, G., Kulawik, S. S., Aghedo, A. M., and Worden, J. R.:
Evaluation of ACCMIP outgoing longwave radiation from tropospheric ozone using TES satellite
observations, Atmos. Chem. Phys., 13, 4057– 4072, https://doi.org/10.5194/acp-13-4057-2013,
2013.

Chang, C. -Y., Faust, E., Hou, X., Lee, P., Kim, H. C., Hedquist, B. C., and Liao, K. -J.:
Investigating ambient ozone formation regimes in neighboring cities of shale plays in the northeast
United States using photochemical modeling and satellite retrievals, Atmos. Environ., 142, 152–
170. doi:10.1016/j.atmosenv.2016.06.058, 2016.

Chen, Y., Wang, M., Yao, Y., Zeng, C., Zhang, W., Yan, H., Gao P., Fan L., Ye, D. (2023).
Research on the ozone formation sensitivity indicator of four urban agglomerations of China using
Ozone Monitoring Instrument (OMI) satellite data and ground-based measurements. *Science of*
*The Total Environment*, *869*, 161679.

Chu Sophie N., Sands S, Tomasik M. R., Lee P. S., and McNeill V. F., Ozone
Oxidation of Surface-Adsorbed Polycyclic Aromatic Hydrocarbons:
Role of PAH-Surface Interaction, J. Am. Chem. Soc. 2010, 132, 45, 15968–
15975, https://doi.org/10.1021/ja1014772

Cohen, Y., Petetin, H., Thouret, V., Marécal, V., Josse, B., Clark, H., Sauvage, B., Fontaine, A.,
Athier, G., Blot, R., Boulanger, D., Cousin, J.-M., and Nédélec, P.: Climatology and long-term evolution of ozone and carbon monoxide in the upper troposphere–lower stratosphere (UTLS) at
northern midlatitudes, as seen by IAGOS from 1995 to 2013, Atmos. Chem. Phys., 18, 5415–5453,
https://doi.org/10.5194/acp-18-5415-2018, 2018.

Cooper O. R, Parrish D. D., Ziemke J., Balashov N. V., Cupeiro M.,. Galbally I. E, Gilge S.,
Horowitz L, Jensen N. R., Lamarque J.-F., Naik V., Oltmans S. J., Schwab J., Shindell D. T.,
hompsonA. M. T, Thouret V., Wang Y., Zbinden R. M., Global distribution and trends of
tropospheric ozone: An observation-based review, Collections: Knowledge Domain: Atmospheric
Science, *Elementa: Science of the Anthropocene* (2014) 2: 000029,
https://doi.org/10.12952/journal.elementa.000029

Cuesta, J., Costantino, L., Beekmann, M., Siour, G., Menut, L., Bessagnet, B., ... & Eremenko, M.
(2022). Ozone pollution during the COVID-19 lockdown in the spring of 2020 over Europe,
analysed from satellite observations, in situ measurements, and models. *Atmospheric Chemistry*
*and Physics*, *22*(7), 4471-4489.

Cuesta, J., Eremenko, M., Liu, X., Dufour, G., Cai, Z., Höpfner, M., ... & Flaud, J. M. (2013).
Satellite observation of lowermost tropospheric ozone by multispectral synergism of IASI thermal
infrared and GOME-2 ultraviolet measurements over Europe. Atmospheric Chemistry and
Physics, 13(19), 9675-9693.

Cuesta, J., Eremenko, M., Liu, X., Dufour, G., Cai, Z., Höpfner, M., ... & Flaud, J. M. (2013).
Satellite observation of lowermost tropospheric ozone by multispectral synergism of IASI thermal
infrared and GOME-2 ultraviolet measurements over Europe. *Atmospheric Chemistry and Physics*,
*13*(19), 9675-9693.

Cuesta, J., Kanaya, Y., Takigawa, M., Dufour, G., Eremenko, M., Foret, G., ... & Beekmann, M.
(2018). Transboundary ozone pollution across East Asia: daily evolution and photochemical
production analysed by IASI+ GOME2 multispectral satellite observations and models.
*Atmospheric Chemistry and Physics*, *18*(13), 9499-9525.

De Smedt, I., Stavrakou, T., Hendrick, F., Danckaert, T., Vlemmix, T., Pinardi, G., ... & Van
Roozendael, M. (2015). Diurnal, seasonal and long-term variations of global formaldehyde
columns inferred from combined OMI and GOME-2 observations. *Atmospheric Chemistry and*
*Physics*, *15*(21), 12519-12545.

Domenico Taraborrelli, David Cabrera-Perez2 , Sara Bacer , Sergey Gromov , Jos Lelieveld , Rolf
Sander , and Andrea PozzerI, Influence of aromatics on tropospheric gas-phase composition ,
Atmos. Chem. Phys., 21, 2615–2636, 2021 https://doi.org/10.5194/acp-21-2615-2021

Duncan, B. N., Lamsal, L. N., Thompson, A. M., Yoshida, Y., Lu, Z., Streets, D. G. Pickering, K.
E. (2016). A space-based, high-resolution view of notable changes in urban NOx pollution around
the world (2005–2014). Journal of Geophysical Research: Atmospheres, 121, 976–996.

Duncan, B., Yoshida, Y., Olson, J., Sillman, S., Martin, R., Lamsal, L., Hu, Y., Pickering, K.
Retscher, D. Allen, D., and Crawford, J.: Application of OMI observations to a space-based
indicator of NOx and VOC controls on surface ozone formation, Atmos. Environ., 44, 2213–2223,
https://doi.org/10.1016/j.atmosenv.2010.03.010, 2010.

Edwards, J. M. and Slingo, A.: Studies with a flexible new radiation code. I: Choosing a
configuration for a large scale model, Q. J. Roy. Meteor. Soc., 122, 689–719,
https://doi.org/10.1002/qj.49712253107, 1996.

Elshorbany, Y., Ziemke, J., Strode, S., Petetin, H., Miyazaki, K., De Smedt, I., ... & Huang, M.
(2024). Tropospheric Ozone Precursors: Global and Regional Distributions, Trends and
Variability. EGU sphere, 2024, 1-57.

Emmons, L. K., Apel, E. C., Lamarque, J.-F., Hess, P. G., Avery, M., Blake, D., Brune, W.,
Campos, T., Crawford, J., DeCarlo, P. F., Hall, S., Heikes, B., Holloway, J., Jimenez, J. L., Knapp,
D. J., Kok, G., Mena-Carrasco, M., Olson, J., O'Sullivan, D., Sachse, G., Walega, J., Weibring, P.,
Weinheimer, A., and Wiedinmyer, C.: Impact of Mexico City emissions on regional air quality
from MOZART-4 simulations, Atmos. Chem. Phys., 10, 6195–6212, https://doi.org/10.5194/acp-
10-6195-2010, 2010.

EPA, O. U. Air Pollutant Emissions Trends Data. 2023. https://www.epa.gov/air-emissions-
inventories/air-pollutant-emissions-trends-data

Eyring, V., J. M. Arblaster, I. Cionni, J. Sedláček, J. Perlwitz, P. J. Young, S. Bekki, D. J.
Bergmann, P. Cameron-Smith, W. J. Collins, G. Faluvegi, D. Shindell, *et al.* (2013), Long-term
ozone changes and associated climate impacts in CMIP5 simulations, *J. Geophys. Res., 118*, 5029-
5060, doi:10.1002/jgrd.50316

Fadnavis S., Chavan P., Joshi A., Sonbawne S.M., Acharya A., Devara P.C.S., Rap A., Ploeger F.,
Müller R., Tropospheric warming over the northern Indian Ocean caused by South Asian
anthropogenic aerosols: possible impact on the upper troposphere and lower stratosphere,
Atmospheric Chemistry and Physics, 22, June 2022, DOI:10.5194/acp-22-7179-2022 , 7179–719.

Fadnavis S., Heinold B., Sabin T.P., Kubin A., Huang K., Rap A., Müller R., Air pollution
reductions caused by the COVID-19 lockdown open up a way to preserve the Himalayan glaciers.,
Atmospheric Chemistry and Physics, 23, September 2023, DOI:10.5194/acp-23-10439-2023,
10439-10449

Fadnavis S., Müller R., Chakraborty T., Sabin T.P., Laakso A., Rap A., Griessbach S., Vernier
J.V., Tilmes S., The role of tropical volcanic eruptions in exacerbating Indian droughts, Scientific
Reports, 11: 2714, 2021a, DOI:10.1038/s41598-021-81566-0, 1-13.,

Fadnavis S., Müller R., Kalita G., Rowlinson M., Rap A., Li J-L.F., Gasparini B., Laakso A.,
Impact of recent changes in Asian anthropogenic emissions of SO2 on sulfate loading in the upper
troposphere and lower stratosphere and the associated radiative changes, Atmospheric Chemistry
and Physics, 19, 2019a, DOI:10.5194/acp-19-9989-2019, 9989–10008.

Fadnavis S., Sabin T.P., Rap A., Müller R., Kubin A., Heinold B., The impact of COVID-19
lockdown measures on the Indian summer monsoon , Environmental Research Letters, 16: 074054,
July 2021b, DOI: 10.1088/1748-9326/ac109c, 1-13

Fadnavis S., Sabin T.P., Roy C., Rowlinson M., Rap A., Vernier J.-P., Sioris C.E., Elevated aerosol
layer over South Asia worsens the Indian droughts, Scientific Reports, 9:10268, 2019b,
DOI:10.1038/s41598-019-46704-9, 1-11

Fiore Arlene Met al 2022 Understanding recent tropospheric ozone trends in the context of large
internal variability: a new perspective from chemistry-climate model ensembles, Environ. Res.:
Climate 1 025008, DOI 10.1088/2752-5295/ac9cc2

Fiore, A. M., Hancock, S. E., Lamarque, J. F., Correa, G. P., Chang, K. L., Ru, M., ... & Ziemke,
J. R. (2022). Understanding recent tropospheric ozone trends in the context of large internal
variability: a new perspective from chemistry-climate model ensembles. Environmental Research:
Climate, 1(2), 025008.

Fleming, Z. L., Doherty, R. M., Von Schneidemesser, E., Malley, C. S., Cooper, O. R., Pinto, J.
P., ... & Feng, Z. (2018). Tropospheric Ozone Assessment Report: Present-day ozone distribution
and trends relevant to human health. Elem Sci Anth, 6, 12.

Gaudel, A., Cooper, O. R., Ancellet, G., Barret, B., Boynard, A., Burrows, J. P., ... & Ziemke, J.
(2018). Tropospheric Ozone Assessment Report: Present-day distribution and trends of
tropospheric ozone relevant to climate and global atmospheric chemistry model evaluation. Elem
Sci Anth, 6, 39.

Gaudel, A., I. Bourgeois, M. Li, O. Cooper, K.-L. Chang, B. Sauvage, J. R. Ziemke, A. Thompson,
R. Stauffer, and N. Smith, In-situ and satellite-based quantification of tropical tropospheric ozone,
Atmos. Chem. Phys., in review, 2024.

Glissenaar, I. A., Anglou, I., Boersma, K. F., & Eskes, H. (2024). ESA CCI+ TROPOMI L3 monthly mean NO2 columns [Data set]. Royal Netherlands Meteorological Institute (KNMI). https://doi.org/10.21944/CCI-NO2-TROPOMI-L3

Goldberg, D. L., Anenberg, S. C., Kerr, G. H., Mohegh, A., Lu, Z., and Streets, D. G.: TROPOMI NO2 in the United States: A Detailed Look at the Annual Averages, Weekly Cycles, Effects of Temperature, and Correlation With Surface NO2 Concentrations, Earth's Future, 9, e2020EF001665, https://doi.org/10.1029/2020EF001665, 2021

Griffiths, P. T., Murray, L. T., Zeng, G., Shin, Y. M., Abraham, N. L., Archibald, A. T., Deushi, M., Emmons, L. K., Galbally, I. E., Hassler, B., Horowitz, L. W., Keeble, J., Liu, J., Moeini, O., Naik, V., O'Connor, F. M., Oshima, N., Tarasick, D., Tilmes, S., Turnock, S. T., Wild, O., Young, P. J., and Zanis, P.: Tropospheric ozone in CMIP6 simulations, Atmos. Chem. Phys., 21, 4187–4218, https://doi.org/10.5194/acp-21-4187-2021, 2021.

Guan, J., Jin, B., Ding, Y., Wang, W., Li, G., & Ciren, P. (2021). Global surface HCHO distribution derived from satellite observations with neural networks technique. *Remote Sensing*, *13*(20), 4055

Guerreiro, C. B. B., Foltescu, V., & de Leeuw, F. (2014). Air quality status and trends in Europe. Atmospheric Environment, 98(c), 376–384. https://doi.org/10.1016/j.atmosenv.2014.09.017

Gulev, S. K., Thorne, P. W. (2021). Changing state of the climate system, https://doi.org/10.1002/2015JD024121

IPCC, 2021: *Climate Change 2021: The Physical Science Basis. Contribution of Working Group I to the Sixth Assessment Report of the Intergovernmental Panel on Climate Change*[Masson-Delmotte, V., P. Zhai, A. Pirani, S.L. Connors, C. Péan, S. Berger, N. Caud, Y. Chen, L. Goldfarb, M.I. Gomis, M. Huang, K. Leitzell, E. Lonnoy, J.B.R. Matthews, T.K. Maycock, T. Waterfield, O. Yelekçi, R. Yu, and B. Zhou (eds.)]. Cambridge University Press, Cambridge, United Kingdom and New York, NY, USA, In press, doi:10.1017/9781009157896.

Jin, X., & Holloway, T. (2015). Spatial and temporal variability of ozone sensitivity over China observed from the Ozone Monitoring Instrument. Journal of Geophysical Research: Atmospheres, 120(14), 7229-7246.

Jin, X., Fiore, A. M., Murray, L. T., Valin, L. C., Lamsal, L. N., Duncan, B., Boersma, K. F., De Smedt, I., Abad, G. G., Chance, K., and Tonnesen, G. S.: Evaluating a Space-Based Indicator of Surface Ozone-NOx-VOC Sensitivity Over Midlatitude Source Regions and Application to Decadal Trends, J. Geophys. Res.-Atmos., 122, 10439–10461, https://doi.org/10.1002/2017JD026720, 2017.

Jin, X., Fiore, A., Boersma, K. F., De Smedt, I., and Valin, L.: Inferring Changes in Summertime Surface Ozone–NOx–VOC Chemistry over U.S. Urban Areas from Two Decades of Satellite and Ground-Based Observations, Environ. Sci. Technol., 54, 6518–6529, https://doi.org/10.1021/acs.est.9b07785, 2020.

Johnson, M. S., Philip, S., Meech, S., Kumar, R., Sorek-Hamer, M., and Shiga, Y. P.: Insights into the long-term (2005–2021) spatiotemporal evolution of summer ozone production sensitivity in the Northern Hemisphere derived with OMI, EGUsphere [preprint], https://doi.org/10.5194/egusphere-2024-583, 2024.

Kinnison, D. E., Brasseur, G. P., Walters, S., Garcia, R. R., Marsh, D. R., Sassi, F., Harvey, V. L., Randall, C. E., Emmons, L., Lamarque, J. F., Hess, P., Orlando, J. J., Tie, X. X., Randel, W., Pan, L. L., Gettelman, A., Granier, C., Diehl, T., Niemeier, U., and Simmons, A. J.: Sensitivity of chemical tracers to meteorological parameters in the MOZART-3 chemical transport model, J. Geophys. Res.-Atmos., 112, 20302, https://doi.org/10.1029/2006JD007879, 2007

Kleinman, L. I. (1994), Low and high $NO_x$ tropospheric photochemistry, *J. Geophys. Res.*, 99(D8), 16831–16838, doi:10.1029/94JD01028.

Kumar, V., Beirle, S., Dörner, S., Mishra, A. K., Donner, S., Wang, Y., ... & Wagner, T. (2020). Long-term MAX-DOAS measurements of NO 2, HCHO, and aerosols and evaluation of corresponding satellite data products over Mohali in the Indo-Gangetic Plain. *Atmospheric Chemistry and Physics*, *20*(22), 14183-14235

Lamarque J. F. et al., 2010. Historical (1850–2000) gridded anthropogenic and biomass burning emissions of reactive gases and aerosols: methodology and application, Atmos. Chem. Phys., 10, 7017-7039, doi:10.5194/acp-10-7017-2010

Van Vuuren, D.P. et al., 2011. Representative concentration pathways: an overview. Climatic Change (2011) 109:5–31, DOI 10.1007/s10584-011-0148-z

Lamarque, J. F., Emmons, L. K., Hess, P. G., Kinnison, D. E., Tilmes, S., Vitt, F., ... & Tyndall, G. K. (2012). CAM-chem: Description and evaluation of interactive atmospheric chemistry in the Community Earth System Model. *Geoscientific Model Development*, *5*(2), 369-411.

Lee, H. J., Chang, L. S., Jaffe, D. A., Bak, J., Liu, X., Abad, G. G., ... & Kim, C. H. (2022). Satellite-based diagnosis and numerical verification of ozone formation regimes over nine megacities in East Asia. *Remote Sensing*, *14*(5), 1285.

Lefohn, A. S., Malley, C. S., Simon, H., Wells, B., Xu, X., Zhang, L., & Wang, T. (2017).
Responses of human health and vegetation exposure metrics to changes in ozone concentration
distributions in the European Union, United States, and China. Atmospheric Environment, 152,
123–145, https://doi.org/10.1016/j.atmosenv.2016.12.025

Lin M, Horowitz LW, Oltmans SJ, et al. Tropospheric ozone trends at Mauna Loa Observatory 18
494 tied to decadal climate variability. Nature Geosci, 2014, 7: 136-143 495

Lin, X., M. Trainer, and S. C. Liu (1988), On the nonlinearity of the tropospheric ozone production,
*J. Geophys. Res.*, 93(D12), 15879–15888, doi:10.1029/JD093iD12p15879.

Lu X, Zhang L, Liu X, et al. Lower tropospheric ozone over India and its linkage to the South 496
Asian monsoon. Atmos Chem Phys, 2018, 18: 3101-3118

Ma X., Huang J., Hegglin M. I., Jöckel P., and Zhao T.,  Causes of growing middle-upper
tropospheric ozone over the Northwest Pacific region, ACPD, 20024,
https://doi.org/10.5194/egusphere-2023-2411

Maddox E. M. and G. L. Mullendore Determination of Best Tropopause Definition for Convective
Transport Studies, Journal of the Atmospheric Sciences, 3433–3446, 2018,
https://doi.org/10.1175/JAS-D-18-0032.1

Mahajan, A. S., De Smedt, I., Biswas, M. S., Ghude, S., Fadnavis, S., Roy, C., & van Roozendael,
M. (2015). Inter-annual variations in satellite observations of nitrogen dioxide and formaldehyde
over India. Atmospheric Environment, 116, 194-201.

Martin, R. V., Fiore, A. M., and Van Donkelaar, A.: Space-based diagnosis of surface ozone
sensitivity to anthropogenic emissions, Geophys. Res. Lett., 31, L06120.
doi:10.1029/2004GL019416, 2004.

Matricardi, M., Chevallier, F., Kelly, G., & Thépaut, J. N. (2004). An improved general fast
radiative transfer model for the assimilation of radiance observations. Quarterly Journal of the
Royal Meteorological Society, 130(596), 153-173.

Millet, D. B., Jacob, D. J., Boersma, K. F., Fu, T. M., Kurosu, T. P., Chance, K., ... & Guenther,
A. (2008). Spatial distribution of isoprene emissions from North America derived from
formaldehyde column measurements by the OMI satellite sensor. Journal of Geophysical
Research: Atmospheres, 113(D2).

Mills, G., Pleijel, H., Malley, C. S., Sinha, B., Cooper, O. R., Schultz, M. G., ... & Xu, X. (2018).
Tropospheric Ozone Assessment Report: Present-day tropospheric ozone distribution and trends
relevant to vegetation. Elem Sci Anth, 6, 47.

Monks, P. S., Archibald, A. T., Colette, A., Cooper, O., Coyle, M., Derwent, R., Fowler, D.,
Granier, C., Law, K. S., Mills, G. E., Stevenson, D. S., Tarasova, O., Thouret, V., von
Schneidemesser, E., Sommariva, R., Wild, O., and Williams, M. L.: Tropospheric ozone and its
precursors from the urban to the global scale from air quality to short-lived climate forcer, Atmos.
Chem. Phys., 15, 8889–8973, https://doi.org/10.5194/acp-15-8889-2015, 2015.

O3 (IASI+GOME2) – IASI portal: https://iasi.aeris-data.fr/o3_iago2/, last access: 28 May 2024

Okamoto, S., Cuesta, J., Beekmann, M., Dufour, G., Eremenko, M., Miyazaki, K., Boonne C.,
Tanimoto H., Akimoto, H. (2023). Impact of different sources of precursors on an ozone pollution
outbreak over Europe analysed with IASI+GOME2 multispectral satellite observations and model
simulations. *Atmospheric Chemistry and Physics*, *23*(13), 7399-7423.

Pope R. J. Rap A., Pimlott M. A, Barret B., Flochmoen E. Le, Kerridge B. J., Richard Siddans,
Barry G. Latter, Lucy J. Ventress, Anne Boynard, Christian Retscher, Wuhu Feng, Richard
Rigby, Sandip S. Dhomse, Catherine Wespes, and Martyn P. Chipperfield, Quantifying the
tropospheric ozone radiative effect and its temporal evolution in the satellite era, Atmos. Chem.
Phys., 24, 3613–3626, 2024 https://doi.org/10.5194/acp-24-3613-2024

Pope, R. J., Savage, N. H., Chipperfield, M. P., Arnold, S. R., & Osborn, T. J. (2014). The influence
of synoptic weather regimes on UK air quality: analysis of satellite column NO2. *Atmospheric*
*Science Letters*, *15*(3), 211-217.

Rap, A., Richard, N. A. D., Forster, P. M., Monks, S. A., Arnold, S. R., and Chipperfield, M. P.:
Satellite constraint on the tropospheric ozone radiative effect, Geophys. Res. Lett., 42, 5074– 5081,
https://doi.org/10.1002/2015GL064037, 2015.

Reick, C. H., Raddatz, T., Brovkin, V., and Gayler, V.: Representation of natural and
anthropogenic land cover change in MPI-ESM, J. Adv. Model. Earth Sy., 5, 459–482,
https://doi.org/10.1002/jame.20022, 2013.

Ren, J., Guo, F., and Xie, S.: Diagnosing ozone–$NO_x$–VOC sensitivity and revealing causes of
ozone increases in China based on 2013–2021 satellite retrievals, Atmos. Chem. Phys., 22, 15035–
15047, https://doi.org/10.5194/acp-22-15035-2022, 2022.

Rosanka S , Franco B. , Clarisse L , Coheur P-F , Pozzer A. , Wahner A , and Taraborrelli D., The
impact of organic pollutants from Indonesian peatland fires on the tropospheric and lower stratospheric composition, Atmos. Chem. Phys., 21, 11257–11288, 2021 https://doi.org/10.5194/acp-21-11257-2021.

Rossow, W. B. and Schiffer, R. A.: Advances in understanding clouds from ISCCP, B. Am. Meteorol. Soc., 80, 2261–2287, https://doi.org/10.1175/1520-0477(1999)080>2.0.co;2,1999

Rowlinson, M. J., Rap, A., Hamilton, D. S., Pope, R. J., Hantson, S., Arnold, S. R., Kaplan, J. O., Arneth, A., Chipperfield, M. P., Forster, P. M., and Nieradzik, L.: Tropospheric ozone radiative forcing uncertainty due to pre-industrial fire and biogenic emissions Atmos. Chem. Phys., 19, 8669–8686, https://doi.org/10.5194/acp-19-8669-2019, 2019.

Rowlinson, M. J., Rap, A., Hamilton, D. S., Pope, R. J., Hantson, S., Arnold, S. R., Kaplan, J. O., Arneth, A., Chipperfield, M. P., Forster, P. M., and Nieradzik, L.: Tropospheric ozone radiative forcing uncertainty due to pre-industrial fire and biogenic emissions, Atmos. Chem. Phys., 20, 10937–10951, https://doi.org/10.5194/acp-20-10937-2020, 2020

Saunders, R., Matricardi, M., & Brunel, P. (1999). An improved fast radiative transfer model for assimilation of satellite radiance observations. Quarterly Journal of the Royal Meteorological Society, 125(556), 1407-1425.

Schroeder, J. R., Crawford, J. H., Fried, A., Walega, J., Weinheimer, A., Wisthaler, A., ... & Tonnesen, G. S. (2017). New insights into the column CH2O/NO2 ratio as an indicator of near-surface ozone sensitivity. Journal of Geophysical Research: Atmospheres, 122(16), 8885-8907.

Schultz, M. G., Schröder, S., Lyapina, O., Cooper, O. R., Galbally, I., Petropavlovskikh, I, Von Schneidemesser, E., Tanimoto, H., Elshorbany, Y., Naja, M., Seguel, R. J., Dauert, U., Eckhardt, P., Feigenspan, S., Fiebig, M., Hjellbrekke, A., Hong, Y., Kjeld, P. C., Koide, H., . . . Zhiqiang, M. (2016). Tropospheric Ozone Assessment Report: Database and metrics data of global surface ozone observations. *Elementa: Science of the Anthropocene*, 5. https://doi.org/10.1525/elementa.244

Schultz, M. G., Schröder, S., Lyapina, O., Cooper, O. R., Galbally, I., Petropavlovskikh, I., ... & Zhiqiang, M. (2017). Tropospheric Ozone Assessment Report: Database and metrics data of global surface ozone observations. Elem Sci Anth, 5, 58.

Schultz, M. G., Stadtler, S., Schröder, S., Taraborrelli, D., Franco, B., Krefting, J., Henrot, A., Ferrachat, S., Lohmann, U., Neubauer, D., Siegenthaler-Le Drian, C., Wahl, S., Kokkola, H., Kühn, T., Rast, S., Schmidt, H., Stier, P., Kinnison, D., Tyndall, G. S., Orlando, J. J., and Wespes, C.: The chemistry–climate model ECHAM6.3-HAM2.3-MOZ1.0, Geosci. Model Dev., 11, 1695–1723, https://doi.org/10.5194/gmd-11-1695-2018, 2018.

Scott, C. E., Monks, S. A., Spracklen, D. V., Arnold, S. R., Forster, P. M., Rap, A., Aijala, M.,
Artaxo, P., Carslaw, K. S., Chipperfield, M. P., Ehn, M., Gilardoni, S., Heikkinen, L., Kulmala,
M., Petaja, T., Reddington, C. L. S, Rizzo, L. V., Swietlicki, E., Vignati, E., and Wilson, C.: Impact
on short-lived climate forcers increases projected warming due to deforestation, Nat. Commun.,9,
157, https://doi.org/10.1038/s41467-017-02412-4, 2018

Shen, L., Jacob, D. J., Zhu, L., Zhang, Q., Zheng, B., Sulprizio, M. P., Li K.,  Smedt I. D.,  Abad G.
G.,  Cao H.,  Fu T.-M.,  Liao, H. (2019). The 2005–2016 trends of formaldehyde columns over
China observed by satellites: Increasing anthropogenic emissions of volatile organic compounds
and decreasing agricultural fire emissions. Geophysical Research Letters, 46(8), 4468-4475.

Sillman, S.: The use of NOy, H2O2, and HNO3 as indicators for O3-NOx-hydrocarbon sensitivity
in urban locations, J. Geophys. Res. Atmos., 100, 14175-14188, doi:10.1029/94JD02953, 1995.

Sillman, S., J. A. Logan, and S. C. Wofsy (1990), The sensitivity of ozone to nitrogen oxides and
hydrocarbons in regional ozone episodes, *J. Geophys. Res.*, 95(D2), 1837–1851,
doi:10.1029/JD095iD02p01837.

Skeie, R.B., Myhre, G., Hodnebrog, Ø. *et al.* Historical total ozone radiative forcing derived from
CMIP6 simulations. *npj Clim Atmos Sci* 3, 32 (2020). https://doi.org/10.1038/s41612-020-00131-
0

Sofieva, V. F., Tamminen, J., Kyrölä, E., Mielonen, T., Veefkind, P., Hassler, B., and Bodeker, G.
E.: A novel tropopause-related climatology of ozone profiles, Atmos. Chem. Phys., 14, 283–299,
https://doi.org/10.5194/acp-14-283-2014, 2014.

son, M. S., Souri, A. H., Philip, S., Kumar, R., Naeger, A., Geddes, J., Judd, L., Janz, S., Chong,
H., and Sullivan, J.: Satellite remote-sensing capability to assess tropospheric-column ratios of
formaldehyde and nitrogen dioxide: case study during the Long Island Sound Tropospheric Ozone
Study 2018 (LISTOS 2018) field campaign, Atmos. Meas. Tech., 16, 2431–2454,
https://doi.org/10.5194/amt-16-2431-2023, 2023.

Souri, A. H., Choi, Y., Jeon, W., Woo, J. -H., Zhang, Q., and Kurokawa J.-i.: Remote sensing
evidence of decadal changes in major tropospheric ozone precursors over East Asia, J. Geophys.
Res. Atmos., 122, 2474-2492, doi:10.1002/2016JD025663, 2017.

Stadtler, S., Simpson, D., Schröder, S., Taraborrelli, D., Bott, A., & Schultz, M. (2018). Ozone
impacts of gas–aerosol uptake in global chemistry transport models. *Atmospheric chemistry and
physics*, *18*(5), 3147-3171.

Stevens, B., Giorgetta, M., Esch, M., Mauritsen, T., Crueger, T., Rast, S., ... & Roeckner, E. (2013).
Atmospheric component of the MPI-M Earth system model: ECHAM6. *Journal of Advances in*
*Modeling Earth Systems*, *5*(2), 146-172.

Stevenson, D. S., Dentener, F. J., Schultz, M. G., Ellingsen, K., van Noije, T. P. C., Wild, O.,
Zeng, G., Amann, M., Atherton, C. S., Bell, N., Bergmann, D. J., Bey, I., Butler, T., Cofala, J.,
Collins, W. J., Derwent, R. G., Doherty, R. M., Drevet, J., Eskes, H. J., Fiore, A. M., Gauss, M.,
Hauglustaine, D. A., Horowitz, L. W., Isaksen, I. S. A., Krol, M. C., Lamarque, J. F., Lawrence,
M. G., Montanaro, V., Muller, J. F., Pitari, G., Prather, M. J., Pyle, J. A., Rast, S., Rodriguez, J.
M., Sanderson, M. G., Savage, N. H., Shindell, D. T., Strahan, S. E., Sudo, K., and Szopa, S.:
Multimodel ensemble simulations of present-day and near-future tropospheric ozone, J. Geophys.
Res., 111, D08301, https://doi.org/10.1029/2005jd006338, 2006.

Stevenson, D. S., Dentener, F. J., Schultz, M. G., Ellingsen, K., van Noije, T. P. C., Wild, O.,
Zeng, G., Amann, M., Atherton, C. S., Bell, N., Bergmann, D. J., Bey, I., Butler, T., Cofala, J.,
Collins, W. J., Derwent, R. G., Doherty, R. M., Drevet, J., Eskes, H. J., Fiore, A. M., Gauss, M.,
Hauglustaine, D. A., Horowitz, L. W., Isaksen, I. S. A., Krol, M. C., Lamarque, J. F., Lawrence,
M. G., Montanaro, V., Muller, J. F., Pitari, G., Prather, M. J., Pyle, J. A., Rast, S., Rodriguez, J.
M., Sanderson, M. G., Savage, N. H., Shindell, D. T., Strahan, S. E., Sudo, K., and Szopa, S.:
Multimodel ensemble simulations of present-day and near-future tropospheric ozone, J. Geophys.
Res., 111, D08301, https://doi.org/10.1029/2005jd006338, 2006.

Surl, L., Palmer, P. I., & González Abad, G. (2018). Which processes drive observed variations of
HCHO columns over India?. *Atmospheric Chemistry and Physics*, *18*(7), 4549-4566.

Tao, M., Fiore, A. M., Jin, X., Schiferl, L. D., Commane, R., Judd, L. M., Janz, S., Sullivan, J. T.,
Miller, P. J., Karambelas, A., Davis, S., Tzortziou, M., Valin, L., Whitehill, A., Civerolo, K., and
Tian, Y.: Investigating changes in ozone formation chemistry during summertime pollution vents
over the northeastern United States, Environ. Sci. Technol., 56, 15312–15327,
https://doi.org/10.1021/acs.est.2c02972, 2022.

Tarasick, D. W., Carey-Smith, T. K., Hocking, W. K., Moeini, O., He, H., Liu, J., Osman, M. K.,
Thompson, A. M., Johnson, B. J., Oltmans, S. J., and Merrill, J. T.: Quantifying stratosphere-
troposphere transport of ozone using balloon-borne ozonesondes, radar windprofilers and
trajectory models, Atmos. Environ., 198, 496–509,
https://doi.org/10.1016/j.atmosenv.2018.10.040, 2019.

Veefkind, J. P., Aben, I., McMullan, K., Förster, H., de Vries, J., Otter, G., et al. (2012). TROPOMI
on the ESA Sentinel-5 Precursor: A GMES mission for global observations of the atmospheric composition for climate, air quality and ozone layer applications. Remote Sensing of Environment, 120(2012), 70–83. https://doi.org/10.1016/j.rse.2011.09.027

Vignati, E., Wilson, J., and Stier, P.: M7: An efficient size-resolved aerosol microphysics module for large-scale aerosol transport models, J. Geophys. Res.-Atmos., 109, D22202, https://doi.org/10.1029/2003jd004485, 2004

Wang, H., Lu, X., Jacob, D. J., Cooper, O. R., Chang, K.-L., Li, K., Gao, M., Liu, Y., Sheng, B., Wu, K., Wu, T., Zhang, J., Sauvage, B., Nédélec, P., Blot, R., and Fan, S.: Global tropospheric ozone trends, attributions, and radiative impacts in 1995–2017: an integrated analysis using aircraft (IAGOS) observations, ozonesonde, and multi-decadal chemical model simulations, Atmos. Chem. Phys., 22, 13753–13782, https://doi.org/10.5194/acp-22-13753-2022, 2022.

Wang, W., van der A, R., Ding, J., van Weele, M., and Cheng, T.: Spatial and temporal changes of the ozone sensitivity in China based on satellite and ground-based observations, Atmos. Chem. Phys., 21, 7253–7269, https://doi.org/10.5194/acp-21-7253-2021, 2021.

Wang, Y., Lampel, J., Xie, P., Beirle, S., Li, A., Wu, D., & Wagner, T. (2017). Ground-based MAX-DOAS observations of tropospheric aerosols, NO 2, SO 2 and HCHO in Wuxi, China, from 2011 to 2014. *Atmospheric Chemistry and Physics*, *17*(3), 2189-2215

WMO, Meteorology – A three-dimensional science, World Meteorological Organisation, Bulletin 6, (Oct), 134–138, 1957.

Young, P. J., Naik, V., Fiore, A. M., Gaudel, A., Guo, J., Lin, M. Y., ... & Zeng, G. (2018). Tropospheric Ozone Assessment Report: Assessment of global-scale model performance for global and regional ozone distributions, variability, and trends. Elem Sci Anth, 6, 10.

Zhang Y, Cooper OR, Gaudel A, et al. Tropospheric ozone change from 1980 to 2010 dominated 492 by equatorward redistribution of emissions. Nature Geosci, 2016, 9: 875-879 493

Ziemke, J. R., Chandra, S., & Bhartia, P. K. (2005). A 25-year data record of atmospheric ozone in the Pacific from Total Ozone Mapping Spectrometer (TOMS) cloud slicing: Implications for ozone trends in the stratosphere and troposphere. *Journal of Geophysical Research: Atmospheres*, *110*(D15).

Ziemke, J. R., L. D. Oman, S. A. Strode, A. R. Douglass, M. A. Olsen, R. D. McPeters, P. K. Bhartia, L. Froidevaux, G. J. Labow, J. C. Witte, A. M. Thompson, D. P. Haffner, N. A. Kramarova, S. M. Frith, L. K. Huang, G. R. Jaross, C. J. Seftor, M. T. Deland, and S. L. Taylor, Trends in global tropospheric ozone inferred from a composite record of TOMS/OMI/MLS/OMPS

satellite measurements and the MERRA-2 GMI simulation, Atmos. Chem. Phys., 19, 3257-3269,
https://doi.org/10.5194/acp-19-3257-2019, 2019.

Ziemke, J. R., S. Chandra, B. N. Duncan, L. Froidevaux, P. K. Bhartia, P. F. Levelt, and J. W.
Waters, Tropospheric ozone determined from Aura OMI and MLS: Evaluation of measurements
and comparison with the Global Modeling Initiative's Chemical Transport Model, J. Geophys.
Res., 111, D19303, doi:10.1029/2006JD007089, 2006.

Supplementary table-1: Primary volatile organic compounds. BIGALKANE is a lumped species
for all alkanes C4 and greater, BIGENE lumps all alkenes C4 and greater.

| S.N. | VOCs name |
| --- | --- |
| 1 | Benzene |
| 2 | BIGALKANE |
| 3 | BIGENE |
| 4 | acetylene |
| 5 | ethene |
| 6 | ethanol |
| 7 | ethane |
| 8 | propene |
| 9 | propane |
| 10 | formaldehyde |
| 11 | acetaldehyde |
| 12 | acetone |
| 13 | acetic acid |
| 14 | methanol |
| 15 | methane |
| 16 | formic acid |
| 17 | butan-2-one |
| 18 | toluene |
| 19 | xylenes |